# WRF-Chem simulation of aerosol seasonal variability in the San Joaquin Valley

Longtao Wu[1], Hui Su[1], Olga V. Kalashnikova[1], Jonathan H. Jiang[1], Chun Zhao[2], Michael J. Garay[1], James R. Campbell[3] and Nanpeng Yu[4]

*1. Jet Propulsion Laboratory, California Institute of Technology, Pasadena, CA, USA*

*2. School of Earth and Space Sciences, University of Science and Technology of China, Hefei, Anhui, China*

*3. Naval Research Laboratory, Monterey, CA, USA*

*4. University of California, Riverside, Riverside, CA, USA*

Submitted to *Atmospheric Chemistry and Physics*

April, 2017

_________________________

*Corresponding author address:* Longtao Wu, 4800 Oak Grove Dr., Pasadena, CA 91109
E-mail: Longtao.Wu@jpl.nasa.gov

Highlights:
1.  The WRF-Chem simulation successfully captures aerosol variations in the cold season in the

20        San Joaquin Valley (SJV), but has poor performance in the warm season.

2.  High resolution model simulation can better resolve inhomogeneous distribution of

22        anthropogenic emissions in urban areas, resulting in better simulation of aerosols in the cold

23        season in the SJV.

3.  Observations show that dust is a major component of aerosols in the SJV, especially in the

warm season. Poor performance of the WRF-Chem model in the warm season is mainly due

to misrepresentation of dust emission and vertical mixing.

## Abstract

WRF-Chem simulations of aerosol seasonal variability in the San Joaquin Valley (SJV), California are evaluated by satellite and in-situ observations. Results show that the WRF-Chem model successfully captures the distribution, magnitude and variation of SJV aerosols during the cold season. However, aerosols are not well represented in the warm season. Aerosol simulations in urban areas during the cold season are sensitive to model horizontal resolution, with better simulations at 4 km resolution than at 20 km resolution, mainly due to inhomogeneous distribution of anthropogenic emissions and better represented precipitation in the 4 km simulation. In rural areas, the model sensitivity to grid size is rather small. Our observational analysis reveals that dust is a primary contributor to aerosols in the SJV, especially during the warm season. Aerosol simulations in the warm season are sensitive to parameterization of dust emission in WRF-Chem. The GOCART (Goddard Global Ozone Chemistry Aerosol Radiation and Transport) dust scheme produces very little dust in the SJV while the DUSTRAN (DUST TRANsport model) scheme overestimates dust emission. Vertical mixing of aerosols is not adequately represented in the model based on CALIPSO (Cloud-Aerosol Lidar and Infrared pathfinder Satellite Observation) aerosol extinction profiles. Improved representation of dust emission and vertical mixing in the boundary layer are needed for better simulations of aerosols during the warm season in the SJV.

## 1. Introduction

The San Joaquin Valley (SJV) in the southern portion of the California Central Valley is

surrounded by coastal mountain range to the west and the Sierra Nevada range to the east. With
cool wet winters and hot dry summers, the unique natural environment makes SJV one of the most
productive agricultural regions in the world (SJV APCD, 2012 and references therein). However,
SJV is also one of the most polluted regions in US due to its unique geographical location. Frequent
stagnant weather systems are conducive to air pollution formation, while the surrounding
mountains block air flow and trap pollutions. Large seasonal and spatial variation of aerosol
occurrence and distribution are observed in the SJV. Although significant progress made to
improving local air quality in past decades has been achieved through strong emission controls,
PM2.5 (particulate matter with diameter $\leq 2.5$ μm) concentrations in the SJV remain well above
the national ambient air quality standards (NAAQS) threshold of 12 μg m$^{-3}$ on an annual basis and
35 μg m$^{-3}$ on daily basis, occurring mainly during the cold season. Improved understanding of the
aerosol variability and impacts is needed to provide further guidance for emission control strategies
in the SJV.

Air quality models are a useful tool to understanding the formation and evolution of

aerosols and their impacts on air quality, weather and climate. However, it is quite challenging to
accurately simulate aerosol properties (Fast et al., 2014). Fast et al. (2014) summarized the factors
contributing to the errors in regional-scale modeling of aerosol properties. They include 1)
emission sources; 2) meteorological parameterizations; 3) representation of aerosol chemistry; 4)
limited understanding of the formation processes of secondary organic aerosol (SOA); 5) spatial
resolution; and 6) boundary conditions.

As one of the advanced regional air quality models available presently to the community,

the Weather Research and Forecasting model with Chemistry (WRF-Chem) has been widely used
to study aerosols and their impacts on regional air quality, weather and climate (e.g., Misenis and
Zhang, 2010; Zhang et al., 2010; Zhao et al., 2010; 2013a, 2013b; 2014; Wu et al., 2011a, 2011b,
2013; Fast et al., 2012, 2014; Scarino et al., 2014; Tessum et al., 2015; Campbell et al., 2016; Hu
et al., 2016). For example, Fast et al. (2014) showed that WRF-Chem simulations at 4 km
horizontal resolution captured the observed meteorology and boundary layer structure over
California in May and June of 2010 and the spatial and temporal variations of aerosols were
reasonably simulated. Aerosol simulations by WRF-Chem are usually sensitive to both local
emission and long-range transport of aerosols from the boundary conditions provided by the global
Model for Ozone and Related chemical Tracers, version 4 (MOZART-4). With a similar model
set-up, Zhao et al. (2013b) conducted a one-year simulation at 12 km horizontal resolution and
found that the WRF-Chem model represented the observed seasonal and spatial variation of
surface particulate matter (PM) concentration over California. However, underestimation of
elemental carbon (EC) and organic matter (OM) were noticed in the model simulation, with weak
sensitivity to horizontal resolution.

In this study, we focus on simulating aerosol seasonal variability in the SJV, California

using similar model configurations as that used in Zhao et al. (2013b) and Fast et al. (2014). This
paper serves as the first step for future investigation of the aerosol impact on regional climate and
the water cycle in California. Previous studies have demonstrated that aerosols are better simulated
at higher model resolution (Misenis and Zhang et al., 2010; Qian et al., 2010; Stround et al., 2011;
Fountoukis et al., 2013). However, most regional climate studies are still performed with coarse
model resolutions (on the order of 10 km) due to the availability of computational resources. This

90 study will investigate the sensitivity of aerosol simulations to horizontal resolution and identify

91 optimal model physical choices for reasonable representation of aerosol variabilities in the SJV.

92   Another application of air quality modeling is to provide initial *a priori* fields for remote

93 sensing retrievals. The WRF-Chem model has been proposed as an input for retrieval algorithms

94 to be developed for the recently-selected NASA (National Aeronautics and Space Administration)

95 MAIA (Multi-Angle Imager for Aerosols) mission, which aims to map PM component

96 concentrations in major urban areas (including the SJV, a testbed for the MAIA retrieval algorithm

97 development). A significant challenge for aerosol remote sensing in retrieving spatial information

98 on specific aerosol types, especially near the surface, is caused by the lack of information on the

99 vertical distribution of aerosols in the atmospheric column and limited instrument sensitivity to

100 aerosol types over land. The WRF-Chem model will be used to provide near-real-time estimation

101 of particle properties, aerosol layer heights, and aerosol optical depths (AOD) to constrain the

102 instrument-based PM retrievals. A reasonable estimate of aerosol properties from WRF-Chem is

103 critical to ensuring retrieval speed and quality. Considering the sensitivity of WRF-Chem

104 simulations to various factors such as initial and boundary conditions, model parameterizations

105 and emission sources (e.g., Wu and Petty, 2010; Zhao et al., 2010, 2013a, 2013b; Wu et al., 2011a,

106 2015; Fast et al., 2014; Campbell et al., 2016; Morabito et al., 2016), careful model evaluations

107 are needed before the simulations can be used operationally for remote sensing retrievals. Thus,

108 this study is important for the development of MAIA retrieval algorithms, critical to the success

109 of the MAIA mission.

110   This paper is organized as follows. Section 2 describes observational datasets used for

111 model evaluation. Section 3 provides the description of the WRF-Chem model and experiment

setup. Model simulations and their comparison with observations are discussed in section 4.
Section 5 presents the conclusions.

## 114    2. Observations

### 115    2.1 Column-integrated Aerosol Optical Properties

AOD is a measure of column-integrated light extinction by aerosols and a proxy for total
aerosol loading in the atmospheric column. The Aerosol Robotic Network (AERONET) provides
ground measurements of AOD every 15 minutes during daytime under clear skies (Holben et al.,
1998), with an accuracy approaching $\pm0.01$ (Eck et al., 1999; Holben et al., 2001; Chew et al.,
2011). The monthly level 2.0 AOD product with cloud screening and quality control is used in this
study. Ångström exponent (AE) is an indicator of aerosol particle size. Small (large) AE values
are generally associated with large (small) aerosol particles (Ångström, 1929; Schuster et al.,
2006). The AE between 0.4 μm and 0.6 μm is derived from AERONET observed AODs, and is
used to evaluate the model-simulated AE. For comparison with simulated AOD, AERONET AOD
is interpolated to 0.55 μm from 0.50 μm and 0.675 μm using the AE. In the SJV, only one
AERONET station at Fresno, CA (36.79°N, 119.77°W) has regular observations throughout the
California water year 2013 (WY2013) from October 2012 to September 2013.
The Multiangle Imaging Spectroradiometer (MISR) (Diner et al., 1998) instrument
onboard the Terra satellite has provided global coverage of AOD once a week since December
1999. The standard MISR retrieval algorithm provides AOD observations at 17.6 km resolution
using 16x16 pixels of 1.1 km x 1.1 km each. About 70% of MISR AOD retrievals are within 20%
of the paired AERONET AOD, and about 50% of MISR AOD falls within 10% of the AERONET
AOD, except in dusty and hybrid (smoke+dust) sites (Kahn et al., 2010). We use version 22 of
Level 3 monthly AOD product at 0.5° resolution in this study.
**2.2 Surface Mass Concentration**
Surface $PM_{2.5}$ speciation and $PM_{10}$ (particulate matter with diameter $\leq$ 10 μm) data are
routinely collected by two national chemical speciation monitoring networks: Interagency
Monitoring of Protected Visual Environments (IMPROVE) and the $PM_{2.5}$ National Chemical
Speciation Network (CSN) operated by Environmental Protection Agency (EPA) (Hand et al.
2011; Solomon et al., 2014). IMPROVE collects 24-h aerosol speciation every third day at mostly
rural sites since 1988. The same frequency of aerosol speciation dataset was collected at EPA CSN
sites in urban and suburban areas since 2000. The observed organic carbon is converted to OM by
multiplying by 1.4 (Zhao et al., 2013b; Hu et al., 2016). Some precursors of aerosol pollutions
(such as $NO_2$ and $SO_2$) are observed hourly by EPA (data available at:
https://aqsdr1.epa.gov/aqsweb/aqstmp/airdata/download_files.html) and are used in this study.
Selected IMPROVE and EPA CSN sites used in this study are shown in Figure 1a.
**2.3 Aerosol Extinction Profile**
The aerosol extinction coefficient profile reflects the attenuation of the light passing
through the atmosphere due to the scattering and absorption by aerosol particles as a function of
range. Version 3 Level 2 532 nm aerosol extinction profiles derived from Cloud-Aerosol Lidar
with Orthogonal Polarization (CALIOP) backscatter profiles collected onboard the Cloud-Aerosol
Lidar and Infrared pathfinder Satellite Observation (CALIPSO) satellite are used (Omar et al.,
2009; Young and Vaughan, 2009). Seasonal mean profiles are derived for WY2013 based on the
methodology outlined in Campbell et al. (2012), whereby quality-assurance protocols are applied
to individual profiles before aggregating and averaging the data. We highlight that no individual
profiles are included in the averages if the CALIOP Level 2 retrieval failed to resolve any
extinction within the column, a potential issue to create bias that has recently been described by
Toth et al. (2017). Level 2 532 nm aerosol extinction data classify aerosols into 6 types: clean
marine, dust, polluted continental, clean continental, polluted dust and smoke. Dust and polluted
dust are distinguished in the averages in this study for their contribution to total extinction and the
vertical profile seasonally in the SJV.
**2.4 Meteorology**
AIRS (Atmospheric Infrared Sounder) onboard the Aqua satellite (Susskind et al., 2003;
Divakarla et al., 2006) has provided global coverage of the tropospheric temperature and moisture
at approximately 01:30 and 13:30 local time since 2002. AIRS retrievals have root-mean-squared
(RMS) error of ~1 K for temperature and ~15% for water vapor (Divakarla et al., 2006). Level 3
monthly temperature and moisture retrievals (version 6) at 1° x 1° grid are used in this study.
Vertical gradient of equivalent potential temperature ($\theta_e$) marks atmospheric stability and is
computed from temperature and moisture profiles observed by AIRS. Vertical profiles from the
European Center for Medium-Range Weather Forecasts Interim Re-Analysis (ERA-Interim; Dee
et al., 2011) are also used for comparison. Surface observations, including air temperature, relative
humidity (RH) and wind speed, are routinely collected at the California Irrigation Management
Information System (CIMIS; http://www.cimis.water.ca.gov/). Precipitation used in this study is
the Climate Prediction Center (CPC) Unified Gauge-Based Analysis of Daily Precipitation product
at 0.25° x 0.25° resolution.
**3. Model Description and Experiment Setup**
The WRF-Chem model Version 3.5.1 (Grell et al., 2005) updated by Pacific Northwest
National Laboratory (PNNL) is used in this study (Zhao et al., 2014). This study uses the CBM-Z
(carbon bond mechanism) photochemical mechanism (Zaveri and Peters, 1999) coupled with the
sectional-bin MOSAIC (Model for Simulating Aerosol Interactions and Chemistry) aerosol
scheme (Zaveri et al., 2008) as the chemical driver. The major components of aerosols (nitrate,
ammonium, EC, primary OM, sulfate, sea salt, dust, water and other inorganic matter) as well as
their physical and chemical processes are simulated in the model. For computational efficiency,
aerosol particles in this study are partitioned into four-sectional bins with dry diameter within
0.039-0.156 μm, 0.156-0.625 μm, 0.625-2.5 μm, and 2.5-10.0 μm. Zhao et al. (2013a) compared
the impact of aerosol size partition on dust simulations. It showed that the 4-bin approach
reasonably produces dust mass loading and AOD compared with the 8-bin approach. The size
distribution of the 4-bin approach follows that of the 8-bin approach with coarser resolution,
resulting in ±5% difference on the ratio of $PM_{2.5}$-dust/$PM_{10}$-dust in dusty regions (more large
particles and less small particles). Dust number loading and absorptivity are biased high in the 4-
bin approach compared with the 8-bin approach.

Aerosols are considered to be spherical and internally mixed in each bin (Barnard et al.,

2006; Zhao et al., 2013b). The bulk refractive index for each particle is calculated by volume
averaging in each bin. Mie calculations as described by Ghan et al. (2001) are used to derive
aerosol optical properties (such as extinction, single-scattering albedo, and the asymmetry
parameter for scattering) as a function of wavelength. Aerosol radiation interaction is included in
the shortwave and longwave radiation schemes (Fast et al., 2006; Zhao et al., 2011). By linking
simulated cloud droplet number with shortwave radiation and microphysics schemes, aerosol
cloud interaction is effectively simulated in WRF-Chem (Chapman et al., 2009). Aerosol snow
interaction is implemented in this version of WRF-Chem (Zhao et al., 2014) by considering aerosol
deposition on snow and the subsequent radiative impacts through the SNICAR (SNow, ICe, and
Aerosol Radiative) model (Flanner and Zender, 2005, 2006).
The model simulations start on 1 September 2012 and run continuously for 13 months.
With the first month used for the model spin-up, our analysis focuses on WY2013 from October
2012 to September 2013. The model is configured with 40 vertical levels and a model top at 50
hPa. The vertical resolution from the surface to 1 km gradually increases from 28 m to 250 m. The
model center is placed at 38°N, 121°W, with 250 x 350 grid points at 4 km horizontal resolution
(referred to as "4km" hereafter; Table 1), covering California and the surrounding area. To test the
sensitivity of the aerosol simulations to horizontal resolution, one simulation with the same model
settings and domain coverage is conducted at 20 km horizontal resolution (referred to as "20km"
hereafter).
The physics parameterizations used in the simulations include the Morrison double-
moment microphysics scheme (Morrison et al., 2009), Rapid Radiative Transfer Model for General
circulation model (RRTMG) shortwave and longwave radiation schemes (Iacono et al., 2008),
Community Land Model (CLM) Version 4 land surface scheme (Lawrence et al., 2011). The
Yonsei University (YSU) planetary boundary layer (PBL) scheme (Hong et al., 2006) is used in
all of the simulations, except one sensitivity experiment that uses the ACM2 (Asymmetric
Convective Model with non-local upward mixing and local downward mixing; Pleim, 2007) PBL
scheme (referred to as "20km_P7" hereafter, Table 1). Previous studies showed that both YSU and
ACM2 schemes have good performance in simulating boundary layer properties (e.g., Hu et al.,
2010; Xie et al., 2012; Cuchiara et al., 2014; Banks and Baldasano, 2016; Banks et al., 2016; Chen
et al., 2017). Subgrid convection, convective transport of chemical constituents and aerosols, and
wet deposition from subgrid convection are parameterized using the Grell 3D ensemble cumulus
scheme (Grell and Devenyi, 2002) in the 20 km simulations while convective processes are
resolved in the 4 km simulations. The ERA-Interim reanalysis serves as initial and boundary
meteorological conditions for WRF-Chem. The MOZART-4 global chemical transport model
(Emmons et al., 2010) is used for initial and boundary chemical conditions. Fast et al. (2014) found
that the MOZART-4 model overestimates aerosols in the free troposphere over California, which
is also found in one of our sensitivity experiments ("20km_BC1" in the supplementary). Following
Fast et al. (2014), the chemical initial and boundary conditions from MOZART-4 are divided by
two in all simulations except 20km_BC1.

Anthropogenic emissions are provided by US EPA 2005 National Emissions Inventory

(NEI05), with area-type emissions on a structured 4-km grid and point-type emissions at specific
latitude and longitude locations (US EPA, 2010). Nineteen gases (including $SO_2$, $NO$, $NH_3$ etc.)
are emitted, and aerosol emissions include $SO_4$, $NO_3$, EC, organic aerosols, and total $PM_{2.5}$ and
$PM_{10}$ masses. Anthropogenic emissions are updated every hour to account for diurnal variability,
while its seasonal variation is not considered in the simulations. A sensitivity experiment with
2011 NEI emissions ("20km_NEI11" in the supplementary) does not produce significantly
different results from the 2005 NEI emissions. Biogenic emissions are calculated online using the
Model of Emissions of Gases and Aerosols from Nature (MEGAN) model (Guenther et al., 2006).
Biomass burning emissions are obtained from the Global Fire Emissions Database version 2.1,
with eight-day temporal resolution (Randerson et al., 2007) and updated monthly. Sea salt
emissions are derived from the PNNL-updated sea salt emission scheme that includes the
correction of particles with radius less than 0.2 μm (Gong et al., 2003) and dependence on sea
surface temperature (Jaeglé et al., 2011).

Following Zhao et al. (2013b), dust emission is computed from the GOCART (Goddard

Global Ozone Chemistry Aerosol Radiation and Transport) dust scheme (Ginoux et al., 2001) in
the 20km and 4km simulations. The GOCART dust scheme estimates the dust emission flux $F$ as
$$F = CSs_p u_{10m}^2 (u_{10m} - u_t) \qquad ,$$
where $C$ is an empirical proportionality constant, $S$ is a source function for potential wind erosion
that is derived from 1° x 1° GOCART database (Freitas et al., 2011), $s_p$ is a fraction of each size
class dust in emission, $u_{10m}$ is 10-m wind speed and $u_t$ is a threshold speed for dust emission.
As shown later, a significant amount of dust is observed in the SJV, whereas the GOCART
dust scheme produces little dust. Two sensitivity experiments at 20 km and 4 km horizontal
resolution (hereafter referred to as "20km_D2" and "4km_D2", respectively) are conducted by
switching the dust emission scheme to the DUST TRANsport model (DUSTRAN) scheme (Shaw
et al., 2008). The DUSTRAN scheme estimates $F$ as
$$F = \alpha C u_*^4 (1 - \frac{f_w u_{*t}}{u_*}) \qquad ,$$
where $C$ is an empirical proportionality constant, $\alpha$ is the vegetation mask, $u_*$ is the friction
velocity, $u_{*t}$ is a threshold friction velocity and $f_w$ is the soil wetness factor. The $C$ value in both
GOCART and DUSTRAN is highly tunable for different regions. The original $C$ values, 1.0 μg s$^2$
m$^{-5}$ in GOCART (Ginoux et al., 2001) and 1.0×10$^{-14}$ g cm$^{-6}$ s$^{-3}$ in DUSTRAN (Shaw et al., 2008),
are used in this study.

## 264  4. Model Simulation Results

Shown in Fig. 1a, our model domain includes three urban sites (Fresno, Bakersfield and
Modesto) and two rural sites (Pinnacles and Kaiser) where surface measurements of aerosols are
available. Because aerosols properties and model performance are similar at all urban sites, our
discussion is focused on the results at Fresno and the simulations for other urban sites are provided
in the supplementary materials. Model simulations in the rural areas are presented in the last
subsection.

## 271  4.1 Sensitivity to Horizontal Resolution

Figure 1 features daily mean anthropogenic $PM_{2.5}$ emission rates used in the 20km and

4km simulations, respectively. Although both emission rates are derived from the 4 km NEI05
dataset, localized high emission rates with sharp gradients are evident in urban areas from the 4km
simulation (Fig. 1b). The 20km simulation exhibits lower emission rates at the urban areas with
weaker gradients due to the reapportionment process (Fig. 1a). As precipitation is an important
process that removes aerosols, we examine the simulated precipitation for the 20km and 4km runs
and find that the 20km simulation produces 51% more precipitation, although the domain averaged
precipitation is lower in the 20km run than the 4km run (Fig. 2a).

Consistent with higher emission rates and lower precipitation at Fresno, the 4km run

simulates higher AOD than the 20km run in the cold season (October-November-December and
January-February-March; OND and JFM in Fig. 3). Averaged over a broad area encompassing
Fresno and Bakersfield, the most polluted region in the SJV (red box in Fig. 1a), the AOD is 0.090
in the 4km and 0.073 in the 20km, a 23% difference. Compared to the MISR observations, the
4km simulation reproduces the spatial distribution and magnitude of AOD in the cold season.
However, the AOD difference between the 20km and 4km runs is small in the warm season (April-
May-June and July-August-September; AMJ and JAS in Fig. 3), and both runs underestimate AOD
by ~50% with respect to the MISR observations.

Comparing the point values at Fresno in the 4km and 20km simulations (Fig. 4a), we find

similar results: the 4km AOD is closer to the AERONET measurements and is about 23% higher
than that in the 20km run during the cold season, while both runs are biased low in AOD during
the warm season. The different model sensitivities to horizontal resolution between the cold and
warm seasons suggest that the dominant aerosol sources may be different for the two seasons. We
will elaborate upon the aerosol composition in the following section. MISR and AERONET
observations display weak seasonal AOD variation in the SJV and at Fresno, respectively, which
is not well represented in the 20km and 4km simulations (Fig. 3 and 4a).
Aside from AOD, significant seasonal variability of AE (Fig. 4b) is shown at Fresno. AE
exhibits a maximum about 1.50 in January and a minimum of 0.98 in April, suggesting relatively
small particles in the winter and large particles in the spring. A relatively large AE value of 1.40
(corresponding to small particles) is observed in July, possibly related to the wild fires in late July
in the SJV. WRF-Chem captures the seasonal variability of the AE well, with a correlation of 0.90
in both the 20km and 4km simulations. The magnitude of AE is also approximately simulated in
the cold season, with a mean of 1.15 (1.20) in the 20km (4km) runs compared to 1.33 in the
observation. However, the simulated AE is underestimated by ~30% in the warm season,
indicating that the simulated particle size is biased high during this period.
Significant seasonal variability of $PM_{2.5}$ is observed in the SJV urban areas (Fig. 5a and
Supplementary Fig. 4a and 5a). $PM_{2.5}$ at Fresno peaks in January (26.18 μg m$^{-3}$) and reaches a
minimum of 7.03 μg m$^{-3}$ in June, with an annual nonattainment value of 12.64 μg m$^{-3}$ (Fig. 5a).
Both the 20km and 4km runs approximately capture the observed seasonal variability of $PM_{2.5}$,
with a correlation around 0.90 (Table 2). In the cold season, the 4km simulation overestimates
$PM_{2.5}$ by 27% while the 20km simulation exhibits a low bias of 19% compared with IMPROVE
observations at Fresno (Table 3). The 4km simulation of $PM_{10}$ is in good agreement with
IMPROVE in the winter (December, January and February), but has significant low biases of
between 30% and 85% in other months (Fig. 5b). The 20km simulation underestimates $PM_{10}$
throughout WY2013.
$PM_{2.5}$ is a mixture of nitrate ($NO_3$), ammonia ($NH_4$), OM, EC, sulfate ($SO_4$), dust and other
aerosols. High concentrations of $PM_{2.5}$ are primarily the result of $NO_3$ at Fresno (Fig. 5c). Both
simulations produce the seasonal variability of $NO_3$ with a correlation of 0.94, but high bias of 17%
(75%) is found in the 20km (4km) simulations during the cold season. As one precursor of $NO_3$,
$NO_2$ is underestimated by 43% in the 20km run (Fig. 6a). The overestimation in $NO_3$ and
underestimation in $NO_2$ suggest that the precursor emissions may not the reason for the high biases
in $NO_3$. $NH_4$ shows a similar performance to $NO_3$, with an overestimation by 38% (111%) in the
20km (4km) runs during the cold seasons (Fig. 5d). As shown later in section 4.3, both $NO_3$ and
$NH_4$ simulations are quite sensitive to the PBL scheme applied.

OM, the second largest contributing species to cold season $PM_{2.5}$ in the SJV (Table 3), is

significantly underestimated by 82% in the 20km simulation (Fig. 5f). The 4km simulation
produces higher OM, but it is still lower than the IMPROVE observations by 63%. The
underestimation of OM is expected, because SOA processes are not included in our model
infrastructure. Fast et al. (2014) used the simplified two-product volatility basis set
parameterization to simulate equilibrium SOA partitioning in WRF-Chem although SOA was still
underestimated in their simulation. It remains ongoing research how to correctly represent SOA
processes in regional climate models.

Both the 20km and 4km simulations reproduce the seasonal variability of EC, with a

correlation of 0.98 between the modeled and observed time series (Table 2). The 20km simulation
underestimates EC by 52% (16%) in the cold (warm) season (Fig. 5e and Table 3). The 4km
simulated EC (1.12 μg m$^{-3}$) exhibits good agreement with IMPROVE (1.08 μg m$^{-3}$) in the cold
season, but overestimates EC by 53% in the warm season.

The seasonal variability of $SO_4$ at Fresno is very different from other $PM_{2.5}$ species. It peaks

in May at 1.35 μg m$^{-3}$ and reaches the minimum of 0.67 μg m$^{-3}$ in August (Fig. 5g). The 20km
simulated $SO_4$ exhibits good correlation of 0.63 with the observation (Table 2), but is biased low
by 28% to 63% throughout WY2013 (Fig. 5g). Although the observed $SO_2$, the precursor of $SO_4$,
has approximately similar seasonal variation to the observed $SO_4$ (Fig. 6b), the 20km simulated
seasonal variability of $SO_2$ resembles other anthropogenic emissions, with high values in the cold
season and low values in the warm season, out of phase with the simulated $SO_4$ and the observed
$SO_2$. The 4km simulation produces higher $SO_4$ than the 20km run, resulting in better agreement
with the observation (0.82 μg m$^{-3}$ vs. 0.87 μg m$^{-3}$) during the cold season (Fig. 5g and Table 3).
However, the 4km run produces an increase of $SO_4$ by only 13% comparing to the 20km run in
the warm season, resulting in a correlation of -0.16 between the 4km simulation and the
observation.

To explore the possible cause for the underestimation of $SO_4$ and $SO_2$ in the warm season

in both the 20km and 4km simulations, we conduct a sensitivity experiment with different chemical
boundary conditions from the baseline runs (20km_BC1 in the supplementary). We find that $SO_4$
in the SJV is partly contributed to by marine intrusions (the different chemical boundary conditions
between 20km_BC1 and 20km_D2) throughout the year (supplementary Fig. 2g), as pointed out
by Fast et al. (2014). Including the marine intrusions, the 20km_BC1 simulated $SO_4$ tracks the
observation at a correlation of 0.78. Doubled chemical boundary conditions in the 20km simulation
results in 41% increase in $SO_4$ at Fresno, with a stronger increase in the warm season. Compared
to the observed $SO_4$ of 1.04 μg m$^{-3}$ in the warm season, the simulated $SO_4$ of 0.79 μg m$^{-3}$ in the
20km_BC1 run is closer to the observation than that simulated in the 20km_D2 run (0.53 μg m$^{-3}$).
The relative contributions of local emissions and remote transports (as well as other emission
sources, such as wild fires) to $SO_4$ concentrations in different seasons of the SJV require further
investigation.

Overall, the 4km simulation produces higher AOD and surface PM than the 20km

simulation in urban areas of the SJV, especially during the cold season, resulting in better

agreement with satellite and surface observations than the 20km simulation. Both the 20km and

4km simulations approximately capture the seasonal variability of $PM_{2.5}$ and most of its speciation.

However, significant low biases of AOD and $PM_{10}$ are found during the warm season in both

simulations. The underestimation also exists in a sensitivity experiment (not shown) with the same

model setups except initialized in April, indicating that the identified model biases during the warm

season are not caused by potential model drift after a relatively long simulation period. The

relatively good performance in simulating $PM_{2.5}$ but not $PM_{10}$ during the warm season suggests

that coarse aerosol particle mass (CM; 10 μm $\geq$ particulate matter with diameter $>$ 2.5 μm), mainly

dust in the SJV, is not properly represented in the model. The impact of dust parameterizations is

investigated in the 4km_D2 experiment.

**4.2 Sensitivity to Dust Scheme**

Limited amounts of $PM_{2.5}$_dust (dust with diameter $\leq$ 2.5 μm) are observed in the SJV cold

season, with a minimum of 0.37 μg m$^{-3}$ in December (Fig. 7a). The amount of $PM_{2.5}$_dust increases

in the warm season, with a peak of 3.86 μg m$^{-3}$ in September. The 4km simulation produces

comparable $PM_{2.5}$_dust relative to IMPROVE in the winter, but almost no dust in other months

(Fig. 7 and upper panel in Fig. 8). On the other hand, the dust emission rate in the 4km_D2 run is

significantly higher than the 4km run. We have found that the source function, *S*, for potential

wind erosion in the SJV is set to zero in the 1°x1° GOCART dataset used for the 4km simulation

(Fig. 9). An updated source function, *S*, at higher resolution is needed for the GOCART dust

scheme to correctly represent dust emissions in the SJV.

The 4km_D2 simulation reproduces the amount of $PM_{2.5}$\_dust in OND (Fig. 7a). However,
it overestimates $PM_{2.5}$\_dust by up to a factor of 3 in the warm season, resulting in an overestimation
of $PM_{2.5}$ by 52% (Fig. 7b and Table 3). $PM_{2.5}$\_dust is not sensitive to long-range transport (from
chemical boundary conditions in the model simulation; Supplementary Fig. 2h). Both the 4km and
4km_D2 simulations capture the seasonal variability of $PM_{2.5}$, but not that of $PM_{10}$ (Fig. 7c). The
magnitude of $PM_{10}$ in the 4km_D2 run is larger than the 4km simulation. $PM_{10}$ in the 4km_D2 run
is overestimated in April-May-June (AMJ) but underestimated in July-August-September (JAS),
leading to a comparable season mean of 38.12 μg m$^{-3}$ with IMPROVE observed 34.82 μg m$^{-3}$. The
overestimation of AMJ $PM_{10}$ and $PM_{2.5}$\_dust in the 4km_D2 run is likely associated with the high
bias in the simulated wind speed (Fig. 2b).
On the relative contribution of different aerosol species, IMPROVE observations at Fresno
show that $NO_3$ is the primary contributor (32.3%) to $PM_{2.5}$ while only 5.3% of $PM_{2.5}$ is dust in the
cold season (panel 1 of Fig. 10). Both the 4km and 4km_D2 runs roughly reproduce the relative
contributions to $PM_{2.5}$ in the cold season, with an overestimation of $NO_3$ and $NH_4$ and an
underestimation of OM, consistent with the time series in Fig. 5. Relative contributions of dust to
$PM_{2.5}$ are better simulated in the 4km_D2 run (7.3%) than the 4km one (<1.0%). IMPROVE shows
that 46.6% of $PM_{10}$ is CM in the cold season (panel 2 of Fig. 10). Both the 4km (6.3%) and
4km_D2 (20.6%) runs underestimate the contribution of CM to $PM_{10}$, mainly in October and
November. In the warm season, dust (24.6%) becomes the primary contributor to $PM_{2.5}$ while the
contribution from $NO_3$ decreases to 9.9% in IMPROVE observations (panel 3 of Fig. 10). Almost
no $PM_{2.5}$\_dust is simulated in the 4km run while too much $PM_{2.5}$\_dust is produced in the 4km_D2
(50.5%) run during the warm season. The relative contribution of CM to $PM_{10}$ is too small (27.6%)
in the 4km run, while the 4km_D2 run reflects an better relative contribution of 66.3% as compared
to an IMPROVE observed 75.8% (panel 4 of Fig. 10).

AOD simulations are improved in the 4km_D2 experiment (Fig. 11), with better agreement

found from MISR (Fig. 3) in AMJ. AOD (0.114) in the 4km_D2 run is comparable to observations
(0.131) in AMJ, but still underestimated by 53% in JAS. Consistent with AOD, the vertical
distribution of aerosol extinction is reasonably simulated during the cold season in the WRF-Chem
simulations, while large discrepancies are found in the warm season (Fig. 12). As observed by
CALIOP at 532 nm, aerosols are confined below 1 km in the cold season and decrease sharply
with height. During AMJ, aerosols are well mixed between the surface and the altitude of 1.5 km
and then decrease with height gradually. During JAS, the well-mixed aerosol layer is shallower
than that in AMJ and the vertical profile of aerosol extinction is in-between the cold season and
AMJ. Model simulations roughly capture the "bottom-heavy" structure of the extinction profiles
observed by CALIOP especially in the cold season, but significant biases exist. One common
problem for all four seasons is the low bias in the boundary layer and high bias in the free
atmosphere. Similar discrepancy between the model simulations and CALIOP is shown in other
studies (Wu et al., 2011a; Hu et al., 2016). The model does not capture the well-mixed aerosol
layer during AMJ. The difference in the aerosol extinction profiles between the 4km and 4km_D2
runs is small during the cold season.

Dust in the boundary layer is a primary factor contributing to aerosol extinction in the SJV,

as illustrated by the differences between the bulk seasonal CALIOP mean profile and those
excluding the contributions of the dust and polluted dust (CALIOP_nodust) profiles (Fig. 12).
Simulated aerosol extinction falls between the two in all seasons, suggesting that dust is the
primary factor contributing to the model biases in aerosol extinction. Although a small portion of
$PM_{2.5}$ is dust in the cold season, it contributes to about 50% of total aerosol extinction (Fig. 12a
and 12b). A predominant portion of aerosol extinction in the lower troposphere is contributed by
dust in the warm season (Fig. 12c and 12d). There, the 4km_D2 simulation produces higher aerosol
extinction between 0.3 km and 3 km than the 4km simulation, although it is still lower than
CALIOP. The simulated aerosol extinction in the free troposphere is close to or larger than
CALIOP, suggesting that aerosols transported from remote areas through chemical boundary
conditions (e.g., the differences between the 20km_BC1 and 20km_D2 runs in Supplementary Fig.
3) may not be the major factor contributing to the underestimation of dust between 0.3 km and 3
km in the SJV.

Overall, the poor simulations of dust play a dominant role in the low bias of aerosols in

the boundary layer during the warm season. Both the GOCART and DUSTRAN dust emission
schemes used in this study have difficulties in reproducing dust emissions in the SJV, with an
underestimation in GOCART and an overestimation in DUSTRAN (Fig. 7). Improvement on the
dust emission schemes is needed for capturing the seasonal variability of aerosols in the SJV.
**4.3 The Role of Meteorology**

The WRF-Chem simulations approximately reproduce the seasonal variations of

meteorological variables near the surface (correlations > 0.80), including temperature, RH, wind
speed and precipitation (Supplementary Fig. 6 and Supplementary Table 1). All of the model
simulations exhibit warm and dry biases near surface and in the boundary layer, with cold and wet
biases in the free atmosphere (Supplementary Fig. 6-8 and Supplementary Table 2). The dry bias
in the 4km_D2 run is about 10% near the surface throughout WY2013. Due to the relative dry
environment (RH<50%) in the warm season, the underestimation of boundary layer aerosol
extinction and column-integrated AOD is unlikely caused by the hygroscopic effects (Feingold
and Morley, 2003). In the cold season, the surface wind speed is underestimated by 0.67 m s$^{-1}$
(1.00 m s$^{-1}$) in the 4km_D2 (20km_D2) runs. In the warm season, the 4km_D2 run overestimates
wind speed by 0.78 m s$^{-1}$, while the 20km_D2 run has an underestimation of 0.16 m s$^{-1}$. These
results suggest that wind speed is not a major factor contributing to the low biases of aerosols in
the boundary layer abetween 0.3 km and 3 km. Furthermore, the seasonal variability of
precipitation is well captured in the simulations, while the magnitude of precipitation is weaker
than the observations during the warm season (Supplementary Table 2). Thus, we conclude that
wet removal processes would not be a primary reason for the aerosol biases in the warm season.
In the warm season, more aerosols are observed above 1.5 km than in the cold season (Fig.
12). A well-mixed layer of aerosols is observed below 1.5 km in AMJ (Fig. 12c), consistent with
the unstable lower troposphere below 1.5 km shown in AIRS and ERA-Interim (Fig. 13c). The
WRF-Chem model simulates neutral (or weakly stable) layers below 1.5 km, which may limit
uplifting of aerosols from the surface, failing to create a deep well-mixed layer of aerosols (Fig.
12c). Although the dust emission at the surface is overestimated in AMJ in the 4km_D2 run, the
simulated neutral or weakly stable thermal structure does not favor convective vertical mixing,
resulting in the low biases of aerosols between 0.3 km and 3 km.
Similar biases of aerosol and instability in the lower troposphere are also shown in JAS
(Fig. 12d and 13d). The stable boundary layer limits vertical transport of aerosols from the surface,
contributing to the low bias of column-integrated AOD in JAS (Fig. 11). In JAS (Fig. 12d), aerosol
extinction close to the CALIOP observation is simulated in the free atmosphere, suggesting that
the low bias in AOD is not due to the halved chemical boundary conditions from MOZART-4. In
the cold season, in spite of some discrepancies in the magnitude of atmospheric stability, all of the
simulations capture the stable lower troposphere (Fig. 13a and 13b), consistent with relatively
good performance of aerosol simulations in the cold season.

As biases in the model simulations are found mainly within the boundary layer, a sensitivity

experiment is conducted at 20 km resolution using the ACM2 PBL scheme (20km_P7). Although
the changes in the meteorological variables (Supplementary Fig. 6-9) and atmospheric static
stability (Fig. 13) are rather small, the simulated surface $NO_3$ and $NH_4$ in the 20km_P7 run
decrease by 50% compared to the 20km_D2 run (Fig. 14c, 14d and Table 3). Considering that
more $NO_3$ and $NH_4$ are simulated at 4 km resolution than at 20 km resolution as shown in section
4.1, the use of the ACM2 PBL scheme at 4 km simulation would largely resolve the high biases
of $NO_3$ and $NH_4$ in the 4km_D2 simulation. The decrease of $NO_3$ and $NH_4$ near the surface is
because more aerosols are transported to the layers above 0.5 km (Fig. 15a and 15b), possibly
resulting from different convective vertical mixing in the PBL schemes. However, $PM_{2.5}$_dust is
significantly overestimated by a factor of 4 in the 20km_P7 simulation (Fig. 14h), leading to a
small decrease of $PM_{2.5}$ by only 8% compared with the 20km_D2 run in the cold season. In the
warm season, $PM_{2.5}$_dust in the 20km_P7 run is overestimated by a factor of 5, causing an
overestimation of $PM_{2.5}$ and $PM_{10}$ (Fig. 14a and 14b). Aerosol extinctions in the boundary layer
above the surface increase in the warm season (Fig. 15c and 15d), possibly related to
overestimation of dust emissions and more conducive convective vertical transport in the PBL
scheme.

In summary, the WRF-Chem model captures the seasonal variations of meteorological

variables (temperature, RH, wind speed and precipitation), despite some deviations in magnitude.
The low biases in aerosol optical properties of the warm season likely do not originate from
hygroscopic effects, wet removal processes or dust emissions associated with the wind speed bias.
The model simulates a stable environment in the warm season, which is opposite to the observed
unstable environment. The simulated stable environment may be most likely responsible for low
biases in the aerosol extinction above the surface (0.3-3 km) and the column-integrated AOD in
the warm season. Switching to the ACM2 PBL scheme leads to improved vertical displacement of
aerosols in the boundary layer, thus an improvement in the simulations of $NO_3$ and $NH_4$ in the cold
season. However, dust emissions are significantly overestimated with the ACM2 PBL scheme,
which contributes partly to the better simulation of aerosol extinction in the boundary layer and
AOD in the column. These results highlight that improving the simulation of boundary layer
structure and processes are critical for capturing the vertical profiles of aerosol extinction.

## 4.4 Results in Rural Areas

In general, low values of PM concentration are observed in the rural areas, Pinnacles and
Kaiser (Fig. 16 and 17). The rural areas share some similar model performance to the urban areas,
such as the overestimation of $NO_3$, reasonable simulation of EC, good representation of $SO_4$ in the
cold season and underestimation of $SO_4$ in the warm season. However, the results are not sensitive
to model resolution. It suggests that high resolution is particularly important for heavily polluted
areas due to the inhomogeneity of emission sources, but less important for relatively lightly
polluted areas.
In late July/early August, MODIS (Moderate Resolution Imaging Spectroradiometer) fire
data (not shown) showed active wild fires close to Kaiser, which resulted in high concentration of
aerosols locally (Fig. 17). Our model simulations with monthly-varying fire emissions fail to
reproduce these fire events. Previous studies (e.g., Grell et al., 2011; Wu et al. 2011a; Archer-
Nicholls et al., 2015) demonstrated that the WRF-Chem model can capture aerosols distributions
from wild fires based on fire locations from satellite observations. Campbell et al. (2016) further
described the difficulties in constraining total aerosol mass from operational satellite fire
observations and the time needed by the model for diffusion within the near-surface layers to
render both reasonable AOD and vertical profiles of aerosol extinction. For operational application
of the WRF-Chem model in MAIA retrievals, the observations of daily fire events need to be more
appropriately considered.

## 526 5. Summary

The WRF-Chem (Weather Research and Forecasting model with Chemistry) model is
employed to simulate the seasonal variability of aerosols in WY2013 (water year 2013) in the SJV
(San Joaquin Valley). Model simulations are evaluated using satellite and in-situ observations. In
general, the model simulations conducted at 4 km resolution reproduce the spatial and temporal
variations of regional aerosols in the cold season, when aerosols are mainly contributed to by
anthropogenic emissions in the SJV. The magnitude of simulated aerosols in the cold season
however, especially in relatively dense urban areas, is sensitive to model horizontal resolution.
The 4km simulation has comparable magnitude to available observations, while the 20km
simulation underestimates aerosols. Differences in aerosol simulation fidelity as a function of
variable resolutions are mainly due to the difference in aerosol emissions and simulated
precipitation. Emissions at higher resolution can better resolve the inhomogeneity of
anthropogenic emissions in the SJV than at lower resolution. The sensitivity to horizontal
resolution is small in rural areas and during warm season, where/when the relative contribution of
anthropogenic emissions is small.
Previous studies in the SJV are mainly focused on $PM_{2.5}$ (particulate matter with diameter
$\leq 2.5$ μm) and during cold season (e.g. Chow et al., 2006; Herner et al., 2006; Pun et al., 2009;
Ying and Kleeman, 2009; Zhang et al., 2010; Chen et al., 2014; Hasheminassab et al., 2014; Kelly
et al., 2014; Baker et al., 2015; Brown et al., 2016). CALIOP (Cloud-Aerosol Lidar with
Orthogonal Polarization) and IMPROVE (Interagency Monitoring of Protected Visual
Environments) observations show that dust is a primary contributor to the aerosols in the SJV,
especially in the warm season. Dust contributes 24.6% to $PM_{2.5}$ while more than 75.8% to $PM_{10}$ in
the warm season. For all seasons, the major component of aerosol extinction in the boundary layer
is dust as observed by CALIOP, consistent with Kassianov et al. (2012). For a complete
understanding of aerosol impacts on air quality, weather and climate, the full spectrum of aerosols
should be considered during all seasons.

All the model simulations conducted fail to capture aerosol vertical distribution and

variability in the SJV warm season, largely due to the misrepresentation of dust emissions, static
stability and vertical mixing in the boundary layer. The GOCART (Goddard Global Ozone
Chemistry Aerosol Radiation and Transport) dust emission scheme significantly underestimates
dust due to the non-active source function, $S$, for potential wind erosion used in this study while
the DUSTRAN (DUST TRANsport model) scheme may overestimate dust emission in the SJV.
Along with the bias in dust emissions, our simulations produce a relatively stable boundary layer
in the warm season, in contrast with observations suggesting a more unstable environment, leading
to a weak vertical mixing of aerosols in the boundary layer. Improved dust emission and better
simulations of the boundary layer properties are needed for accurate simulation of aerosols in the
SJV warm season.

Other biases are also identified in the model simulations. $NO_3$ and $NH_4$ in the cold season

are overestimated in the model, but the results are sensitive to the choice of the PBL (planetary
boundary layer) scheme. The SOA (secondary organic aerosol) processes contribute to the
underestimation of OM (organic matter) in this study. The underestimation of sulfate in the warm
season may be caused by the misrepresentation of emissions and the chemical boundary conditions
related to marine intrusions. Aerosols from wild fires are not captured in the simulations with
monthly updated fire data. Further investigations are needed to improve model simulations in the
SJV for both scientific and operational applications.

## Acknowledgements

This study was carried out at the Jet Propulsion Laboratory, California Institute of
Technology, under a contract with the National Aeronautics and Space Administration. The
authors thank the funding support from the NASA ACMAP program and JPL PDF program. This
work is partially sponsored by California Energy Commission under grant #EPC-14-064. Author
JRC acknowledges the support of the NASA ACCDAM program and its manager Hal Maring.
The authors thank the three anonymous reviewers for their helpful comments.

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

A Federated Instrument Network and Data Archive for Aerosol Characterization, Remote
Sens. Environ., 66, 1–16, 1998.
Holben, B. N., Tanr, D., Smirnov, A., Eck, T. F., Slutsker, I., Abuhassan, N., Newcomb, W. W.,
Schafer, J. S., Chatenet, B., Lavenu, F., Kaufman, Y. J., Castle, J. V., Setzer, A., Markham,
B., Clark, D., Frouin, R., Halthore, R., Karneli, A., O'Neill, N. T., Pietras, C., Pinker, R. T.,
Voss, K., and Zibordi, G.: An emerging ground-based aerosol climatology: Aerosol optical
depth from AERONET, J. Geophys. Res., 106, 12067–12097, 2001.
Hong, S., Noh, Y., and Dudhia, J.: A new vertical diffusion package with an explicit treatment of
entrainment processes, Mon. Weather Rev., 134, 2318–2341, 2006.

Hu, X. M., Nielsen-Gammon, J.W., and Zhang, F.: Evaluation of three planetary boundary layer
schemes in the WRF model, J. Appl. Meteorol. Climatol., 49(9), 1831–1844,
doi:10.1175/2010JAMC2432.1, 2010.

Hu, Z., Zhao, C., Huang, J., Leung, L. R., Qian, Y., Yu, H., Huang, L., and Kalashnikova, O. V.:
Trans-Pacific transport and evolution of aerosols: evaluation of quasi-global WRF-Chem
simulation with multiple observations, Geosci. Model Dev., 9, 1725-1746, doi:10.5194/gmd-
9-1725-2016, 2016.

Iacono, M. J., Delamere, J. S., Mlawer, E. J., Shephard, M. W., Clough, S. A., and Collins, W. D.:
Radiative forcing by long-lived greenhouse gases: calculations with the AER radiative transfer
models, J. Geophys. Res., 113, D13103, doi:10.1029/2008JD009944, 2008.

Jaeglé, L., Quinn, P. K., Bates, T. S., Alexander, B., and Lin, J.-T.: Global distribution of sea salt
aerosols: new constraints from in situ and remote sensing observations, Atmos. Chem. Phys.,
11, 3137–3157, doi:10.5194/acp-11-3137-2011, 2011.

Kahn, R. A., Gaitley, B. J., Garay, M. J., Diner, D. J., Eck, T. F., Smirnov, A., and Holben, B. N.:
Multiangle Imaging SpectroRadiometer global aerosol product assessment by comparison with
the Aerosol Robotic Network, J. Geophys. Res., 115, D23209, doi:10.1029/2010JD014601,
2010.

Kassianov, E., Pekour, M., and Barnard, J.: Aerosols in central California: Unexpectedly large
contribution of coarse mode to aerosol radiative forcing, Geophys. Res. Lett., 39, L20806, doi:
10.1029/2012GL053469, 2012.

Kelly, J. T., Baker, K. R., Nowak, J. B., Murphy, J. G., Markovic, M. Z., VandenBoer, T. C., Ellis,
R. A., Neuman, J. A., Weber, R. J., and Roberts, J. M.: Fine-scale simulation of ammonium
and nitrate over the South Coast Air Basin and San Joaquin Valley of California during
CalNex-2010, J. Geophys. Res.-Atmos., 119, 3600–3614, 2014.

Lawrence, D. M., Oleson, K. W., Flanner, M. G., Thornton, P. E., Swenson, S. C., Lawrence, P.
760        J., Zeng, X., Yang, Z.-L., Levis, S., Sakaguchi, K., Bonan, G. B., and Slater, A. G.:
Parameterization improvements and functional and structural advances in version 4 of the

Community    Land    Model,    J.    Adv.    Model.    Earth    Sys.,    3,    M03001,    doi:
10.1029/2011MS000045,  2011.

Misenis,  C. and Zhang,  Y.: An examination  of sensitivity  of WRF/Chem predictions  to physical
parameterizations,  horizontal  grid  spacing,  and  nesting  options,  Atmos.  Res.,  97,  315–334,
doi:10.1016/j.atmosres.2010.04.005,  2010.

Morabito,  D., Wu, L., and Slobin,  S.: Weather  Forecasting  for Ka-band  Operations:  Initial  Study
Results,    IPN    PR    42-206,    pp.    1-24,    August    15,    2016.    Available    at:
http://ipnpr.jpl.nasa.gov/progress_report/42-206/206C.pdf,  2016.

Morrison,  H., Thompson,  G., and Tatarskii,  V.: Impact  of cloud  microphysics  on the  development
of trailing  stratiform  precipitation  in a simulated  squall  line:  comparison  of one-  and two-
moment  schemes, Mon. Weather  Rev., 137, 991–1007,  2009.

Omar,  A.H., Winker,  D.M., Kittaka,  C., Vaughan,  M.A., Liu,  Z., Hu,  Y., Trepte,  C.R., Rogers,
R.R., Ferrare,  R.A., Lee, K.P., Kuehn,  R.E., Hostetler,  C.A.: The CALIPSO  automated  aerosol
classification  and  lidar  ratio  selection  algorithm.  J.  Atmos.  Ocean.  Technol.  26,  1994–2014,
2009.

Pleim,  J. E.: A combined  local  and nonlocal  closure  model  for the atmospheric  boundary  layer.
Part I: Model  description  and testing, J. Appl. Meteorol.  Clim.,  46, 1383–1395,  2007.

Pun,  B. K., Balmori,  R. T. F., and Seigneur,  C.: Modeling  wintertime  particulate  matter  formation
in central  California,  Atmos. Environ.,  43, 402–409,  2009.

Qian,  Y., Gustafson Jr., W. I., and Fast,  J. D.: An investigation  of the sub-grid  variability  of trace
gases  and  aerosols  for  global  climate  modeling,  Atmos.  Chem.  Phys.,  10,  6917-6946,
doi:10.5194/acp-10-6917-2010,  2010.

Randerson,  J. T., van der Werf,  G. R., Giglio,  L., Collatz,  G. J., and Kasibhatla,  P. S.: Global  Fire
Emissions  Database, Version 2 (GFEDv2.1). Data set. Available  on-line  [http://daac.ornl.gov/]
from  Oak  Ridge  National  Laboratory  Distributed  Active  Archive  Center,  Oak  Ridge,
Tennessee,  U.S.A. doi:10.3334/ORNLDAAC/849,  2007.

San  Joaquin  Valley  Air  Pollution  Control  District:  2012  PM2.5  plan.  Available    from:
http://www.valleyair.org/Air_Quality_Plans/PM25Plans2012.htm,  2012.

Scarino, A. J., Obland, M. D., Fast, J. D., Burton, S. P., Ferrare, R. A., Hostetler, C. A., Berg, L.
K., Lefer, B., Haman, C., Hair, J. W., Rogers, R. R., Butler, C., Cook, A. L., and Harper, D.
B.: Comparison of mixed layer heights from airborne high spectral resolution lidar, ground-
based measurements, and the WRF-Chem model during CalNex and CARES, Atmos. Chem.
Phys., 14, 5547-5560, doi:10.5194/acp-14-5547-2014, 2014.
Shaw, W., Allwine, K. J., Fritz, B. G., Rutz, F. C., Rishel, J. P., and Chapman, E. G.: An evaluation
of the wind erosion module in DUSTRAN, Atmos. Environ., 42, 1907–1921, 2008.
Solomon, P. A., Crumpler, D., Flanagan, J. B., Jayanty, R. K. M., Rickman, E. E., and McDade C.
E.: U.S. National PM 2.5 Chemical Speciation Monitoring Networks – CSN and IMPROVE:
Description of Networks, J. Air Waste Manage., 64, 1410–1438,
doi:10.1080/10962247.2014.956904, 2014.
Susskind, J., Barnet, C. D., and Blaisdell, J.: Retrieval of atmospheric and surface parameters from
AIRS/AMSU/HSB data under cloudy conditions, IEEE Trans. Geosci. Remote Sens., 41(2),
390–409, doi:10.1109/TGRS.2002.808236, 2003.
Schuster, G. L., Dubovik, O., and Holben, B. N.: Angström exponent and bimodal aerosol size
distributions, J. Geophys. Res., 111, D07207, doi:10.1029/2005JD006328, 2006.
Tessum, C. W., Hill, J. D., and Marshall, J. D.: Twelve-month, 12 km resolution North American
WRF-Chem v3.4 air quality simulation: performance evaluation, Geosci. Model Dev., 8, 957-
973, doi:10.5194/gmd-8-957-2015, 2015.
Toth, T. D., Campbell, J. R., Reid, J. S., Tackett, J. L., Vaughan, M. A. and Zhang, J.: Lower
daytime threshold sensitivities to aerosol optical thickness in CALIPSO Level 2 products, J.
Geophys. Res., in review, 2017.
US Environmental Protection Agency, 2010: Technical Support Document: Preparation of
Emissions Inventories for the Version 4, 2005-based Platform, 73 pp., Office of Air Quality
Planning and Standards, Air Quality Assessment Division, available at:
https://www3.epa.gov/crossstaterule/pdfs/2005_emissions_tsd_07jul2010.pdf, 2010.
Wu, L., and Petty, G. W.: Intercomparison of Bulk Microphysics Schemes in Simulations of Polar
lows. Mon. Wea. Rev., 138, 2211-2228. doi: 10.1175/2010MWR3122.1, 2010.
Wu, L., Su, H. and Jiang, J. H.: Regional simulations of deep convection and biomass burning
over South America: 1. Model evaluations using multiple satellite data sets, J. Geophys. Res.,
116, D17208, doi:10.1029/2011JD016105, 2011a.

Wu, L., Su, H. and Jiang, J. H.: Regional simulations of deep convection and biomass burning
over South America: 2. Biomass burning aerosol effects on clouds and precipitation, J.
Geophys. Res., 116, D17209, doi:10.1029/2011JD016106, 2011b.

Wu, L., Su, H. and Jiang, J. H.: Regional simulations of aerosol impacts on precipitation during
the East Asian summer monsoon. J. Geophys. Res. Atmos., 118, doi: 10.1002/jgrd.50527,
2013.

Wu, L., Li, J.-L. F., Pi, C.-J., Yu, J.-Y., and Chen, J.-P.: An observationally based evaluation of
WRF seasonal simulations over the Central and Eastern Pacific, J. Geophys. Res. Atmos., 120,
doi:10.1002/2015JD023561, 2015.

Xie, B., Fung, J. C. H., Chan, A., and Lau, A.: Evaluation of nonlocal and local planetary boundary
layer schemes in the WRF model, J. Geophys. Res., 117, D12103, doi:10.1029/2011JD017080,
2012.

Ying, Q. and Kleeman, M. J.: Regional contributions to airborne particulate matter in central
California during a severe pollution episode, Atmos. Environ., 43, 1218–1228, 2009.

Young, S.A. and Vaughan, M.A.: The retrieval of profiles of particulate extinction from Cloud–
Aerosol Lidar Infrared Pathfinder Satellite Observations (CALIPSO) data: algorithm
description. J. Atmos. Ocean. Technol. 26, 1105–1119, 2009.

Zaveri, R. A. and Peters, L. K.: A new lumped structure photochemical mechanism for large-scale
applications, J. Geophys. Res., 104, 30387–30415, 1999.

Zaveri, R. A., Easter, R. C., Fast, J. D., and Peters, L. K.: Model for Simulating Aerosol
Interactions and Chemistry (MOSAIC), J. Geophys. Res., 113, D13204,
doi:10.1029/2007JD008782, 2008.

Zhang, Y., Liu, P., Liu, X.-H., Pun, B., Seigneur, C., Jacobson, M. Z., and Wang, W.-X.: Fine
scale modeling of wintertime aerosol mass, number, and size distributions in central California,
845        J. Geophys. Res.-Atmos., 115, D15207, doi:10.1029/2009jd012950, 2010.

Zhao, C., Liu, X., Leung, L. R., Johnson, B., McFarlane, S. A., Gustafson Jr., W. I., Fast, J. D.,
and Easter, R.: The spatial distribution of mineral dust and its shortwave radiative forcing over
North Africa: modeling sensitivities to dust emissions and aerosol size treatments, Atmos.
Chem. Phys., 10, 8821–8838, doi: 10.5194/acp-10-8821-2010, 2010.

Zhao, C., Liu, X., Ruby Leung, L., and Hagos, S.: Radiative impact of mineral dust on monsoon
precipitation variability over West Africa, Atmos. Chem. Phys., 11, 1879–1893,
doi:10.5194/acp-11-1879-2011, 2011.

Zhao, C., Chen, S., Leung, L. R., Qian, Y., Kok, J. F., Zaveri, R. A., and Huang, J.: Uncertainty in
modeling dust mass balance and radiative forcing from size parameterization, Atmos. Chem.
Phys., 13, 10733-10753, doi:10.5194/acp-13-10733-2013, 2013a.

Zhao, C., Leung, L. R., Easter, R., Hand, J., and Avise, J.: Characterization of speciated aerosol
direct radiative forcing over California, J. Geophys. Res., 118, 2372–2388, doi:
10.1029/2012JD018364, 2013b.

Zhao, C., Hu, Z., Qian, Y., Ruby Leung, L., Huang, J., Huang, M., Jin, J., Flanner, M. G., Zhang,
R., Wang, H., Yan, H., Lu, Z., and Streets, D. G.: Simulating black carbon and dust and their
radiative forcing in seasonal snow: a case study over North China with field campaign
measurements, Atmos. Chem. Phys., 14, 11475-11491, doi:10.5194/acp-14-11475-2014,
2014.

**List of Table**
Table 1. Experiment description

| Experiment ID | Experiment description |
|---|---|
| 20km | Simulation with the GOCART dust scheme at 20 km horizontal resolution. |
| 20km_D2 | Same as 20km, but with the DUSTRAN dust scheme. |
| 20km_P7 | Same as 20km_D2, but with the ACM2 PBL scheme. |
| 4km | Same as 20km, but at 4 km horizontal resolution. |
| 4km_D2 | Same as 4km, but with the DUSTRAN dust scheme. |


Table 2. Correlation with observations for different species at Fresno, CA

| Species | 20km | 4km | 4km_D2 | 20km_D2 | 20km_P7 |
|---|---|---|---|---|---|
| $PM_{2.5}$ | 0.89 | 0.90 | 0.86 | 0.78 | 0.03 |
| $PM_{2.5}\_NO_3$ | 0.94 | 0.95 | 0.94 | 0.94 | 0.91 |
| $PM_{2.5}\_NH_4$ | 0.97 | 0.96 | 0.96 | 0.98 | 0.96 |
| $PM_{2.5}\_OM$ | 0.93 | 0.93 | 0.94 | 0.93 | 0.91 |
| $PM_{2.5}\_EC$ | 0.98 | 0.98 | 0.98 | 0.98 | 0.96 |
| $PM_{2.5}\_SO_4$ | 0.63 | -0.16 | -0.14 | 0.61 | 0.63 |
| $PM_{2.5}\_dust$ | -0.55 | -0.50 | 0.48 | 0.55 | 0.36 |
| $PM_{10}$ | -0.25 | -0.23 | -0.08 | 0.01 | -0.03 |


Table 3. Surface aerosol mass ($\mu g\ m^{-3}$) for different species at Fresno, CA

| Species | Cold season | | | | | | Warm season | | | | | |
|---|---|---|---|---|---|---|---|---|---|---|---|---|
| | OBS | 20km | 4km | 4km_D2 | 20km_D2 | 20km_P7 | OBS | 20km | 4km | 4km_D2 | 20km_D2 | 20km_P7 |
| $PM_{2.5}$ | 16.84 | 13.71 | 21.38 | 22.48 | 14.90 | 13.77 | 8.44 | 4.91 | 6.29 | 12.85 | 10.12 | 14.85 |
| $PM_{2.5\_}$ $NO_3$ | 5.43 | 6.36 | 9.54 | 9.22 | 6.22 | 3.16 | 0.84 | 0.55 | 0.69 | 0.79 | 0.66 | 0.57 |
| $PM_{2.5\_}$ $NH_4$ | 1.42 | 1.97 | 2.99 | 2.88 | 1.91 | 0.98 | 0.40 | 0.19 | 0.24 | 0.20 | 0.16 | 0.13 |
| $PM_{2.5\_}$ OM | 5.39 | 0.92 | 2.07 | 2.07 | 0.93 | 1.04 | 2.47 | 0.49 | 0.87 | 0.87 | 0.50 | 0.55 |
| $PM_{2.5\_}$ EC | 1.08 | 0.52 | 1.12 | 1.13 | 0.52 | 0.58 | 0.32 | 0.27 | 0.49 | 0.49 | 0.27 | 0.30 |
| $PM_{2.5\_}$ $SO_4$ | 0.87 | 0.53 | 0.82 | 0.81 | 0.53 | 0.46 | 1.04 | 0.54 | 0.61 | 0.60 | 0.53 | 0.49 |
| $PM_{2.5\_}$ dust | 0.90 | 0.11 | 0.11 | 1.65 | 1.50 | 4.18 | 2.08 | 0.04 | 0.03 | 6.49 | 5.16 | 10.05 |
| $PM_{10}$ | 31.55 | 14.93 | 22.81 | 28.32 | 20.10 | 24.52 | 34.82 | 7.08 | 8.69 | 38.12 | 30.19 | 48.02 |


Supplementary Table 1. Correlation with surface observations for meteorological variables at
Fresno, CA

|  | 4km_D2 | 20km_D2 | 20km_P7 |
|---|---|---|---|
| T | 0.94 | 0.94 | 0.94 |
| RH | 0.98 | 0.98 | 0.96 |
| Wind | 0.83 | 0.84 | 0.85 |
| Rain | 0.97 | 0.97 | 0.97 |


Supplementary Table 2. Bias for surface meteorological variables at Fresno, CA

|  | Cold season | | | Warm season | | |
|---|---|---|---|---|---|---|
|  | 4km_D2 | 20km_D2 | 20km_P7 | 4km_D2 | 20km_D2 | 20km_P7 |
| T (K) | 3.89 | 3.56 | 3.69 | 2.44 | 1.50 | 1.35 |
| RH (%) | -9.78 | -14.55 | -19.35 | -9.48 | -9.32 | -11.16 |
| Wind (m/s) | -0.67 | -1.00 | -1.05 | 0.78 | -0.16 | -0.49 |
| Rain (mm/day) | -0.15 | 0.14 | -0.03 | -0.06 | -0.03 | -0.04 |


**List of Figures**

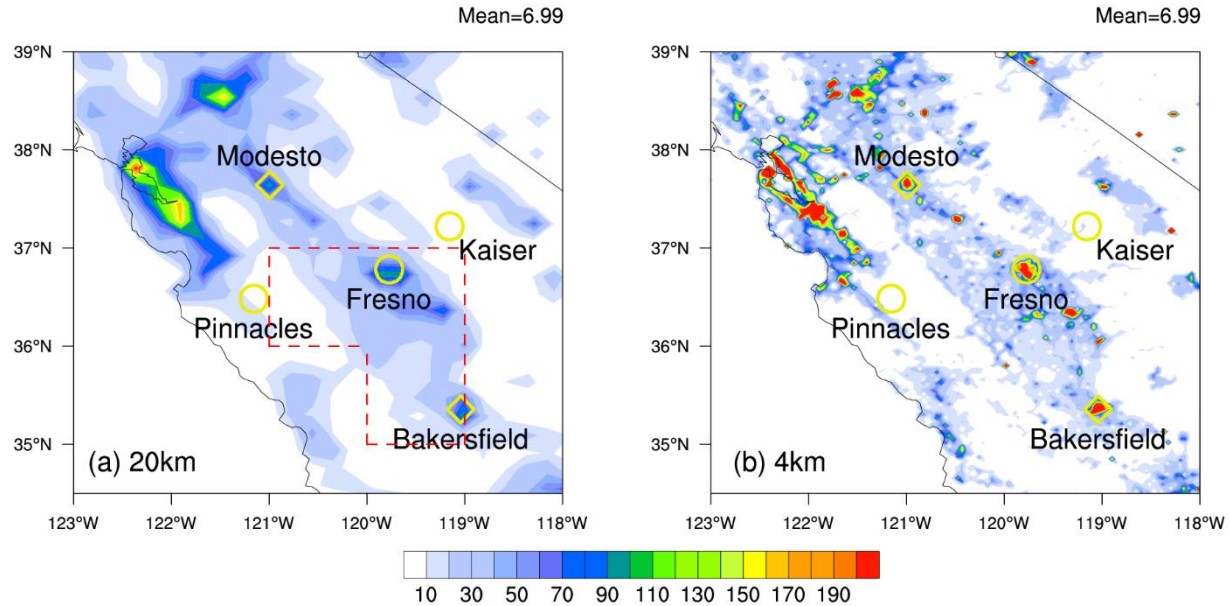

Figure 1. Daily mean anthropogenic $PM_{2.5}$ emission rate (µg m$^{-2}$ hr$^{-1}$) at (a) 20km and (b) 4km
simulation. Domain-averaged emission rate is shown at right corner of each figure. Red dashed
lines in Figure 1a represent the region used for the domain averages in the discussions. Yellow
circle: IMPROVE site; yellow diamond: EPA CSN site. Three urban sites: Fresno, Bakersfield and
Modesto; two rural sites: Pinnacles and Kaiser.

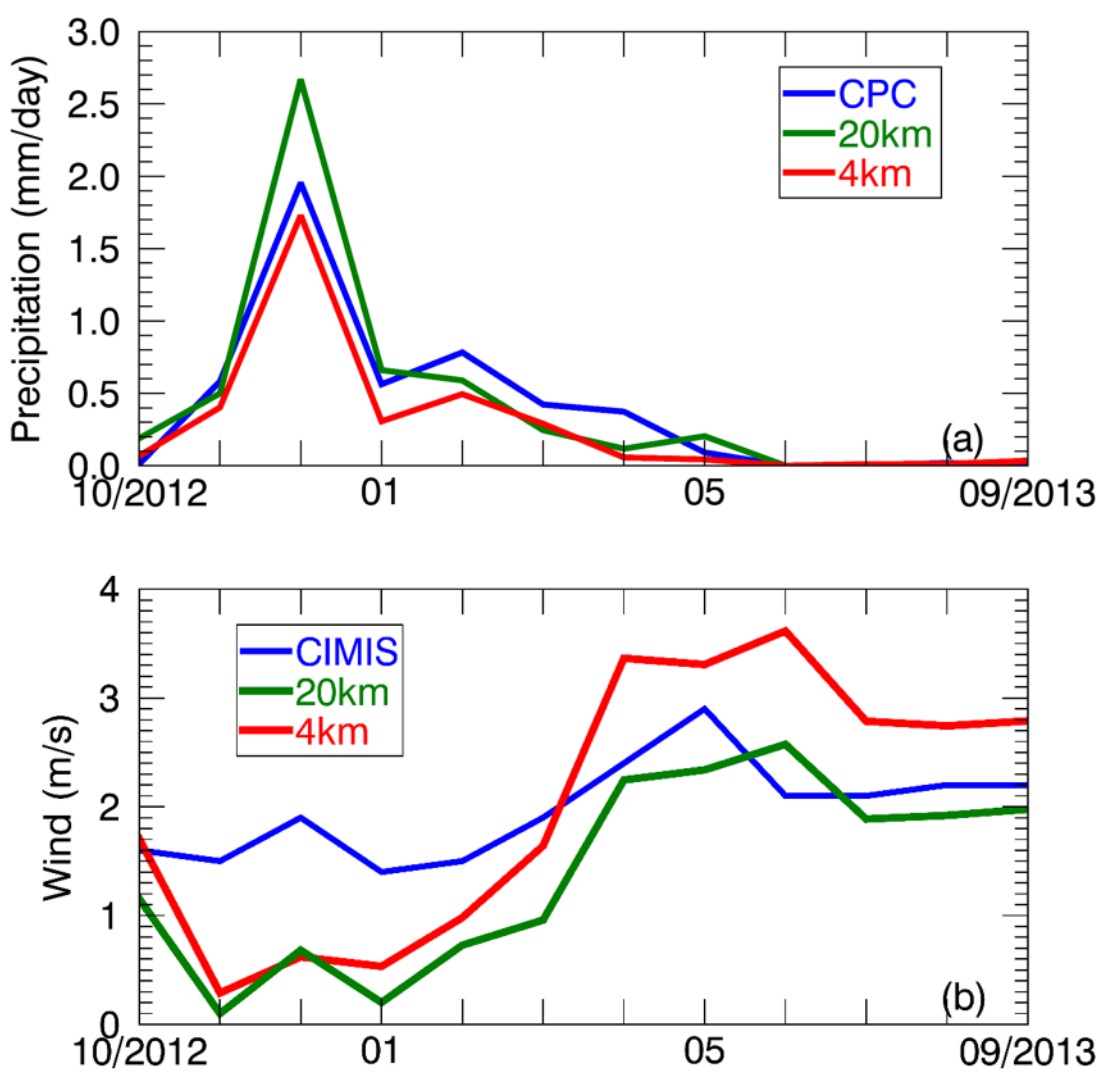


Figure 2. (a) Monthly precipitation (mm/day) from CPC, 20km and 4km; (b) monthly wind speed
(m/s) from CIMIS, 20km and 4km. 4km_D2 (not shown) is similar to 4km.

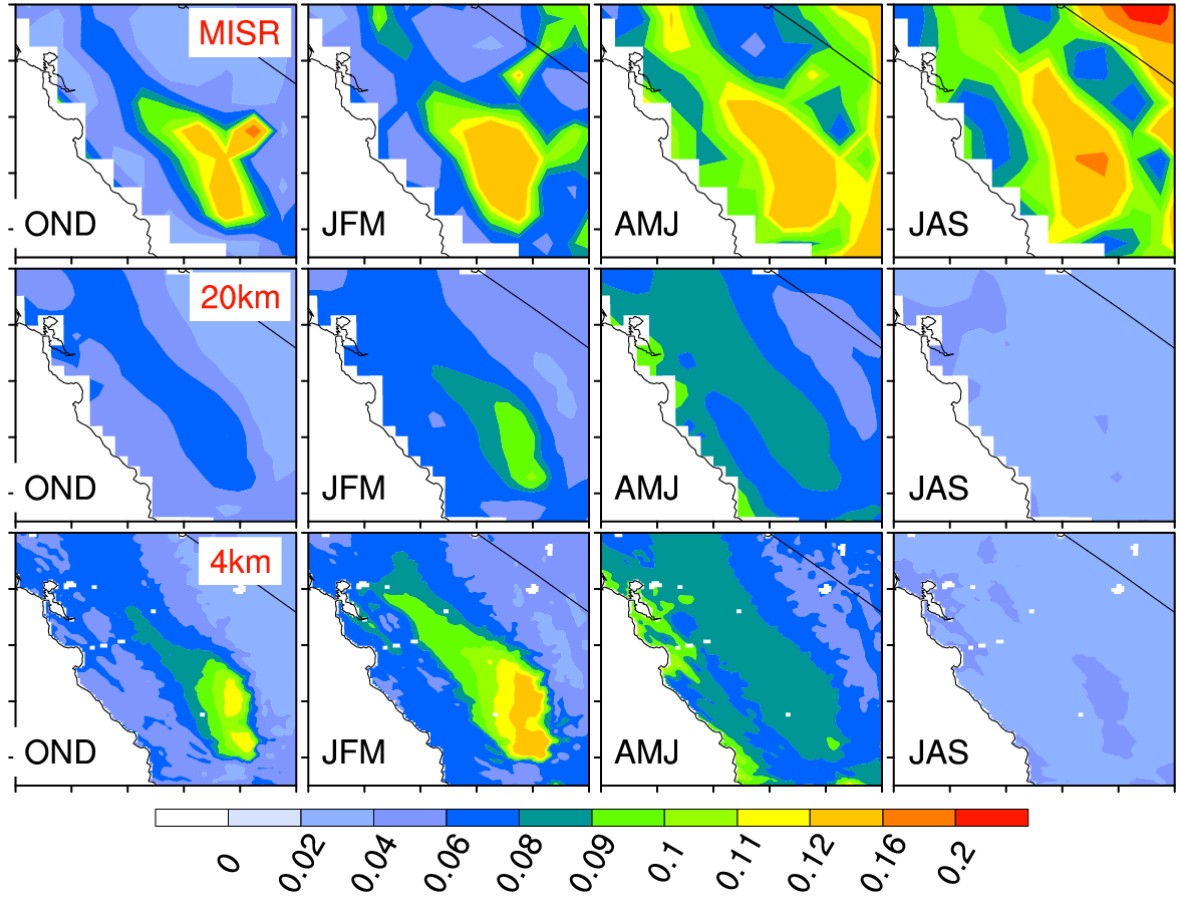


Figure 3. Spatial distribution of seasonal mean 550 nm AOD from MISR and the WRF-Chem

(20km and 4km) simulations in WY2013. OND: October-November-December; JFM: January-

February-March; AMJ: April-May-June; JAS: July-August-September.


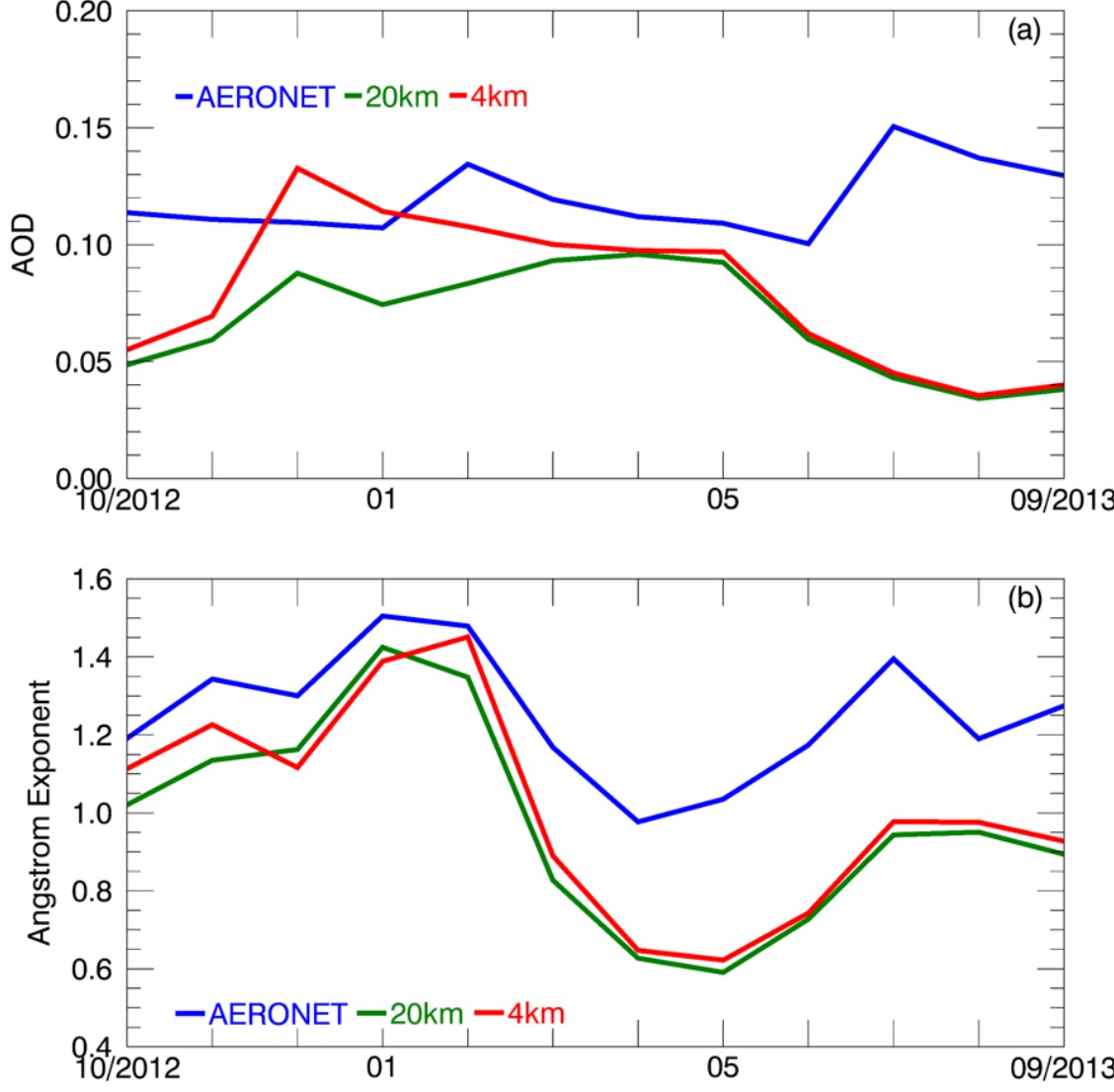


Figure 4. (a) Monthly mean 550 nm AOD; (b) monthly mean 400-600 nm Ångström exponent at
Fresno, CA from October 2012 to September 2013.

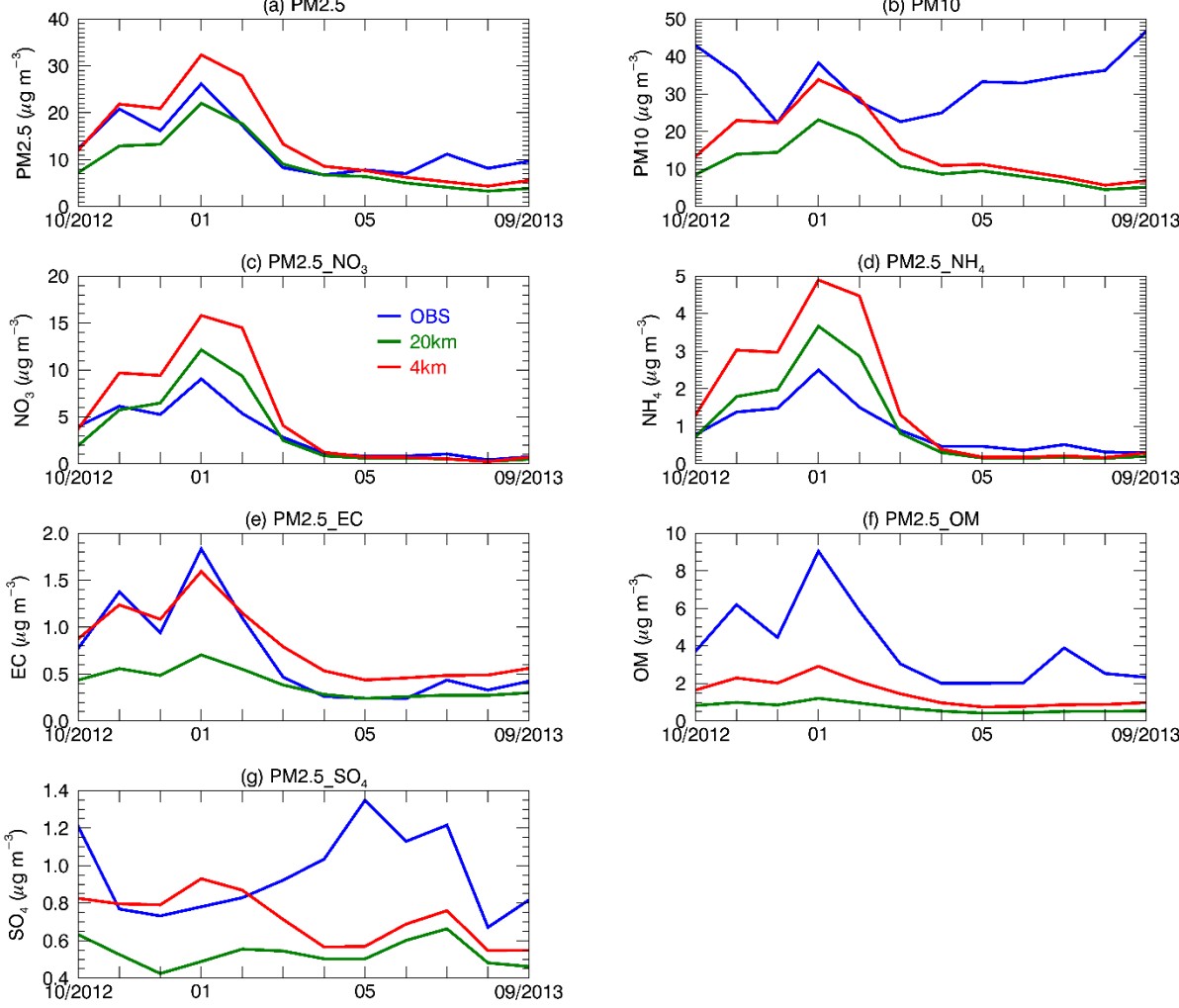


Figure 5. Aerosol mass (μg m⁻³) for different species from OBS, the 20km and 4km simulations at
Fresno, CA. NH$_4$ observations are from EPA; other observations are from IMPROVE. PM$_{2.5}$_NO$_3$
represents NO$_3$ with diameter ≤ 2.5 μm. Similar definition for NH$_4$, EC, OM and SO$_4$ in the figures.

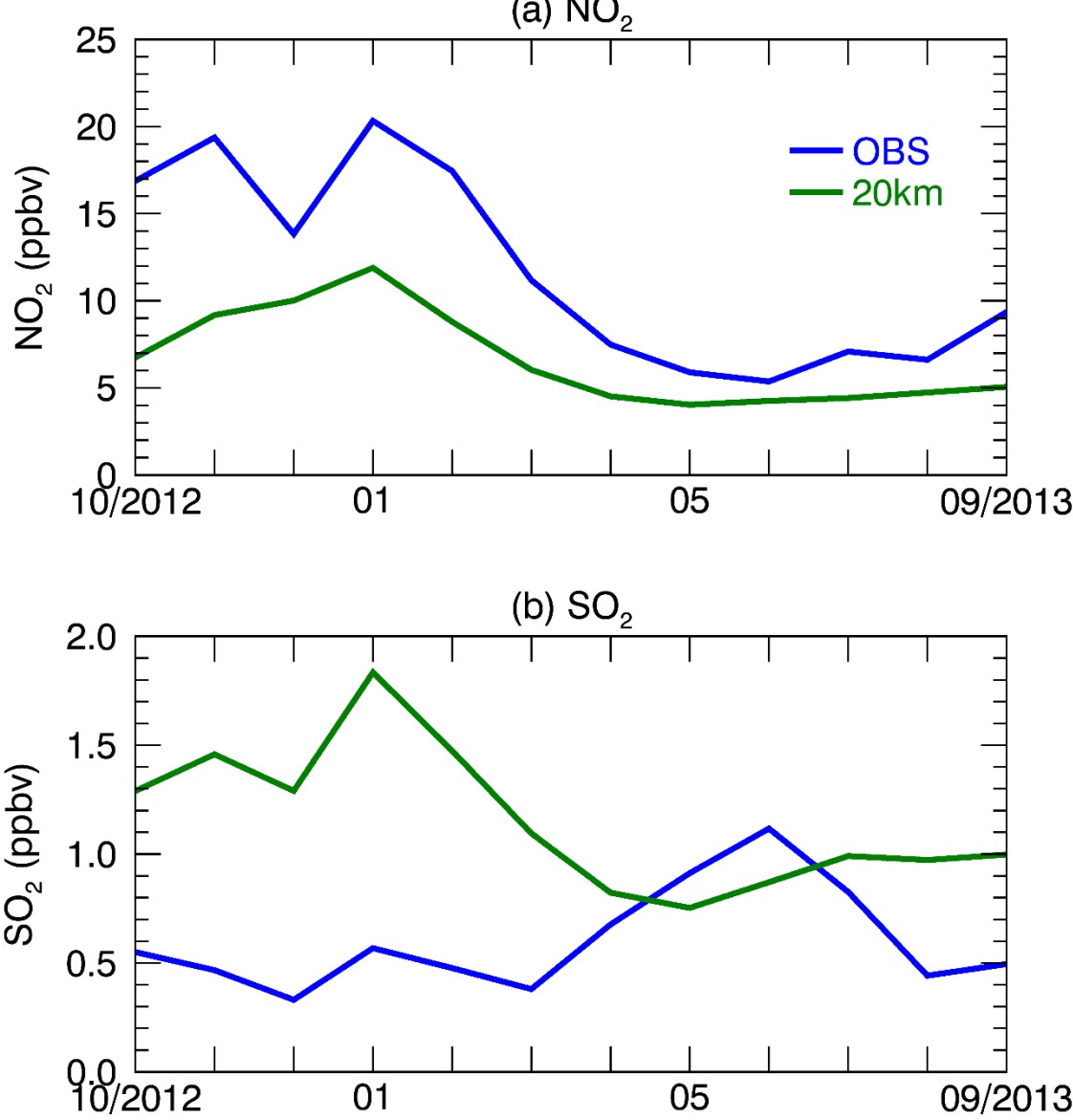


Figure 6. (a) $NO_2$ and (b) $SO_2$ from EPA (OBS) and the 20km run at Fresno, CA.

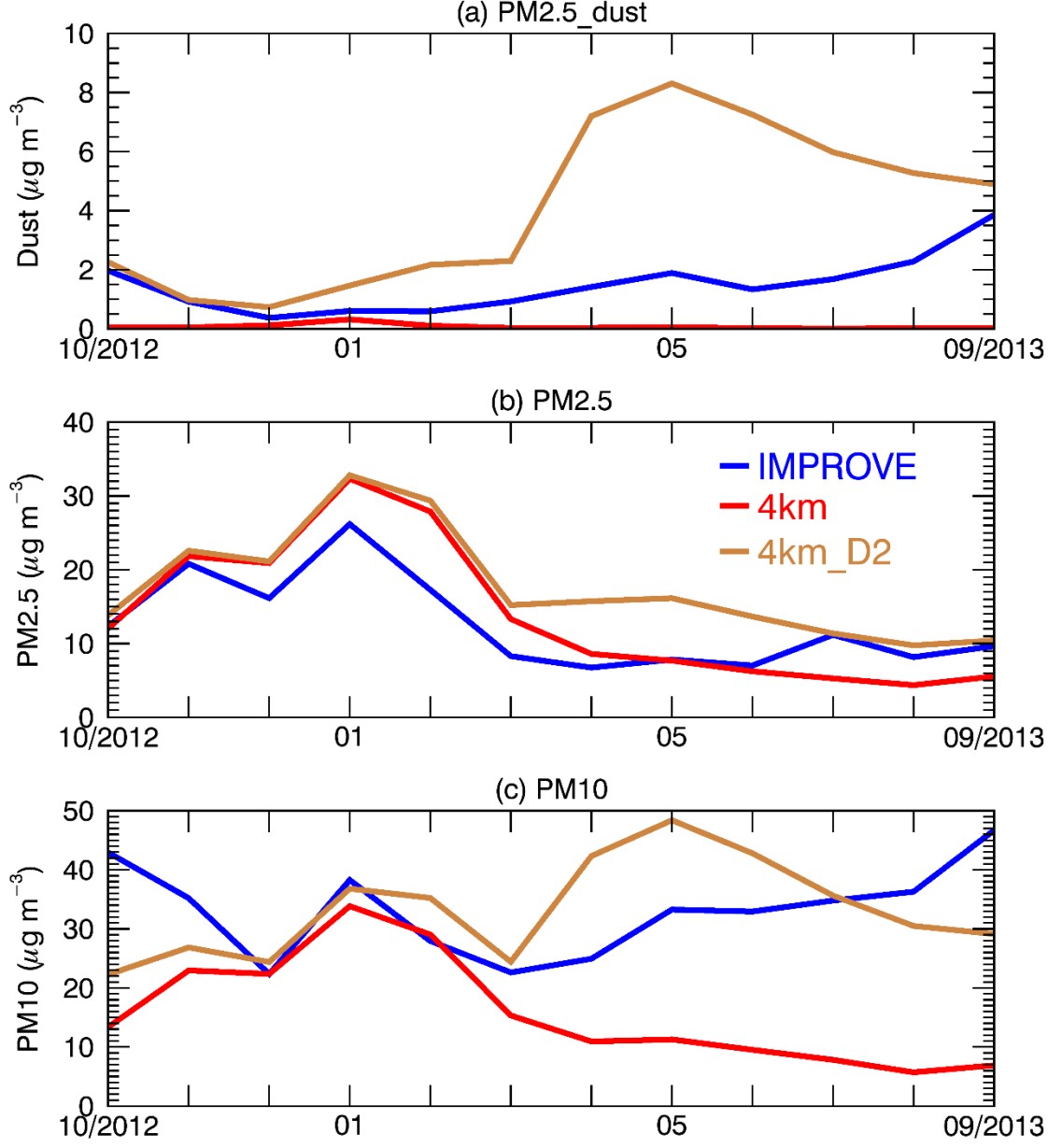


Figure 7. (a) PM$_{2.5}$_dust; (b) PM$_{2.5}$; and (c) PM$_{10}$ from IMPROVE, the 4km and 4km_D2
simulations at Fresno, CA.

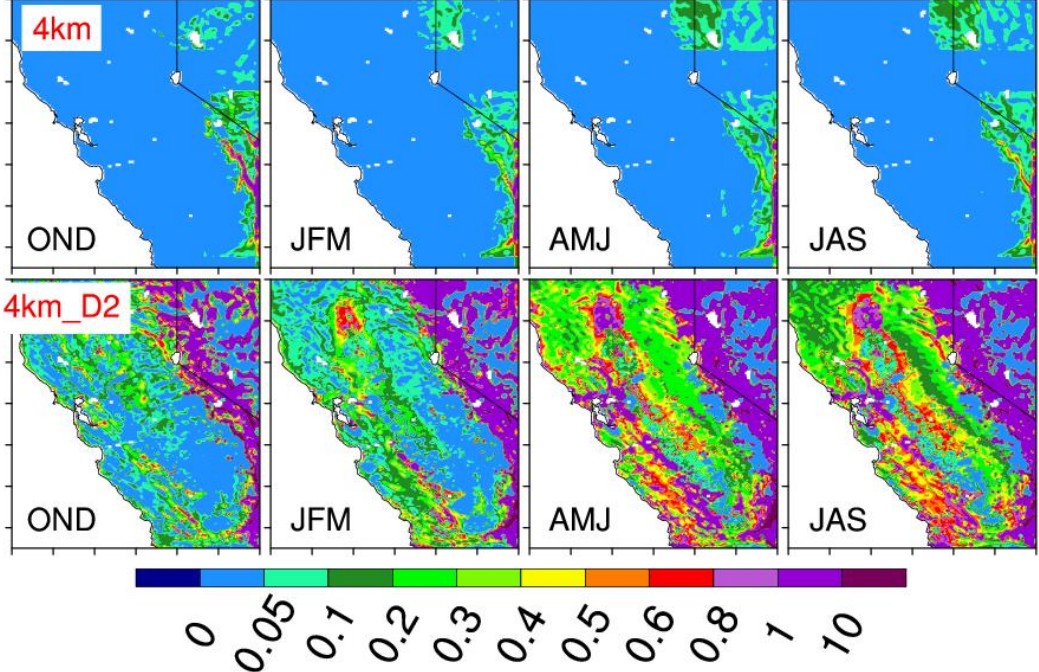


Figure 8. Mean dust emission rate (μg m$^{-2}$ s$^{-1}$) from the 4km and 4km_D2 runs.

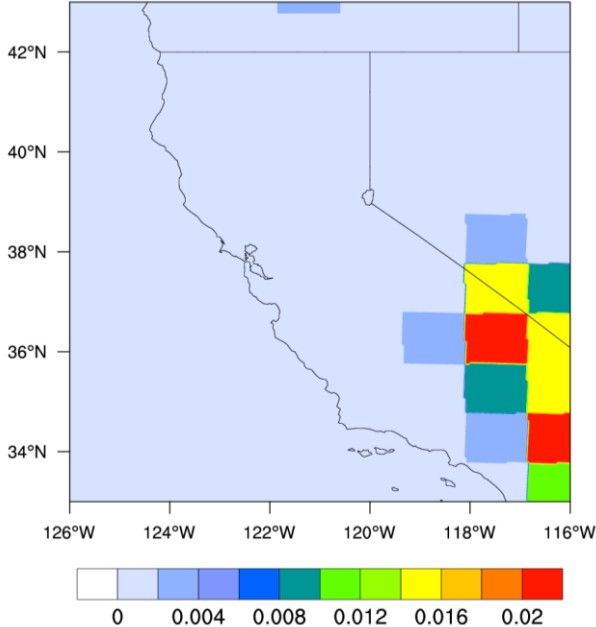


Figure 9. Fraction of erodible surface in the GOCART dataset used in this study.

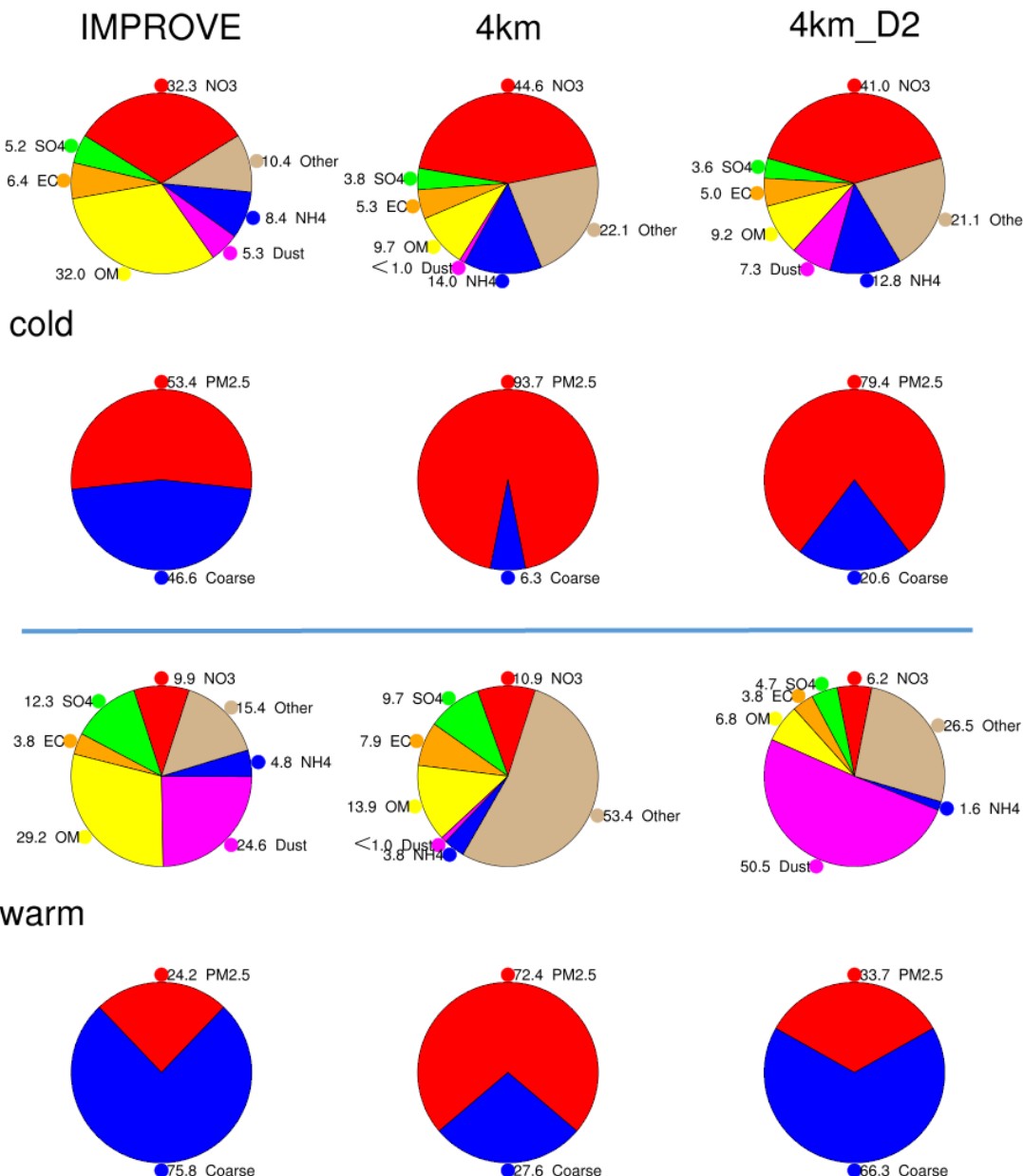

Figure 10. Relative contribution (%) of aerosol species from IMPROVE and the WRF-Chem (4km and 4km_D2) simulations at Fresno, CA in WY2013. (Panel 1) Contribution to $PM_{2.5}$ in the cold season; (Panel 2) relative contribution of $PM_{2.5}$ and coarse mass (CM) to $PM_{10}$ in the cold season; (Panel 3) same as Panel 1 but in the warm season; (Panel 4) same as Panel 2 but in the warm season. "Other" refers to the difference of $PM_{2.5}$ total mass and specified $PM_{2.5}$ ($NO_3$, $NH_4$, OM, EC, $SO_4$ and dust).

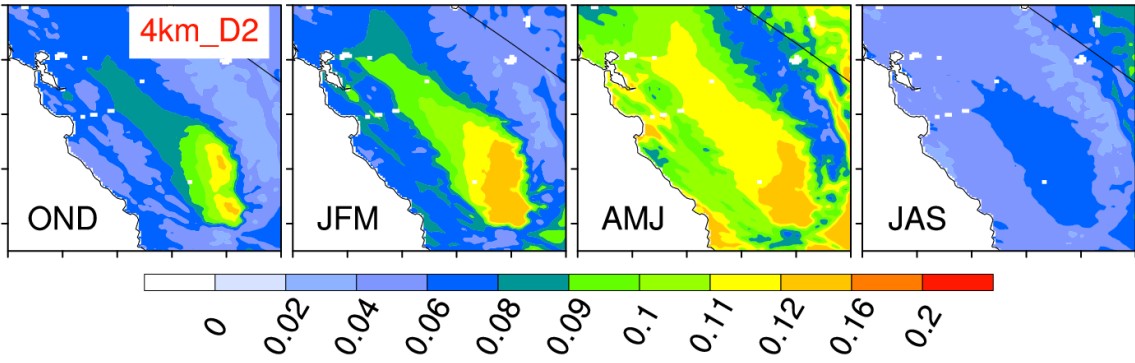

Figure 11. Spatial distribution of seasonal mean 550 nm AOD from the 4km_D2 run in WY2013.

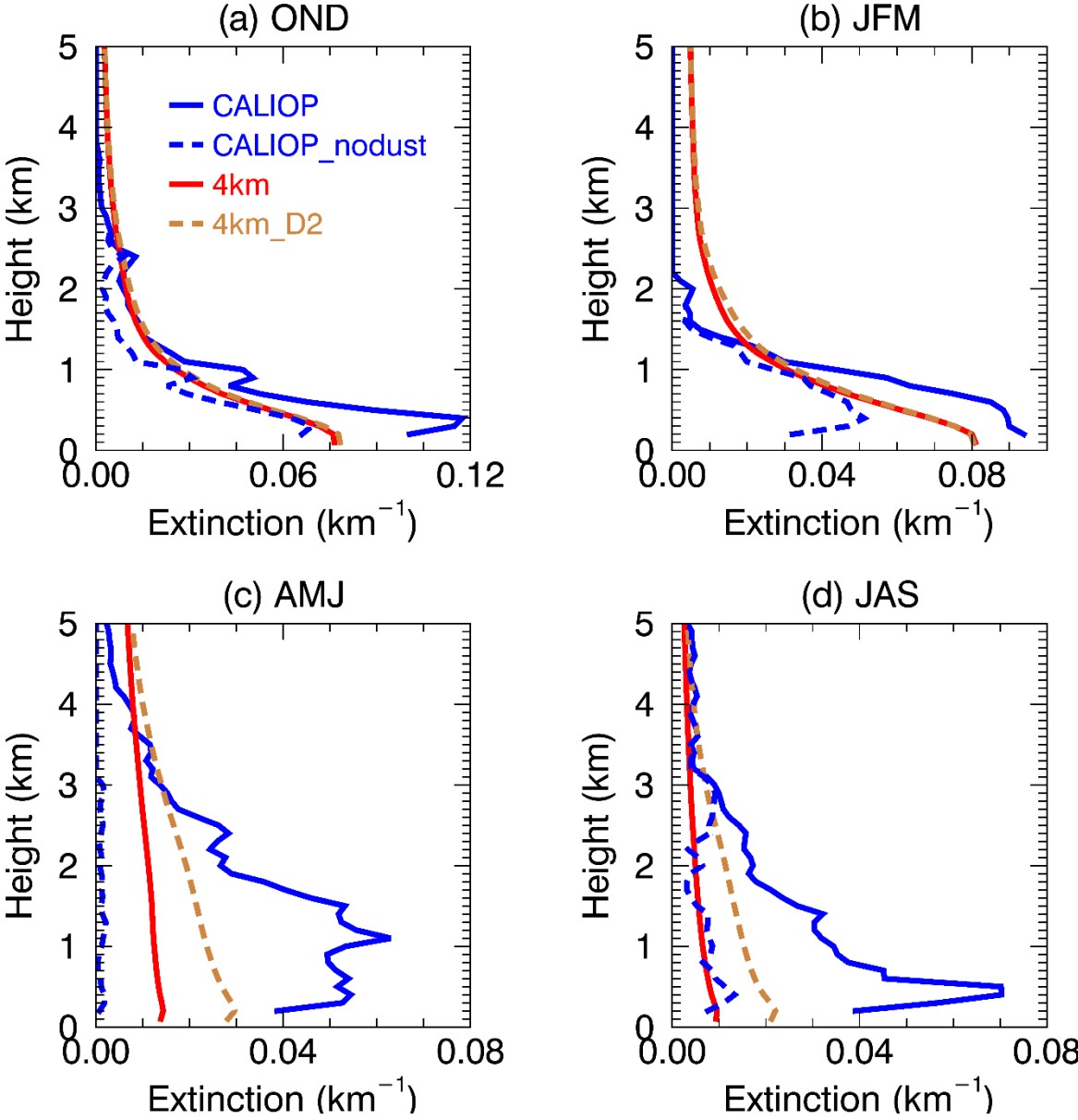


Figure 12. Vertical distribution of seasonal mean 532 nm aerosol extinction coefficient (km⁻¹)
from CALIOP (blue) and the WRF-Chem (4km and 4km_D2) simulations over the red box
region in Fig. 1a in WY2013. Blue dashed lines (CALIOP_nodust) represent the CALIOP
profiles without dust (dust and polluted dust).

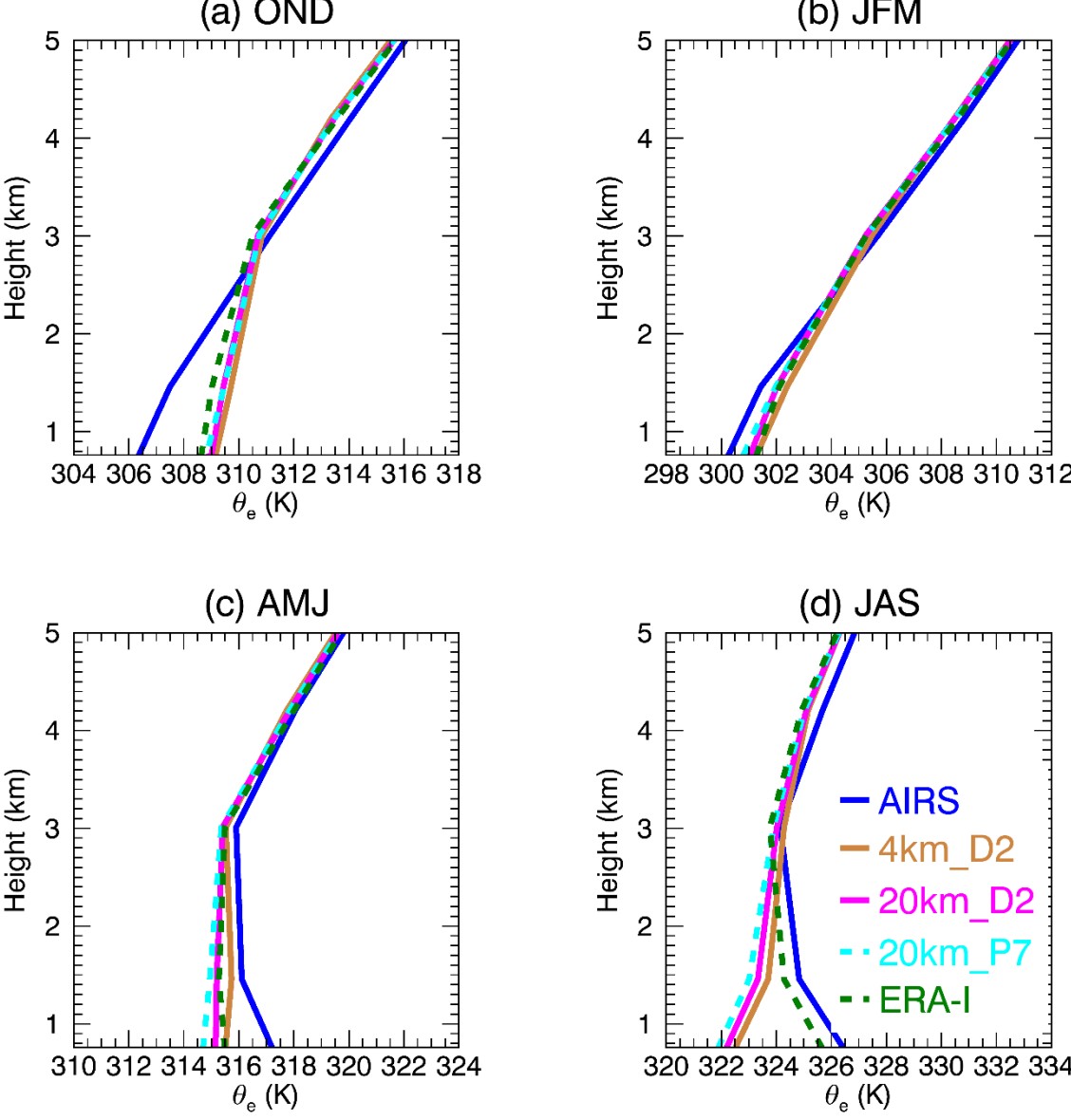


Figure 13. Vertical distribution of season mean equivalent potential temperature ($\theta_e$; K) from AIRS,
ERA-Interim (ERA-I) and the WRF-Chem (4km_D2, 20km_D2 and 20km_P7) simulations over
the red box region in Fig. 1a in WY2013. The 4km run (not shown) is similar to the 4km_D2 run.

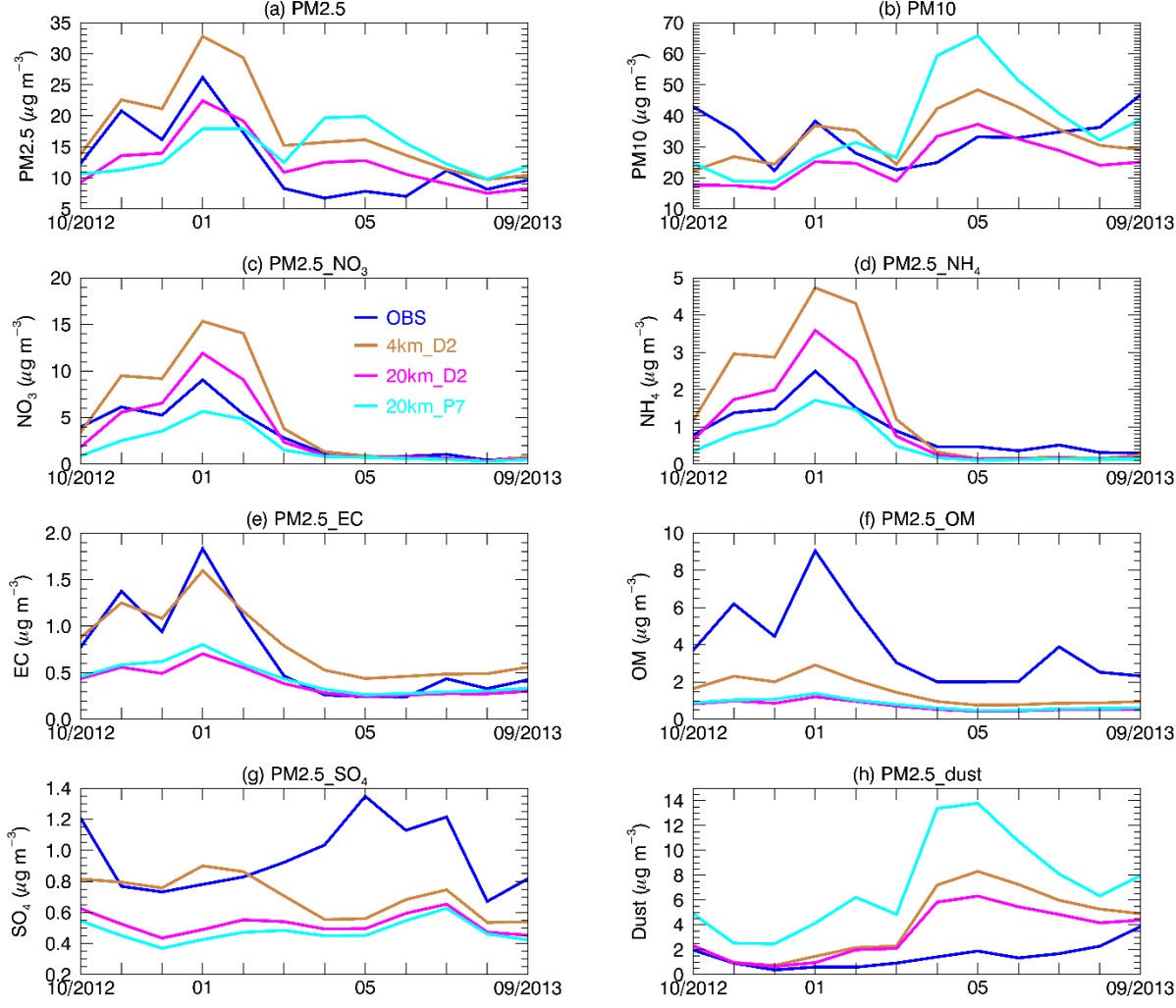

Figure 14. Aerosol mass (μg m⁻³) for different species from OBS, the 4km_D2, 20km_D2 and 20km_P7 simulations at Fresno, CA. $NH_4$ observations are from EPA; other observations are from IMPROVE. $PM_{2.5}$_$NO_3$ represents $NO_3$ with diameter ≤ 2.5 μm. Similar definition for $NH_4$, EC, OM, $SO_4$ and dust in the figures.

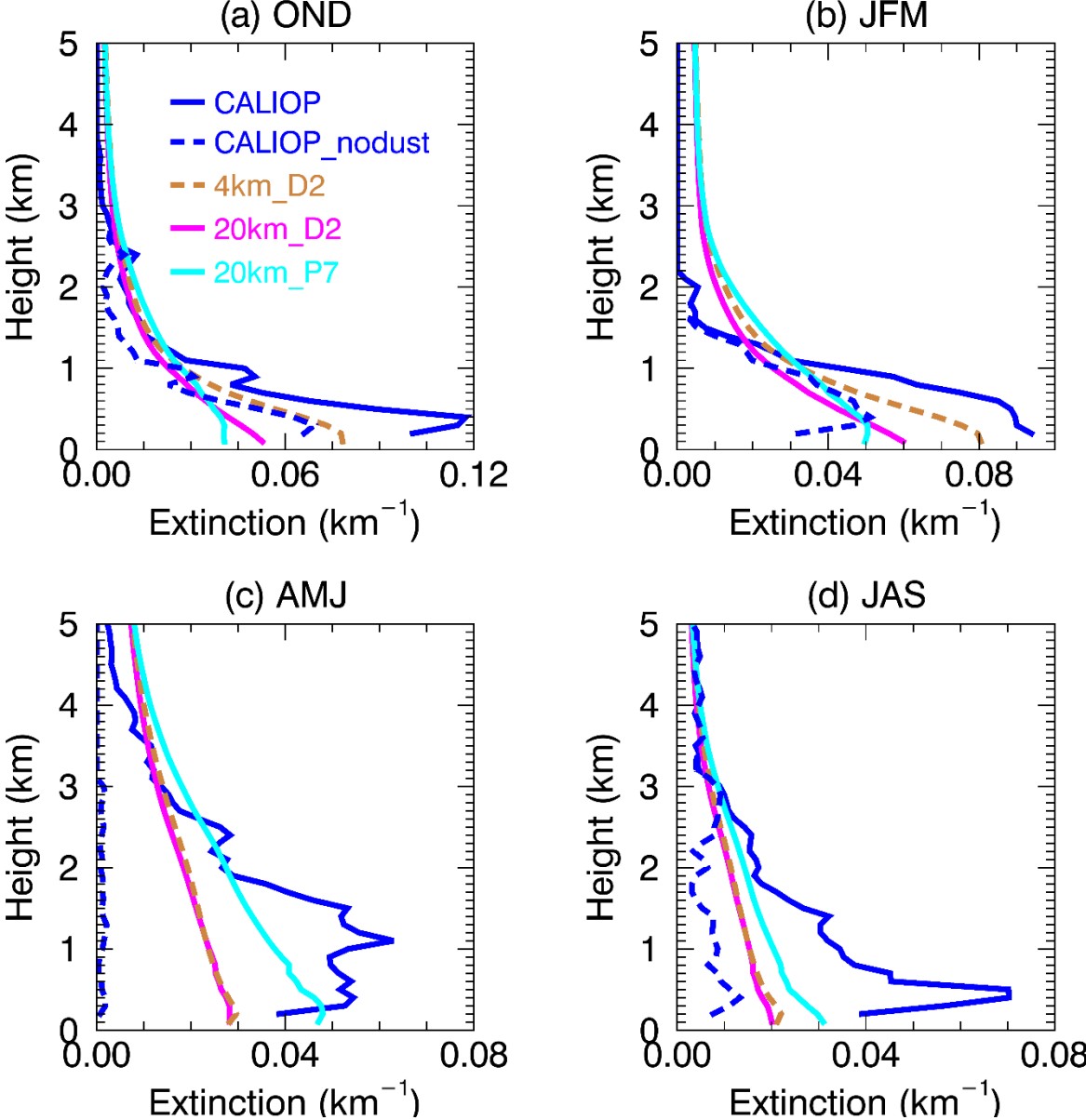

Figure 15. Vertical distribution of seasonal mean 532 nm aerosol extinction coefficient (km-1)

from CALIOP, CALIOP_nodust, and the WRF-Chem (4km_D2, 20km_D2 and 20km_P7)

simulations over the red box region in Fig. 1a in WY2013.

930

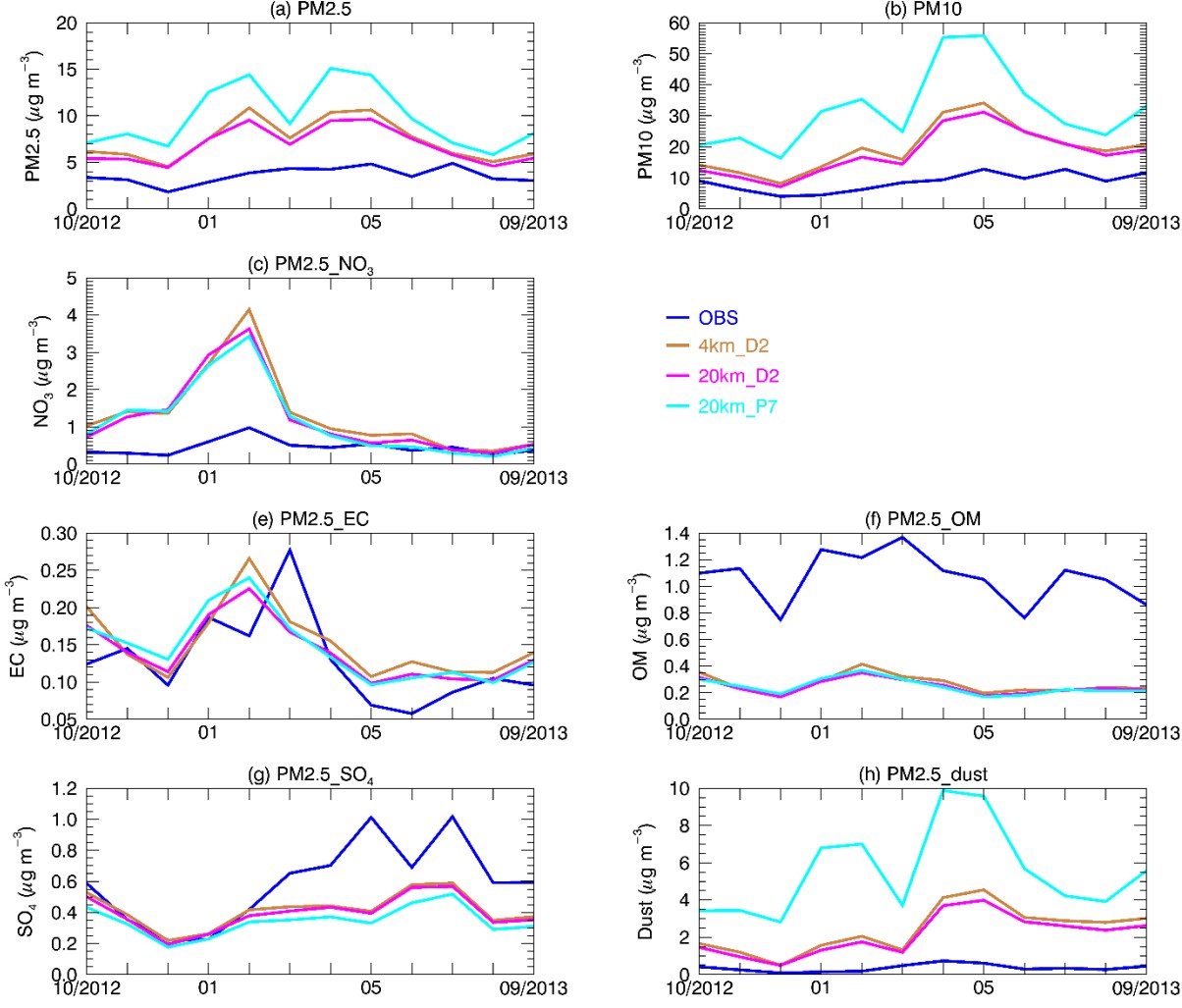

934

Figure 16. Aerosol mass (μg m⁻³) for different species from IMPROVE (OBS), the 4km_D2,

20km_D2 and 20km_P7 simulations at Pinnacles, CA.

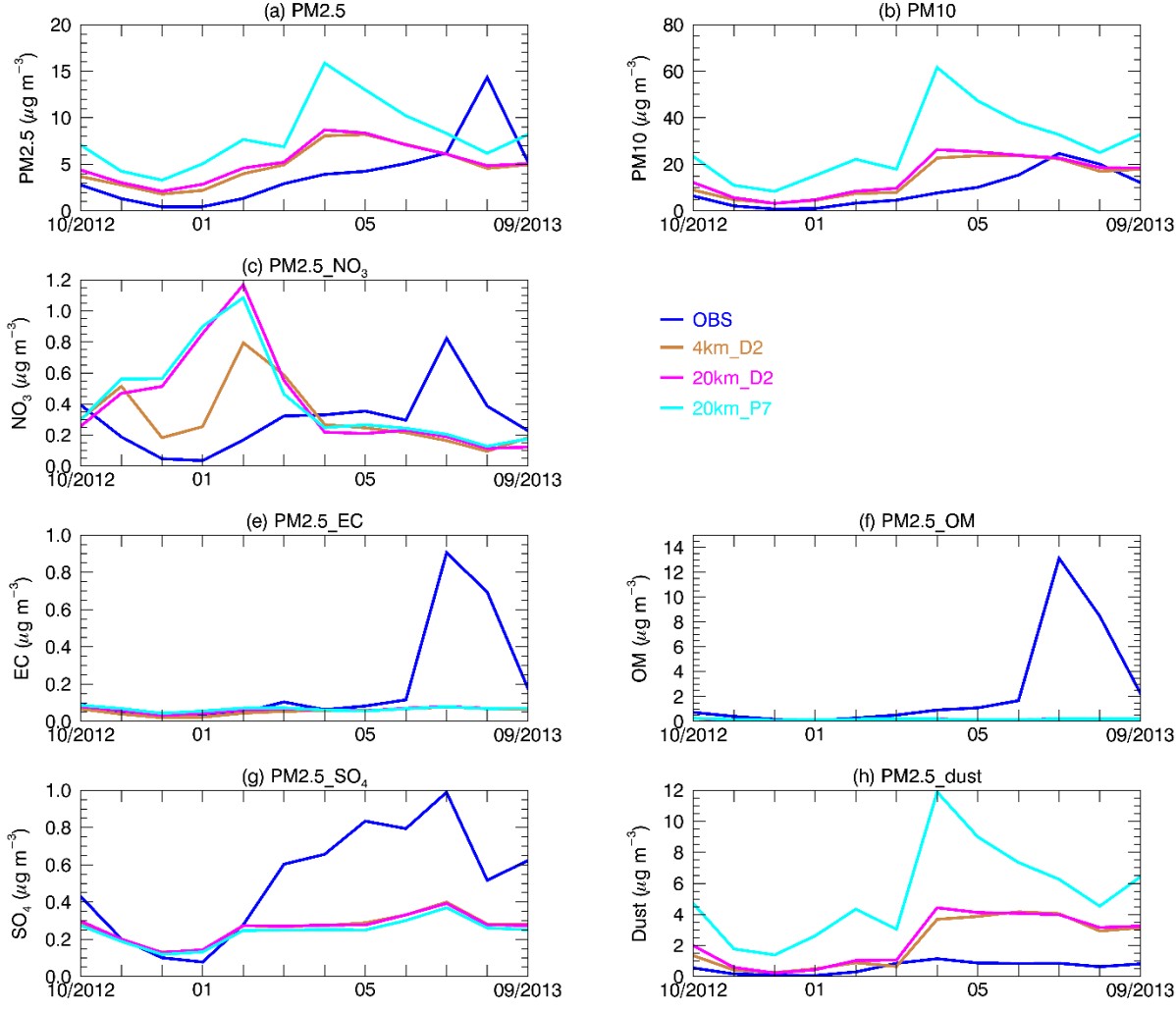

Figure 17. Aerosol mass (μg m⁻³) for different species from IMPROVE (OBS), the 4km_D2, 20km_D2 and 20km_P7 simulations at Kaiser, CA.

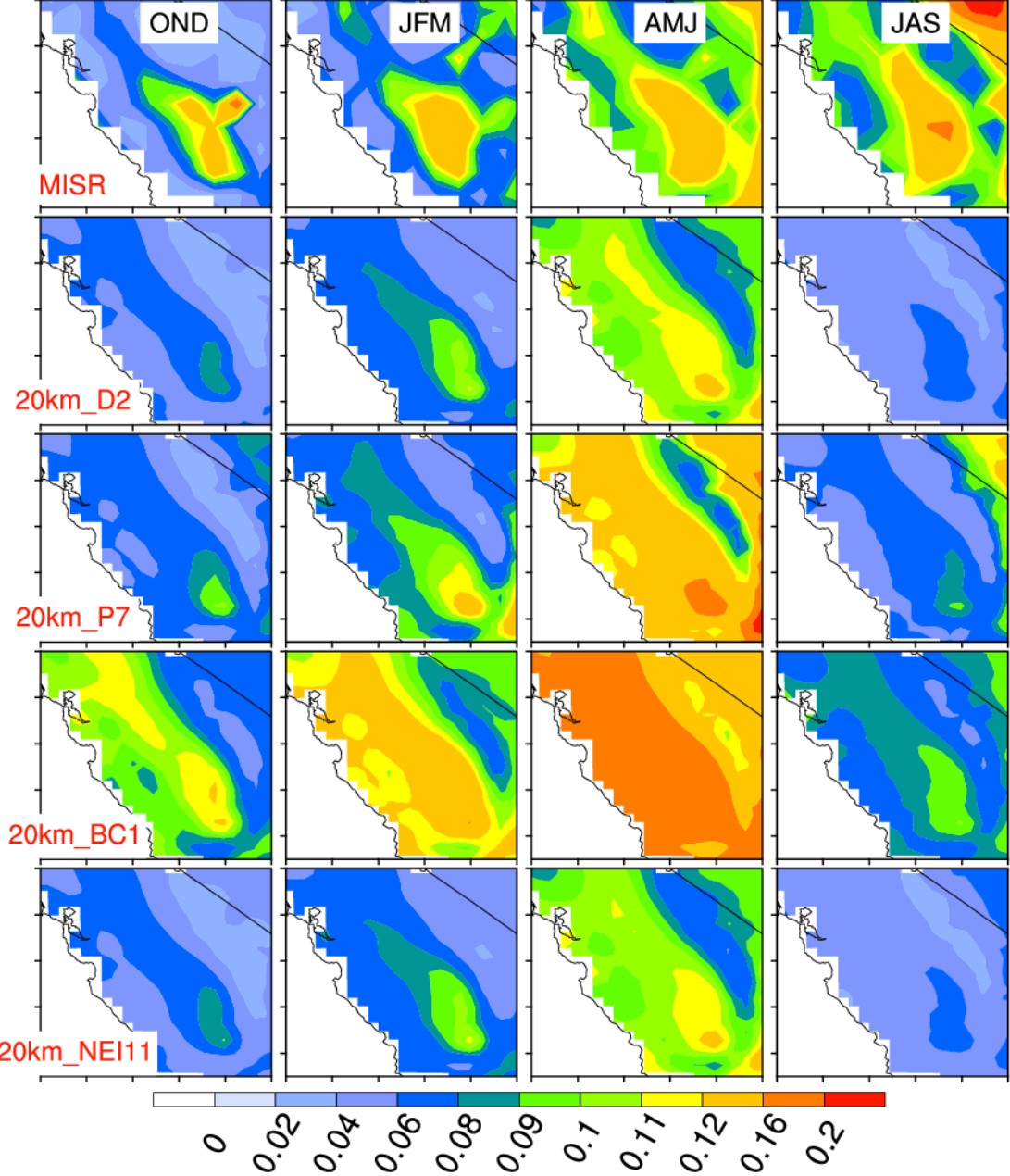

Supplementary Figure 1. Spatial distribution of seasonal mean 550 nm AOD from MISR and the WRF-Chem (20km_D2, 20km_P7, 20km_BC1 and 20km_NEI11) simulations in WY2013. OND: October-November-December; JFM: January-February-March; AMJ: April-May-June; JAS: July-August-September. The 20km_BC1 run is the same as the 20km_D2 run except that chemical boundary conditions use MOZART-4 original data. The 20km_NEI11 run is the same as the 20km_D2 run except with NEI11 anthropogenic emissions.

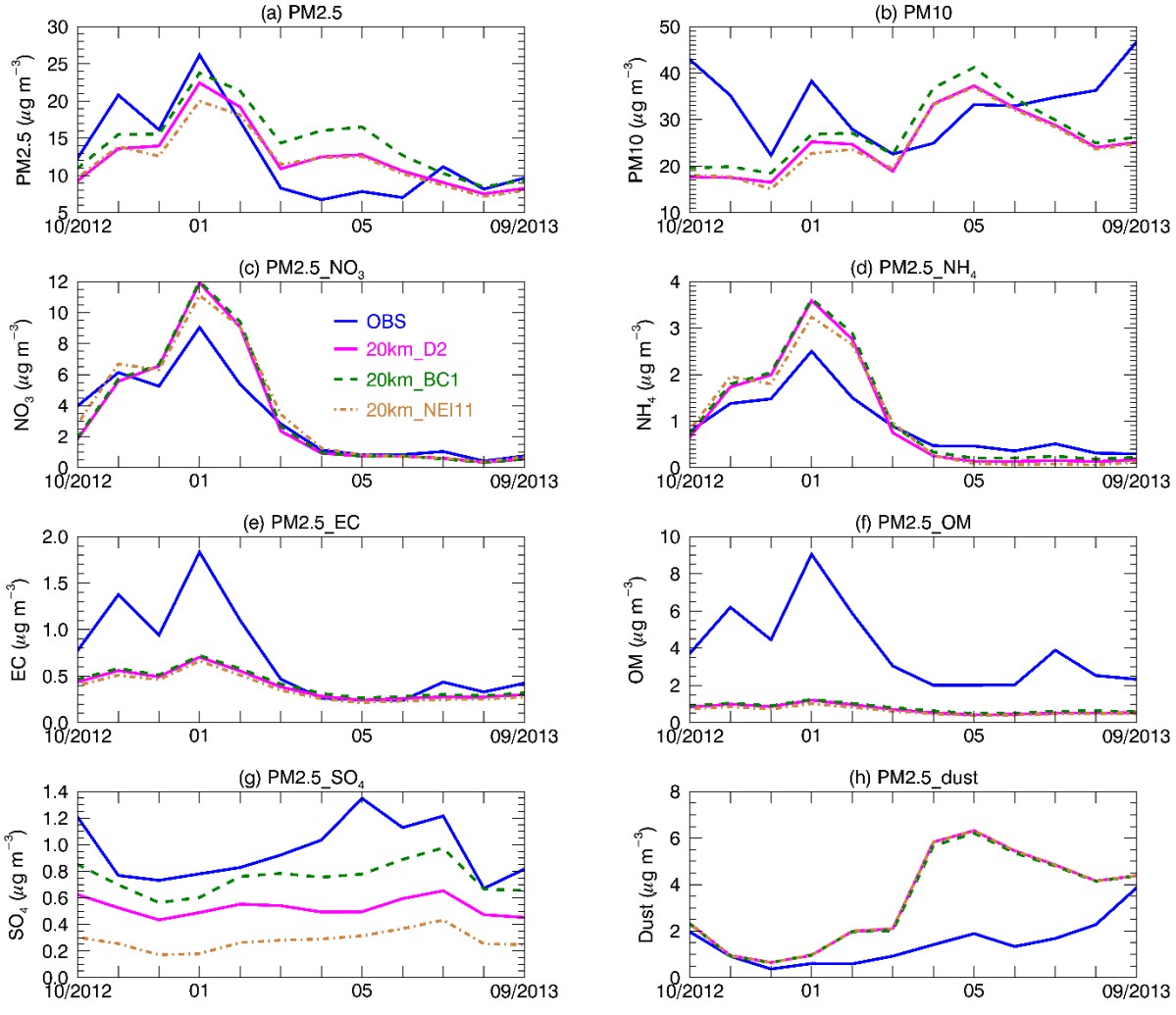


Supplementary Figure 2. Aerosol mass ($\mu g\ m^{-3}$) for different species from OBS, the 20km_D2,
20km_BC1 and 20km_NEI11 simulations at Fresno, CA. $NH_4$ observations are from EPA; other
observations are from IMPROVE. $PM_{2.5}$_$NO_3$ represents $NO_3$ with diameter $\leq 2.5$ μm. Similar
definition for $NH_4$, EC, OM, $SO_4$ and dust in the figures.

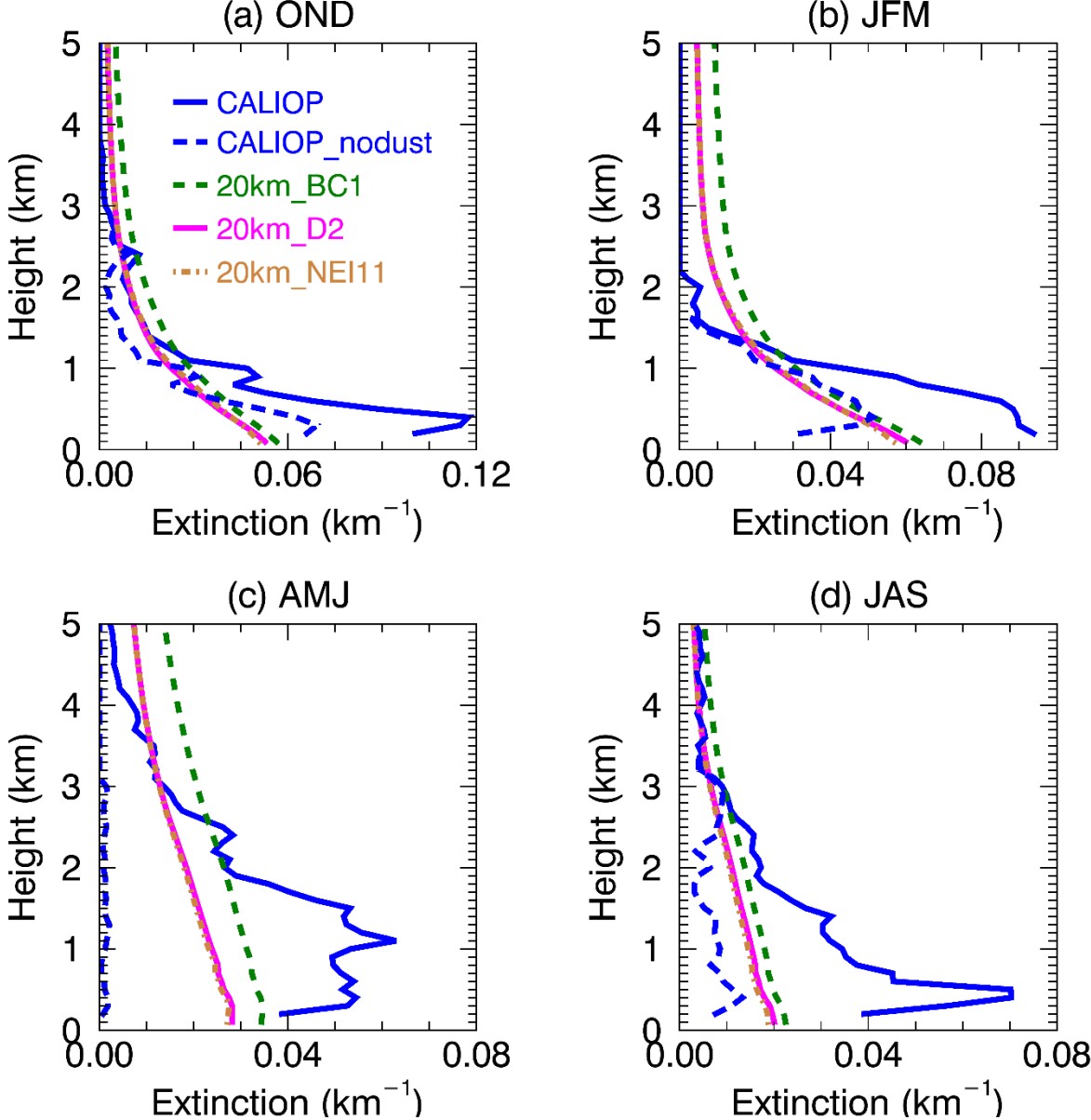


Supplementary Figure 3. Vertical distribution of seasonal mean 532 nm aerosol extinction
coefficient (km⁻¹) from CALIOP, CALIOP_nodust, and the WRF-Chem (20km_D2, 20km_BC1
and 20km_NEI11) simulations over the red box region in Fig. 1a in WY2013.

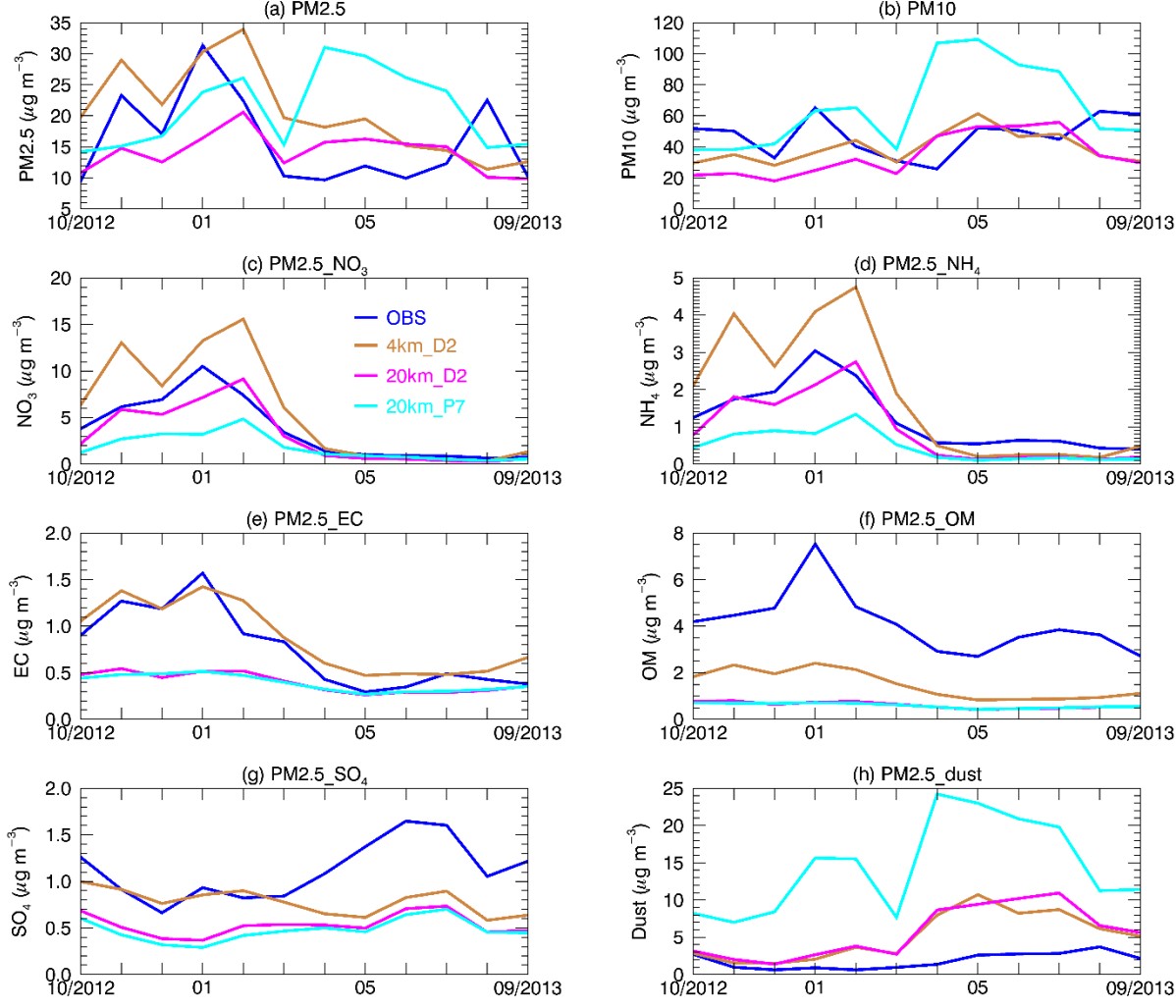


Supplementary Figure 4. Aerosol mass (μg m⁻³) for different species from EPA CSN (OBS), the
4km_D2, 20km_D2 and 20km_P7 simulations at Bakersfield, CA. $PM_{2.5}$_$NO_3$ represents $NO_3$
with diameter ≤ 2.5 μm. Similar definition for $SO_4$, EC, OM, $NH_4$ and dust in the figures.

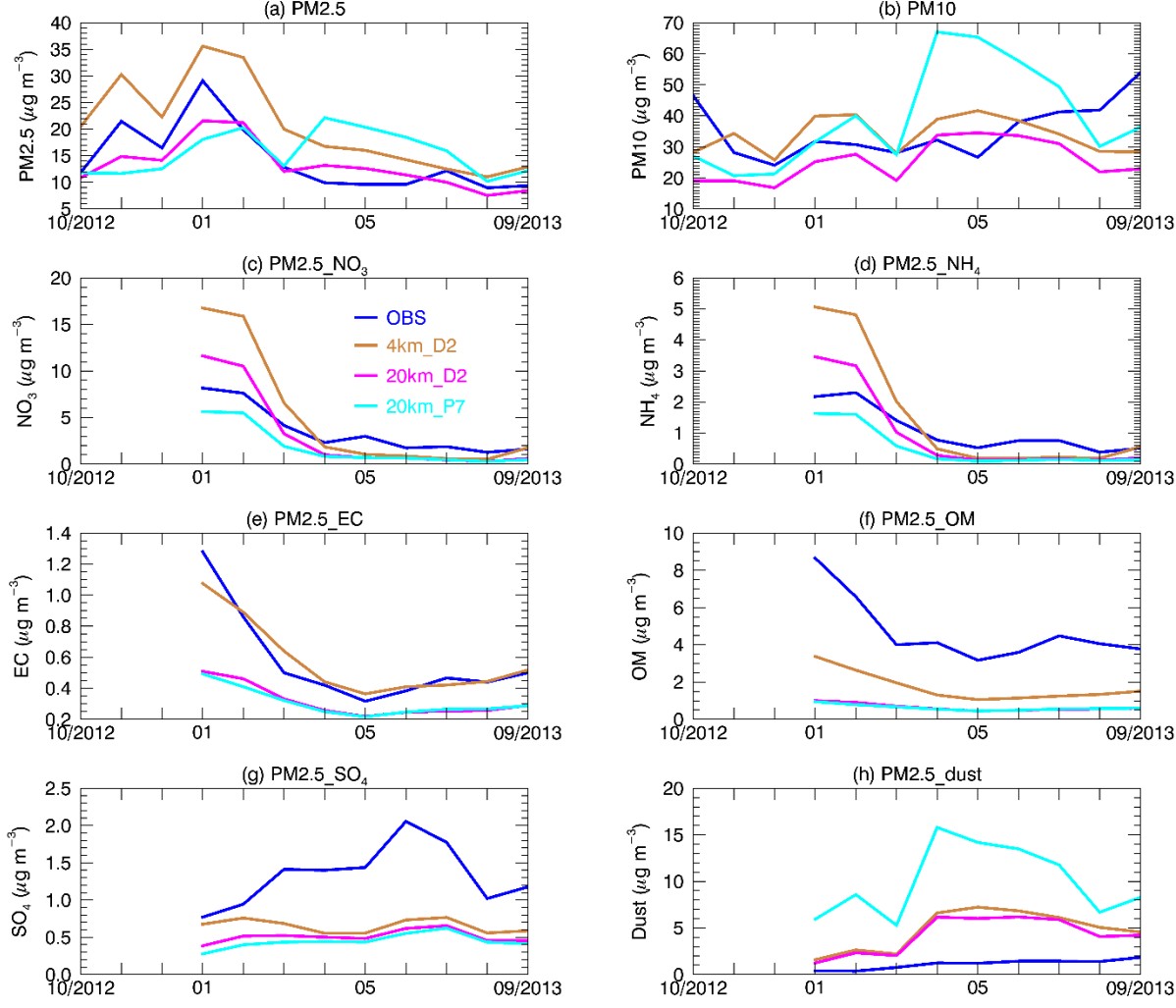


Supplementary Figure 5. Aerosol mass (μg m⁻³) for different species from EPA CSN (OBS), the
4km_D2, 20km_D2 and 20km_P7 simulations at Modesto, CA.

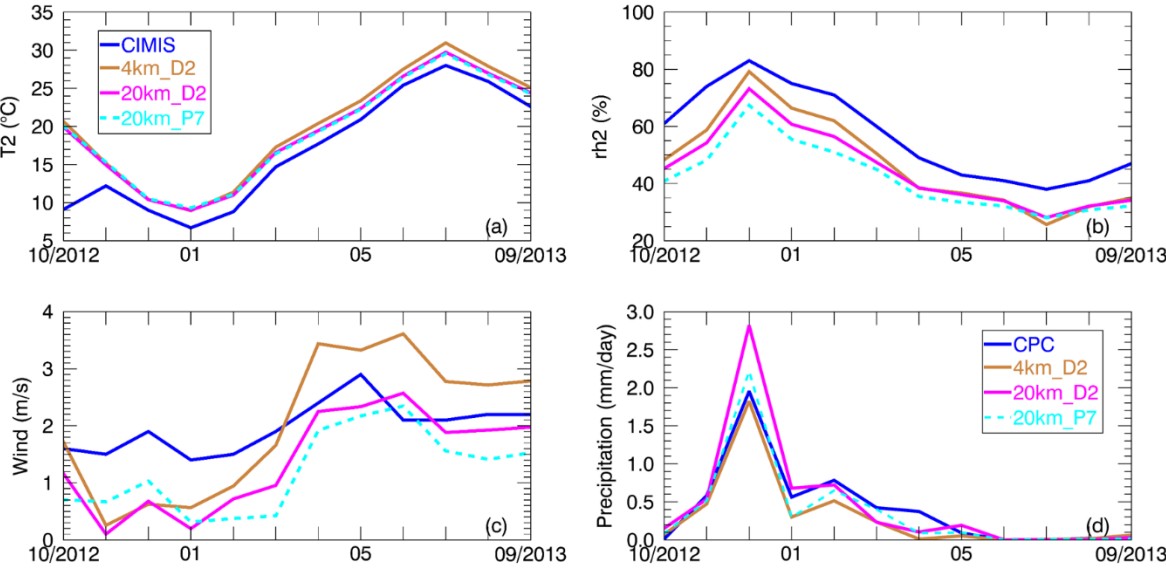


Supplementary Figure 6. Monthly mean of (a) 2-m temperature (°C); (b) 2-m relative humidity
(%); (c) 10-m wind speed (m/s); (d) precipitation (mm/day) at Fresno, CA. The 20km (not shown)
run is similar to the 20km_D2 run while the 4km (not shown) run is similar to the 4km_D2 run.

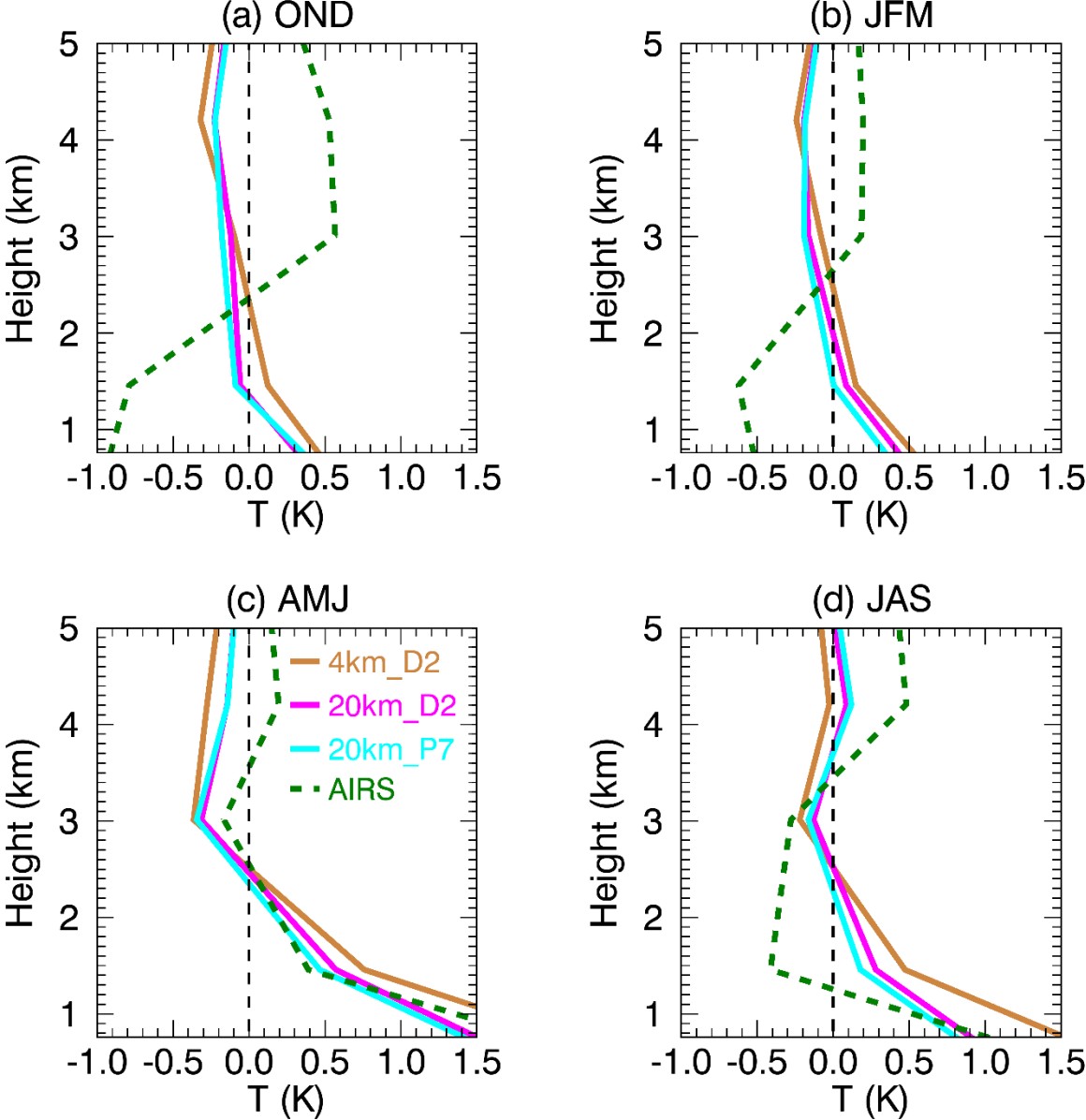


Supplementary Figure 7. Vertical profile of seasonal mean temperature (K) bias in the WRF-Chem
simulations and AIRS comparing to ERA-Interim. The 20km run (not shown) is similar to the
20km_D2 run while the 4km run (not shown) is similar to the 4km_D2 run.

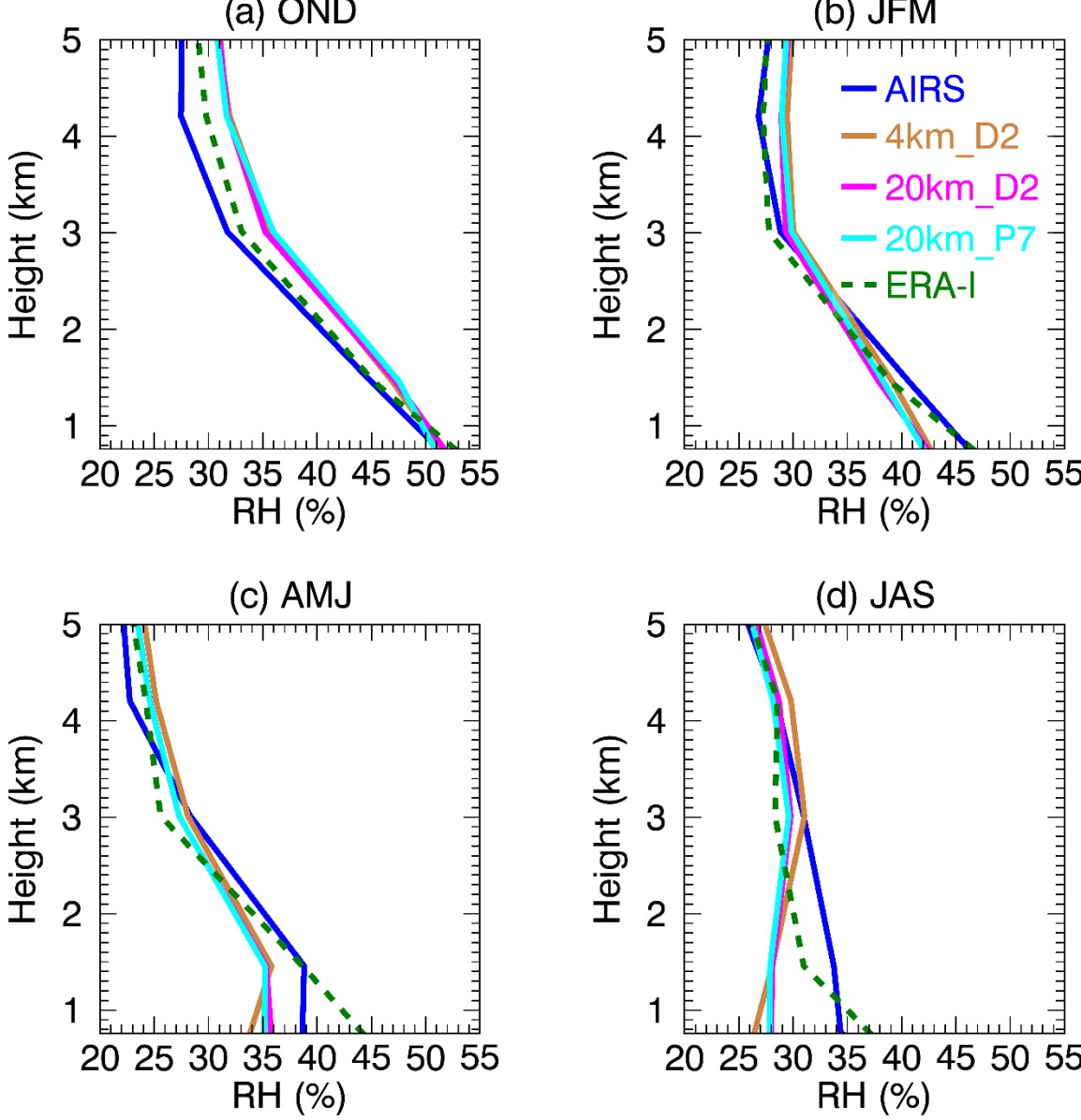


Supplementary Figure 8. Vertical profile of seasonal mean relative humidity (%) in the WRF-Chem
simulations, AIRS and ERA-Interim. The 20km run (not shown) is similar to the 20km_D2 run
while the 4km run (not shown) is similar to the 4km_D2 run.

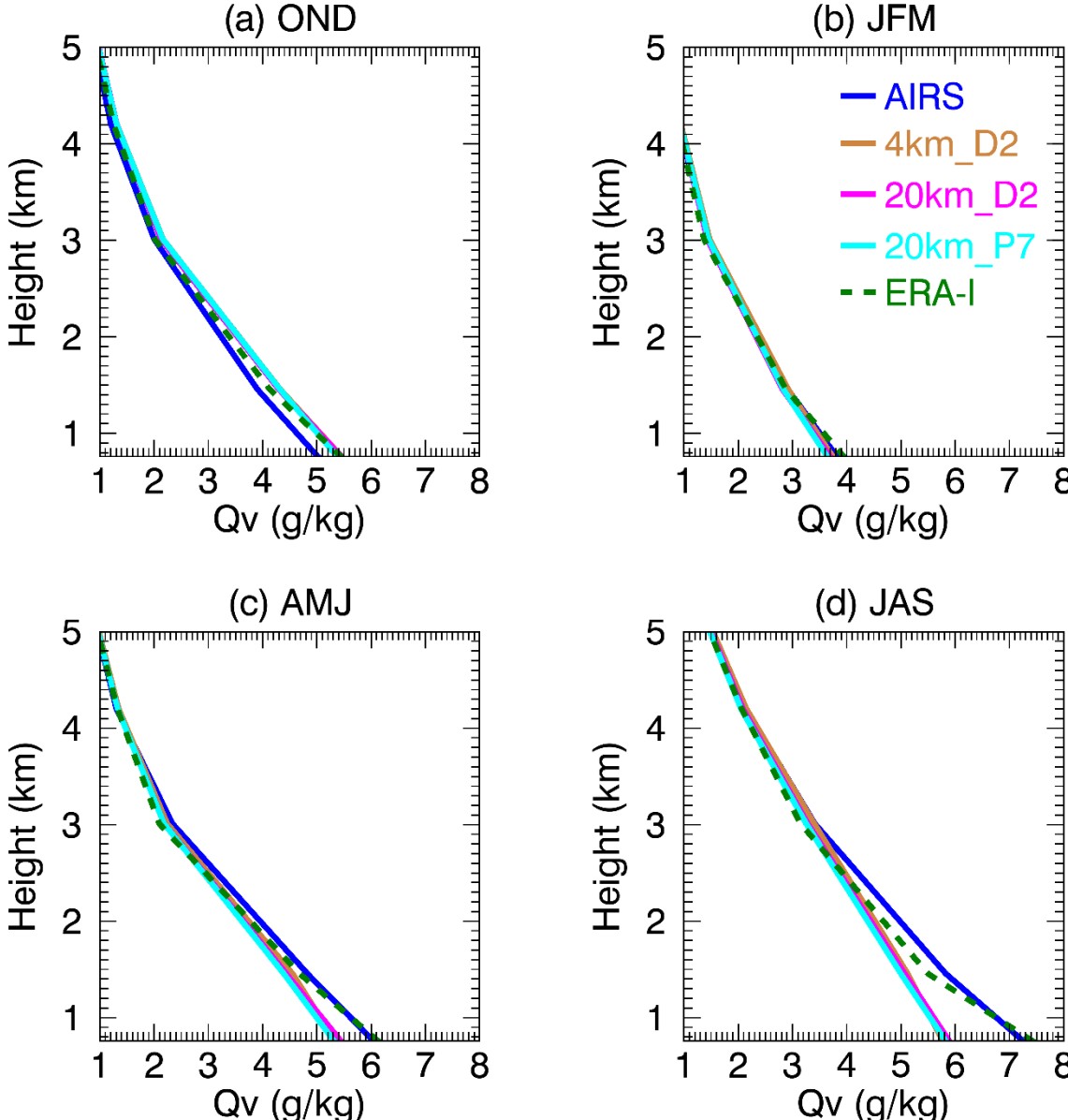


Supplementary Figure 9. Vertical profile of seasonal mean specific humidity (g kg$^{-1}$) in the WRF-Chem simulations, AIRS and ERA-Interim. The 20km run (not shown) is similar to the 20km_D2 run while the 4km run (not shown) is similar to the 4km_D2 run.