# Peer review of "WRF-Chem simulation of aerosol seasonal variability in the San Joaquin Valley"

_Atmospheric Chemistry and Physics, 2016_

## Referee Comment (RC1) · Anonymous Referee #1 · 26 Dec 2016

This paper examines the performance of a regional-scale chemical transport model in representing aerosol properties in the San Joaquin Valley over a one year period. The model is compared with surface measurements of composition and AOD as well as satellite measurements. The motivation for the paper is sufficient (although could be improved), but the main weakness is their approach and interpretation of the simulations. In addition, the paper is poorly written.

Major Comments:

The most important problem the manuscript has is how the model was configured to address the purpose of the study. WRF-Chem is a useful tool, but as with all models can only perform well when it is configured properly. The following is a discussion of items the authors should consider to revise and/or address.

[Figure]

Domain and Dust Emissions: It is clear that the model domain is larger than the one shown in Figure 1. But it is hard for me to assess the importance of dust emissions since those are not shown. For local sources, dust is likely generated in the desert areas to the southwest of the SJV. It would be useful to show the emission regions from GOCART and DUSTRAN. My understanding is that the emission regions in DUSTRAN as implemented in WRF-Chem are rather ad hoc. They may depend on vegetation type. I suspect that dust is being generated locally in the SJV in DUSTRAN but not in GOCART. The authors mention how many grid nodes are used in the vertical direction, but should give an idea of the vertical resolution near the surface that will affect dust emissions. Dust emissions will depend in part on wind speed, and representing wind speed in California depends a lot on circulations affected by terrain. Both a fine horizontal and vertical resolution is needed to represent those winds that will affect dust emissions. It is not clear how well the model performed in winds – particularly over the dust emission regions. While some evaluation of the thermodynamic structure is given, there is nothing for the winds.

Boundary Conditions: The authors half the amount of aerosols from MOZART following Fast et al. (2014). But the errors in a coarse global model, like MOZART, will likely change in time and depend on meteorological conditions. There is no sensitivity results or evidence whether such a change in boundary conditions is warranted in the present study. I believe the version of MOZART the authors use prescribes dust using climatology which would affect the simulations over California. The potential errors in MOZART that will contribute to AOD over California will likely vary over a year-long period.

Simulation Period: On line 167, the authors state that the simulation period is from 2012 to 2013. There is no rationale as to why this period is chosen. Perhaps it does not matter and they are only looking at seasonal variations. But this are these seasons "typical" or not?

Anthropogenic Emissions: The authors use the 2005 NEI, but it would have been more

appropriate to use this 2011 inventory which is closer to the time of the simulation period. Even more ideal, would be to use emissions generated by CARB that are likely to have local emissions in California better represented. There are papers describing this inventory that at least be cited and the changes in SO2 and NH3 emissions in the SVJ valley (which are likely to be very different that the NEI 2005) will contribute to the nitrate and sulfate errors described in the paper. Since dust is an important factor over a large portion of the year, the differences in anthropogenic emissions are not likely to affect that conclusion. But it would affect the relative contribution of anthropogenic to natural sources over the year.

Model Evaluation: The authors used satellite equivalent potential temperature to evaluate the temperature profiles in the model. As seen in Figure 9, it seems that the vertical resolution is coarse so it is not the best source to examine near-surface temperature gradients. Two of the near-surface AIRS profiles look unrealistic to me. In addition it appears to have a 1 deg uncertainty (which is large for temperature) and is from a 1 degree grid – which will average out substantial temperature variations in areas affected by terrain. Using radiosondes would be a much better way to evaluate the model. The coarse vertical resolution of AIRS also leads to misinterpretations about boundary layer mixing. They claim that boundary layer mixing is too weak and explains why the simulated extinction profiles are wrong in AMJ and JAS. There is simply not enough aerosols around, no matter what the vertical distribution.

Missing Aspects: While the authors have evaluated simulated aerosol composition and PM25/PM10 mass, they have not examined aerosol water. During dry conditions of the summer months, this may not be a large factor contributing to extinction. Aerosol water is likely to become more important aloft, where RH is likely to be higher. But one does not know unless it is examined. Is there significant aerosol water in the simulations? Aerosol water will be influenced by simulated RH, so an evaluation of simulated RH is in order. A second missing aspect is SOA. I assume the version of MOSAIC they use does not include SOA. Yet SOA has been shown to be a major factor in PM25 for

much of the year in California. While SOA concentrations will be lower than dust concentrations (when significant dust is present), it seems that omitting SOA is problem. One motivation factor in the study was related to using and air quality model (such as WRF-Chem) to guide emission control strategies. That would include OC emissions. But it seems that only primary OC is included, so that comparing simulated OC to observed OC is misleading. Also, MOSAIC simulates organic matter (both carbon and oxygen), so do the authors account for the missing oxygen parts in the measurements that are labeled OC? The authors also use a 4-bin version of the model which coarsely represents the aerosol size distribution. The authors should at a minimum discuss how this assumption affects their results and conclusions. It would have been useful to see some sort of evaluation of aerosol size distribution, since that also affects extinction and AOD. So the authors are really not probing all the aspects that affect uncertainties in simulated extinction and AOD.

Model Interpretation: All of the above factors will affect the interpretation of the model results and whether local (due to WRF-Chem) or long-range transport (not WRF-Chem related) sources of dust contribute to the errors in simulated dust concentrations and the vertical distributions. As stated in the summary, the authors claim the errors are largely due to errors in the dust emissions (not clear whether they mean local emissions or those from long-range transport) and vertical mixing. Given how the model has been used, they have not provided sufficient evidence to convince me that is the case.

Specific Comments:

Lines30-31: Change "in cold season" to "in the cold season" and similarly "in warm season" to "in the warm season". This is the first instance of poor use of English in the text. I will not comment on other problems since I seem my role as commenting on the science, rather than correcting the grammar. The authors should use an editor if the co-authors are not willing to help out with the English.

Lines 43-45: This statement is an obvious one and I am not sure it is needed. The

focus of the paper seems to be on dust, so this is a secondary issue. Lines 92-104. This paragraph provides an important motivation for the study, but could be strengthened. Many readers will not know why models, such as WRF-Chem, are needed to develop/verify/modify satellite retrievals. It would be useful to add a few sentences describing how such models are used to demonstrate the purpose. Line 214: "averaging process" is a phrase that is not clear or specific enough. It is not clear how the authors apportion the NEI 2005 emissions to the WRF domain, and the procedure should be some sort of "reapportionment" rather than interpolation. Simple interpolation cannot be used since that would not conserve mass. Did they check to make sure the total mass emitted from NEI 2005 with the WRF domain was actually the same as what was used after the emissions were reapportioned to the WRF domains?

Line 257: The sensitivity experiment mentioned does not contain sufficient details for the reader to know why or how it was performed.

Line 264: The authors start discussion Figure 5c before 5a. Why not change the order of the panels then to match the progression of the discussion in the text?

Line 338: There are far more studies evaluation WRF-Chem in simulating biomass burning than simply the one the first author led.

---

## Referee Comment (RC2) · Anonymous Referee #2 · 3 Jan 2017

In this study, the authors use the WRF-Chem model to simulate the seasonal variability of aerosol properties in the San Joaquin Valley. The authors investigate the roles of 1) horizontal resolution of model; 2) dust emission schemes; and 3) meteorology in modeling aerosol properties and compared the model results against ground-based (e.g. IMPROVE) and satellite (e.g. MISR and CALIPSO) observations. This paper has scientific merit to be published on ACP; however, some major revisions are needed.

General comments:

1. Uncertainties in dust schemes

First of all, the authors did not thoroughly describe the dust schemes in the paper, but only cited a paper by Zhao et al. (2010), in which the two dust schemes are used to simulate the dust emissions over Africa. The parameters "C", the empirical

proportionality constants, in both schemes are tuned for the African dust emissions. Whether the authors use updated or original values for "C" is never discussed in the paper. Since the dust emission schemes are associated with such large uncertainties (in terms of values of C), the discussions in section 4.2 (sensitivity to dust scheme) makes not much sense to the reviewer, because both schemes need to be tuned before any new case studies with different domains, simulation periods, and re-analysis inputs.

In addition, in Zhao et al. (2010), the dust emission schemes are coupled with 8-bin version of MOSIAC, while in Zhao et al. (2013) with MADE/SORGAM. In this paper, the dust emission schemes are coupled with 4-bin version of MOSAIC. Please mention how the dust masses are partitioned in these four bins.

Please also discuss the relative importance of local dust vs. transported dust over SJV.

2. Lack of in-depth analyses

In the paper, the authors demonstrate differences in modeled and observed aerosol properties without giving in-depth analyses. The quality of the paper can be significantly improved if the authors can provide more in-depth analyses other than just quoting conclusions from other papers. Here are three examples:

Lines 239-242: To explain the underestimations of OC in 4km and 20km simulation, the authors quote the explanation from Fast et al. (2014): "low bias in WRF-Chem simulation is primarily due to incomplete understanding of SOA processes." To my knowledge, a simple version of VBS SOA scheme is used in Fast et al. (2014) but not in this Wu et al. paper. If this is the case, then the authors' explanation is definitely wrong. If the VBS SOA scheme is also adopted in this Wu et al. paper, then "incomplete understanding of SOA processes" does not explain the differences between the OC loadings in two cases with different horizontal resolutions because SOA processes are treated the same way in two cases.

Lines 245-248: To explain the low bias in modeled sulfate, the author mention that low

bias in sulfate is also shown at one site Bakersfield in Fast et al. (2014). However, in Fast et al. (2014), the sulfate concentrations over some other sites are reasonable compared to observations. The authors are trying to explain their model results (domain integrated; one-year simulation) by comparing against model results over one site and two-month period from Fast et al. (2014). The authors claim, "it [Fast et al. (2014)] suggests that improvement in understanding the photochemical processes involving sulfate is needed to reproduce seasonal variability of sulfate in the SJV. "; However, Fast et al. (2014) never studies the seasonal variability of aerosol properties.

Section 4.3 The Role of Meteorology: In this section, the authors focus on the role of instability only other than "meteorology". The other meteorological fields also strongly control the aerosol properties, but are never discussed or mentioned in the study. For example, between 4km and 20km, the surface wind fields, which are important for dust emissions, are definitely very different. The precipitation fields, which are important for wet removal processes, are definitely very different between two cases too. The reviewer strongly suggests the authors add these results, because they can also partially explain the differences among three cases (4km, 4km_D2, 20km).

Specific comments:

Figure 1: Add domain-integrated values of daily anthropogenic emissions (miug/day) in each sub figures. Similar to anthropogenic emissions, please add dust emissions for three cases too (not necessarily in figure 1).

Table 2 and Figure 6: it seems that table 2 and Figure 6 provide some same information. It may be better to merge table 2 and Figure 6.

Line 337: Please explain the reason to use climatological fire emissions from GFED instead of using daily fire emission from GFED. The fire emissions from GFED are available for 2013 as mentioned on the website (http://www.globalfiredata.org/).

---

## Referee Comment (RC3) · Anonymous Referee #3 · 3 Jan 2017

This paper shows the WRF-Chem simulation of aerosols in the SJV in California for one year and compares the results with observations of AOD from one AERONET site at Fresno and from MISR for a domain covering SJV, as well as measurements of aerosol mass concentrations of PM2.5, PM10, nitrate, sulfate, EC, OC, and dust from IMPROVE measurements. It tests the effects of using two different model resolution and two dust schemes, and attributes the model problems in matching observed AOD and PM10 to mainly the poor simulation of dust. It is stated in the "Introduction" that the paper a) "serves as the first step for future investigation of the aerosol impact on regional climate and water cycle in California" and b) provides a priori input for remote sensing retrievals for air quality for the MAIA mission.

While this paper has clearly shown the WRF-Chem performance over SJV that provides useful information, it lacks the vigor and thoroughness in the analysis and in-

terpretation, and the information presented in the paper is insufficient in helping understand the problems of the model. Given the goal of using such a model for MAIA retrieval and for climate study, much more in-depth analysis and vigorous diagnostics is necessary in order for the model improvements to be useful for those purposes. Although the content is suitable for ACP, major revisions are necessary before the paper can be considered again for publication.

General comments:

1. Dust simulations: The authors have concluded that the dust simulation is the major problem for model to capture the observed aerosol amount and variability in the warm months. Switching from GOCART to DUSTRAN just shows different problems but does not resolve the issue. However, there is no any explanation on the differences between the two schemes in terms of emission strength, source location, parameterization of dust mobilization, and deposition in order to understand why the dust amount and seasonal cycles are so different between the two schemes and yet none can capture the observations. Without understanding the cause of the problem, future improvement is not possible.

2. Non-dust aerosols: Figure 4 clearly shows that the model does not have much skill to simulate sulfate and OC, but the problem has not been investigated. The ammonium is completely left out, which is an important part of total aerosol mass. Also, large fraction of aerosol is classified as "other", but it is not clear what the "other" aerosols are in both model and IMPROVE data.

3. Optical properties: It is also not clear how AOD and aerosol extinction are computed from the simulated aerosol mass. Is aerosol microphysics package used for calculating particle sizes and mixing state? How is mass-based aerosol converted to extinction and AOD? Is the relative humidity considered in these calculations?

4. Chemistry: Nitrate, sulfate, and a significant fraction of OC are secondary aerosols that are produced by chemical reactions of their gaseous precursors in the atmosphere. The authors attribute the high bias of model-simulated nitrate to "high bias in nitrate emission", which is erroneous. The diagnostics should involve investigations of nitrate precursors such as NOx and HNO3, and also the formation of nitrate via heterogeneous reactions on dust and sea salt surfaces and homogeneous reactions in the sulfate-nitrate-ammonium system. It is not clear how WRF-Chem deals with nitrate formations and which is the major reaction pathway for nitrate aerosol production. Same as sulfate – it is formed via gas and aqueous phase reactions of SO2. Better diagnostics of the problem is needed.

5. Other physical processes: Dry and wet depositions are the major removal processes for aerosols. The seasonal cycles of these processes also need to be investigated. For example, can the differences in seasonal variations between model and obs be partly explained by the differences in simulated and measured precipitation amount that determines the wet removal of aerosols? Or if the winds are realistically simulated in WRF-Chem that not only affect the dust emission, but also advection, both have profound effect on aerosol temporal and spatial distributions?

6. Meteorological fields: The only meteorological field compared in the paper is the equivalent potential temperature, which provides information on the atmospheric stability. Other important met fields, such as precipitation and wind speed/direction, as mentioned above, plays key roles in aerosol removal, transport, and wind-driven emissions of dust and sea salt but have not even mentioned in the paper. In addition, these fields and the physical processes driven by them are resolution-dependent, so the role of these met fields should be examined at different spatial resolutions.

7. Lateral boundary conditions: The effects of lateral boundary condition should be examined, or at lease discussed, particularly because of SJV's geophysical locations that is susceptible to the transpacific transport. How much of the aerosol species and their precursor gases are regionally/locally produced vs. imported from the lateral boundary, and how they affect the seasonal cycle? In other words, are the features/problems mainly produced by WRF-Chem? How important is the lateral boundary conditions to

different aerosol species?

8. Emissions: It seems the anthropogenic and biomass burning emissions used in this work are not up to date. For example, why the authors choose to use NEI05 emissions instead of more recent ones (e.g., NEI 2011 or NEI 2014) to better match the simulated time period (2012-2013)? Why GFEDv2 is preferred instead of GFEDv3 that was released a few years ago or GFEDv4 that has been available since 2015?

9. Model-data comparison: 1) For AOD, there is only one AERONET site in the study region, and MISR's spatial coverage is limited. Why not use MODIS, which has a much better spatial coverage to have a better representation of "monthly average", in addition or even instead of using MISR? 2) Which months are defined as "cold" or "warm" months? 3) More statistical quantities are needed to mark the agreement between model and observations, including correlation coefficients and seasonal/annual bias. 4) The authors should avoid using the subjective adjectives, such as "good agreement", "reasonably well", etc., to describe the comparisons between model and observations. More objective and quantitative methods and presentations are needed. 5) Given that air quality changes quite a bit day to day and air quality forecasts are given on daily bases, why all the comparisons are done on monthly time scale instead of daily or sub-daily?

10. The most important step forward is to understand the causes of deficiencies in the model and suggest/incorporate improvements for better results. However, the current paper does not offer those aspects.

Specific comments

Page 5, line 72-82: I wonder why Fast et al 2014 and Zhao et al 2013 were able to "reasonably" represent the observations with the same WRF-Chem model, either in the warm months (Fast) or on annual bases (Zhao), but this work has difficulties to do the same?

[Figure]

Page 5, line 83: I don't think the word "extend" is appropriate – this study only focuses on SJV while Fast and Zhao showed large regions in CA.

Page 6, line 102-104: I don't get it – why simulation for SJV is critical to MAIA? Is MAIA only focuses on SJV?

Page 7, line 116: Why are the original wavelength(s) from AEORNET that you used to interpolate to 550 nm?

Page 8, line 146: What does "speciated" mean here? There is no aerosol species information from the CALIOP data. Marine, polluted continental, etc. provided by CALIOP are aerosol types, not species.

Page 9, line 179-180: How is convective transport (and removal) of aerosols simulated in 4-km resolution?

Page 9-10, line 183-184: Was the overestimation by MOZART in the free troposphere a factor of 2 such that the concentrations had to be divided by 2? If the overestimation was only in the free troposphere, why the concentrations in the lower atmosphere and BL were also divided by 2?

Page 10, line 198: Are the dust emissions in the GOCART and DUSTRAN also available in 20 and 4 km resolutions? What are the major differences between GOCART and DUSTRAN schemes?

Page 11, first paragraph in section 4.1: What PM2.5 species and precursor gases are emitted? Have you checked the domain budget between 4 and 20 km resolution to ensure the total emission for all species are identical with these different resolutions?

Page 11, line 215: How was AOD calculated without having information of PM2.5 composition? For example, dust and BC have very different mass to extinction conversion factor, known as mass extinction efficiency (MEE). There is no single MEE for a generic PM2.5 or PM10.

Page 12, line 237: As I said earlier, nitrate is not emitted but chemically produced. The precursor emission/concentration/transport/chemistry have to be examined to explain the nitrate.

Page 12, line 238: Why is simulation over Texas relevant here?

Page 12, line 242: Be specific on what "SOA processes" is referred here.

Page 12, line 244 and 246: Be quantitative – what is the standard of "good agreement"?

Page 12, line 250: How large is the "large low bias"?

Page 13, line 253-254: "The 4km simulation has better agreement. . .", but only in the cold season.

Page 13, line 254-255: "The 4km simulation captures seasonal variability of PM2.5 and its speciation": From Figure 4, the seasonal variability for the PM2.5 species are very similar between the 4- and 20-km simulations, only the concentrations are higher from the 4km simulation. The seasonal variability of PM2.5 sulfate and OC are not capture by both 4 and 20 km simulations.

Page 13, line 267-268: The 4km_D2 overestimates PM2.5 by 52%, but it overestimates the PM2.5_dust by up to a factor of 4 in the warm season!

Page 13, line 270-272: As I suggested earlier, please show correlation coefficients on all comparisons (in addition to the bias), which indicates how model and data agree on seasonal variations.

Page 14, line 285-286: How much better does 4km_D2 agree with MISR than other simulations? Visually, JAS is still nowhere near MISR, and AMJ is higher than MISR. Please quantify the degree of agreement.

Page 14, line 290-292: I don't understand the statement of "reasonably capture the vertical distribution", even though the model has "low biases in the boundary layer and high biases in the free troposphere". To me, this is rather "unreasonable".

Page 15, line 298-299: "...suggesting relative good performance...": How good? Figure shows poor agreement between obs and model for sulfate and OC, so they are not "good" at all.

Page 15, line 303: How to explain that dust from 4km_D2 is way too high but the extinction in the boundary layer is still way too low?

Page 15, line 313 and 316: If the model has weak vertical mixing, the aerosols should be trapped within the BL and not transported to high altitudes. But the model actually overestimates the aerosol at high altitude – what is the source of high altitude aerosol?

Page 16, line 321-322: This precisely indicates the need to quantify the role of chemical boundary conditions.

Page 16, line 323-324, "good performance...": But in JFM the model results are much higher (by a factor of infinity?) at above 1.5 km! How can that be evaluated as "good"?

Page 16, line 330: "reasonable simulation", "good representation" – what are the measures of reasonable and good here?

Page 16, line 337: Please explain what "climatological fire emissions" mean.

Page 16, line 339-340: Why can Wu et al do it right for South America fire but cannot do it for California? What are the major obstacles?

Page 17, line 371-372: No need to spell out what GOCART and DUSTRAN stand for at the last part of the paper, since they have been introduced and used many times earlier in the text.

Page 17, line 383-385: Unfortunately, I cannot see how the evaluation in this study can be apply to other regions to ensure that aerosols are simulated correctly for the right reasons. This paper has shown the problems but has not shown how to solve the problems with what approach.

---

## Author Comment (AC1) · 7 Mar 2017

The reviewers' insightful comments are highly appreciated. Below we have listed the referees' comments in black and our response in blue.

We have made the following major revisions in the revised manuscript:
1. More descriptions of aerosol properties simulated in the model are added in the revised manuscript.
2. Two aerosol precursors ($NO_2$ and $SO_2$) observed by EPA are included to diagnose model biases in $NO_3$ and $SO_4$, respectively.
3. Analyses of meteorological variables, including temperature, relative humidity, wind speed and precipitation, are included.
4. Analysis of Ångström exponent is included to diagnose the model simulated aerosol particle size.
5. More quantitative information, including correlation and bias, is included in the discussion.
6. We have performed some sensitivity experiments to provide more in-depth analyses on model results, including changing the anthropogenic emission source (20km_NEI11), the chemical boundary conditions (20km_BC1) and the PBL scheme (20km_P7).
7. A bug in calculating equivalent potential temperature is fixed in the revised manuscript. The unit of relative humidity was wrong in previous version. The updated profiles of equivalent potential temperature do not change the conclusions of this study.
8. The OC (organic carbon) from observations are converted to OM (organic matter), which is simulated in the model, by multiplying by 1.4 to account for hydrogen, oxygen, etc.

**Anonymous Referee #1**

This paper examines the performance of a regional-scale chemical transport model in representing aerosol properties in the San Joaquin Valley over a one year period. The model is compared with surface measurements of composition and AOD as well as satellite measurements. The motivation for the paper is sufficient (although could be improved), but the main weakness is their approach and interpretation of the simulations. In addition, the paper is poorly written.

**Major Comments:**
The most important problem the manuscript has is how the model was configured to address the purpose of the study. WRF-Chem is a useful tool, but as with all models can only perform well when it is configured properly.

The following is a discussion of items the authors should consider to revise and/or address.

Domain and Dust Emissions: It is clear that the model domain is larger than the one shown in Figure 1. But it is hard for me to assess the importance of dust emissions since those are not shown. For local sources, dust is likely generated in the desert areas to the southwest of the SJV. It would be useful to show the emission regions from GOCART and DUSTRAN. My understanding is that the emission regions in DUSTRAN as implemented in WRF-Chem are rather ad hoc. They may depend on vegetation type. I suspect that dust is being generated locally in the SJV in DUSTRAN but not in GOCART.

Thanks for the suggestion. Dust emissions are included in Figure 7 in the revised manuscript (also in the following Figure 1). As the reviewer hinted, dust is being generated locally in the SJV in DUSTRAN but not in GOCART. Discussions about the differences between DUSTRAN and GOCART are included in the last two paragraphs of section 3 in the revised manuscript.

[Figure]

Figure 1. Seasonal mean of dust emission rate ($\mu$g m$^{-2}$ s$^{-1}$) for (upper panel) GOCART; (lower panel) DUSTRAN.

The authors mention how many grid nodes are used in the vertical direction, but should give an idea of the vertical resolution near the surface that will affect dust emissions.

The vertical resolution from surface to 1 km gradually increases from 28 m to 250 m. It is clarified in Line 204 of the revised manuscript.

Dust emissions will depend in part on wind speed, and representing wind speed in California depends a lot on circulations affected by terrain. Both a fine horizontal and vertical resolution is needed to represent those winds that will affect dust emissions. It is not clear how well the model performed in winds – particularly over the dust emission regions. While some evaluation of the thermodynamic structure is given, there is nothing for the winds.

The evaluation of wind speed comparing to surface observations from CIMIS (California Irrigation Management Information System) is included in Figure 2b of the revised manuscript. The model simulations underestimate wind speed in the cold season. In the warm season, the 20km run underestimates wind speed except June while the 4km run overestimates wind speed, which indicates wind speed is not the main reason for AOD biases in the warm season. Discussions of wind speed impacts are included in the first paragraph of section 4.3 in the revised manuscript.

[Figure]

Figure 2. Simulated monthly 10-m wind speed (m/s) at Fresno, CA compared to CIMIS (California Irrigation Management Information System) observations.

Boundary Conditions: The authors half the amount of aerosols from MOZART following Fast et al. (2014). But the errors in a coarse global model, like MOZART, will likely change in time and depend on meteorological conditions. There is no sensitivity results or evidence whether such a change in boundary conditions is warranted in the present study. I believe the version of MOZART the authors use prescribes dust using climatology which would affect the simulations over California. The potential errors in MOZART that will contribute to AOD over California will likely vary over a year-long period.

We have run two sensitivity experiments with DUSTRAN at 20 km resolution, one with MOZART divided by 2 (20km_D2) and the other with original MOZART (20km_BC1). AOD maps are shown in the Supplementary Fig. 1 and the following figure. It is clear that the 20km_BC1 run overestimates AOD in the rural regions from OND to AMJ. Both the 20km_D2 (BC0.5) and 20km_BC1 (BC1) runs underestimate AOD in the rural regions in JAS, which indicates chemical boundary condition is not the main reason for the underestimation of JAS AOD in the simulations. Thus, we keep the setting of halving the amount of aerosols from MOZART in the simulations.

[Figure]

Figure 3. Spatial distribution of seasonal mean 550 nm AOD from MISR, the 20km_D2 (BC0.5) and 20km_BC1 (BC1) in WY2013.

Simulation Period: On line 167, the authors state that the simulation period is from 2012 to 2013. There is no rationale as to why this period is chosen. Perhaps it does not matter and they are only looking at seasonal variations. But this are these seasons "typical" or not?
We are only looking at seasonal variations. Similar results are also shown in our initial experiment in WY2012. For further investigation of model performance by comparing with the DISCOVER-AQ field campaign datasets in 2013 (a future study), we switched all our experiments to WY2013.

Anthropogenic Emissions: The authors use the 2005 NEI, but it would have been more appropriate to use this 2011 inventory which is closer to the time of

the simulation period. Even more ideal, would be to use emissions generated by CARB that are likely to have local emissions in California better represented. There are papers describing this inventory that at least be cited and the changes in SO2 and NH3 emissions in the SVJ valley (which are likely to be very different that the NEI 2005) will contribute to the nitrate and sulfate errors described in the paper. Since dust is an important factor over a large portion of the year, the differences in anthropogenic emissions are not likely to affect that conclusion. But it would affect the relative contribution of anthropogenic to natural sources over the year.

The 2011 NEI was not available in the WRF-Chem emission datasets when we initiated this study. We have run two sensitivity experiments with the 2011 NEI (20km_NEI11) and 2005 NEI (20km_D2) at 20 km resolution with the DUSTRAN dust scheme. Results are shown in the supplementary materials and the following figures. The differences between NEI11 and NEI05 are small comparing to the identified model biases in this study. As the reviewer pointed out, the differences in $SO_4$ and $NH_4$ are relatively large. However, $SO_4$ in NEI11 has larger biases than $SO_4$ in NEI05.

As shown in Fast et al. (2014), "reducing the default CARB emissions by 50% led to an overall improvement in many simulated trace gases and black carbon aerosol at most sites and along most aircraft flight paths; however, simulated organic aerosol was closer to observed when there were no adjustments to the primary organic aerosol emissions". We can see all the emission datasets (CARB, NEI11 and NEI05) have uncertainties in the aerosol emissions. We decide to keep our current model setup and include discussions of the uncertainty in the emission data sources in the revised manuscript.

[Figure]

Figure 4. Spatial distribution of seasonal mean 550 nm AOD from the 20km_NEI11 (NEI11) and 20km_D2 (NEI05) runs in WY2013.

[Figure]

Figure 5. Aerosol mass (μg m$^{-3}$) for different species from EPA-CSN (OBS), the NEI05 (20km_D2) and NEI11 (20km_NEI11) runs at Fresno, CA. PM2.5_NO$_3$ represents NO$_3$ with diameter ≤ 2.5 μm. Similar definition for SO$_4$, EC, OM, dust and NH$_4$ in the figures.

Model Evaluation: The authors used satellite equivalent potential temperature to evaluate the temperature profiles in the model. As seen in Figure 9, it seems that the vertical resolution is coarse so it is not the best source to examine near-surface temperature gradients. Two of the near-surface AIRS profiles look unrealistic to me. In addition it appears to have a 1 deg uncertainty (which is large for temperature) and is from a 1 degree grid – which will average out substantial temperature variations in areas affected by terrain. Using radiosondes would be a much better way to evaluate the model. The coarse vertical resolution of AIRS also leads to misinterpretations about boundary layer mixing. They claim that boundary layer mixing is too weak and explains why the simulated extinction profiles are wrong in AMJ and JAS.

There is simply not enough aerosols around, no matter what the vertical distribution.

Unfortunately, there is no routine radiosonde observation available in the SJV. AIRS data have been extensively evaluated using radiosondes in other regions. We agree that the coarse vertical resolution of AIRS data cannot fully resolve near-surface temperature gradients. However, AIRS is the best dataset currently available to evaluate seasonal variations of the vertical temperature/moisture profiles in the model simulations over the SJV. Evaluation of surface temperature/RH is conducted by comparing with surface observations in the revised manuscript. Results are consistent with evaluations of vertical profiles comparing to AIRS. More analyses of aerosol biases in the boundary layer are included in the revised manuscript.

We have found that the unit of RH is wrong in our code to calculate equivalent potential temperature. It is fixed in the revised manuscript. The profiles look reasonable now. It doesn't change the conclusions of this study.

Missing Aspects: While the authors have evaluated simulated aerosol composition and PM25/PM10 mass, they have not examined aerosol water. During dry conditions of the summer months, this may not be a large factor contributing to extinction. Aerosol water is likely to become more important aloft, where RH is likely to be higher. But one does not know unless it is examined. Is there significant aerosol water in the simulations?
Aerosol water will be influenced by simulated RH, so an evaluation of simulated RH is in order.

Evaluation of simulated RH is included in the supplementary and discussed in the revised manuscript. As shown in following figures, there are dry biases in the model simulations. However, due to the relative dry environment (RH<50%) in the warm season, the dry bias may not be responsible for the underestimation of aerosol extinction in the boundary layer and column-integrated AOD through hygroscopic effects (Feingold and Morley, 2003).

[Figure]

Figure 6. Monthly mean 2-m RH (%).

[Figure]

Figure 7. Vertical profile of seasonal mean relative humidity (%) in the WRF-Chem simulations comparing to AIRS. The 20km (not shown) run is similar to the 20km_D2 run while the 4km run (not shown) is similar to the 4km_D2 run.

A second missing aspect is SOA. I assume the version of MOSAIC they use does not include SOA. Yet SOA has been shown to be a major factor in PM25 for much of the year in California. While SOA concentrations will be lower than dust concentrations (when significant dust is present), it seems that omitting SOA is problem. One motivation factor in the study was related to using and air quality model (such as WRF-Chem) to guide emission control strategies. That would include OC emissions. But it seems that only primary OC is included, so that comparing simulated OC to observed OC is misleading. SOA processes are not included in our simulation. Fast et al. (2014) used the simplified two-product volatility basis set parameterization to simulate equilibrium SOA partitioning in the WRF-Chem model. SOA is still underestimated in their simulation in May and June. We tried to run the WRF-Chem model at 20 km resolution (20km_VBS2) following the settings in Fast et al. (2014). However, our simulation can only produce comparable AOD in AMJ while AOD in other seasons are underestimated. Since it is challenging to correctly represent SOA processes in regional climate models, we keep our current settings and discuss the impact of SOA processes in the revised manuscript.

[Figure]

Figure 8. Spatial distribution of seasonal mean 550 nm AOD from MISR, 20km_D2 and 20km_VBS2 in WY2013.

Also, MOSAIC simulates organic matter (both carbon and oxygen), so do the authors account for the missing oxygen parts in the measurements that are labeled OC?

Thanks for your comment. The observed OC is converted to organic matter (multiply by 1.4) to compare with the simulated organic matter in the revised manuscript.

The authors also use a 4-bin version of the model which coarsely represents the aerosol size distribution. The authors should at a minimum discuss how this assumption affects their results and conclusions.

Discussion of the impacts of this assumption is provided in the revised manuscript as following:

"Zhao et al. (2013a) compared the impacts of aerosol size partition on dust simulations. It showed that the 4-bin approach reasonably produces dust mass loading and AOD comparing to the 8-bin approach. The size distribution of the 4-bin approach follows that of the 8-bin approach with coarser resolution, resulting in ±5% difference on the ratio of PM2.5-dust/PM10-dust in dusty regions. Dust number loading and absorptivity are biased high in the 4-bin approach comparing to the 8-bin approach."

It would have been useful to see some sort of evaluation of aerosol size distribution, since that also affects extinction and AOD. So the authors are really not probing all the aspects that affect uncertainties in simulated extinction and AOD.

Evaluation of Ångström exponent (AE), an indicator of aerosol particle size, is included in Fig. 4b of the revised manuscript. WRF-Chem captures the seasonal variability of the AE well, with a correlation of 0.90 in both the 20km and 4km simulations. The magnitude of AE is also approximately simulated in the cold season, with a mean of 1.15 (1.20) in the 20km (4km) runs compared to 1.33 in the observation. However, the simulated AE is underestimated by ~30% in the warm season, indicating that the simulated particle size is biased high during this period.

[Figure]

Figure 9. Monthly mean Ångström Exponent between 600 nm and 400 nm at Fresno, CA.

Model Interpretation: All of the above factors will affect the interpretation of the model results and whether local (due to WRF-Chem) or long-range transport (not WRF-Chem related) sources of dust contribute to the errors in simulated dust concentrations and the vertical distributions. As stated in the summary, the authors claim the errors are largely due to errors in the dust emissions (not clear whether they mean local emissions or those from long-range transport) and vertical mixing. Given how the model has been used, they have not provided sufficient evidence to convince me that is the case.

The simulated aerosol extinction in the free troposphere above the boundary layer is close to or larger than CALIOP, suggesting that aerosols transported from remote areas through chemical boundary conditions (e.g., the differences between the 20km_BC1 and 20km_D2 runs in Supplementary Fig. 3) may not be the major factor contributing to the underestimation of dust in the boundary layer in the SJV. It is clarified in the revised manuscript.

[Figure]

Figure 10. Vertical distribution of seasonal mean 532 nm aerosol extinction coefficient (km⁻¹) from CALIOP, CALIOP_nodust, and the WRF-Chem (20km_D2, 20km_BC1 and 20km_NEI11) simulations over the red box region in Fig. 1a in WY2013.

**Specific Comments:**

Lines30-31: Change "in cold season" to "in the cold season" and similarly "in warm season" to "in the warm season". This is the first instance of poor use of English in the text. I will not comment on other problems since I seem my role as commenting on the science, rather than correcting the grammar. The

authors should use an editor if the co-authors are not willing to help out with the English.

Careful proofreading is provided by the co-authors (James Campbell and Hui Su) for the revised manuscript.

Lines 43-45: This statement is an obvious one and I am not sure it is needed. The focus of the paper seems to be on dust, so this is a secondary issue.

Removed per your suggestion.

Lines 92-104. This paragraph provides an important motivation for the study, but could be strengthened. Many readers will not know why models, such as WRF-Chem, are needed to develop/verify/modify satellite retrievals. It would be useful to add a few sentences describing how such models are used to demonstrate the purpose.

The following sentences are added in the revised manuscript to describe how the WRF-Chem model will be used in the MAIA retrieval algorithm.

"A significant challenge for aerosol remote sensing in retrieving spatial information on specific aerosol types, especially near the surface, is due to the lack of information on the vertical distribution of aerosols in the atmospheric column and limited instrument sensitivity to aerosol types over land. The WRF-Chem model will be used to provide near-real-time estimation of particle properties, aerosol layer heights, and aerosol optical depths (AOD) to constrain the instrument-based PM retrievals."

Line 214: "averaging process" is a phrase that is not clear or specific enough. It is not clear how the authors apportion the NEI 2005 emissions to the WRF domain, and the procedure should be some sort of "reapportionment" rather than interpolation. Simple interpolation cannot be used since that would not conserve mass. Did they check to make sure the total mass emitted from NEI 2005 with the WRF domain was actually the same as what was used after the emissions were reapportioned to the WRF domains?

Reworded to "reapportionment process". We use the standard emission conversion program in the WRF-Chem (convert_emiss.exe) to reapportion the anthropogenic emission. The domain-averaged emission rates for the 20km and 4km simulations are quite similar, as listed in the updated Fig. 1.

Line 257: The sensitivity experiment mentioned does not contain sufficient details for the reader to know why or how it was performed.

Reworded as: "The underestimation also exists in a sensitivity experiment (not shown) with the same model setups except initialized in April, indicating that the identified model biases in the warm season are not caused by potential model drift after a relatively long simulation period."

Line 264: The authors start discussion Figure 5c before 5a. Why not change the order of the panels then to match the progression of the discussion in the text?
Order changed as suggested.

Line 338: There are far more studies evaluation WRF-Chem in simulating biomass burning than simply the one the first author led.
Two more references (Grell et al., 2011; Archer-Nicholls et al., 2015) are included in the revised manuscript.

---

## Author Comment (AC2) · 7 Mar 2017

The reviewers' insightful comments are highly appreciated. Below we have listed the referees' comments in black and our response in blue.

We have made the following major revisions in the revised manuscript:
1. More descriptions of aerosol properties simulated in the model are added in the revised manuscript.
2. Two aerosol precursors ($NO_2$ and $SO_2$) observed by EPA are included to diagnose model biases in $NO_3$ and $SO_4$, respectively.
3. Analyses of meteorological variables, including temperature, relative humidity, wind speed and precipitation, are included.
4. Analysis of Ångström exponent is included to diagnose the model simulated aerosol particle size.
5. More quantitative information, including correlation and bias, is included in the discussion.
6. We have performed some sensitivity experiments to provide more in-depth analyses on model results, including changing the anthropogenic emission source (20km_NEI11), the chemical boundary conditions (20km_BC1) and the PBL scheme (20km_P7).
7. A bug in calculating equivalent potential temperature is fixed in the revised manuscript. The unit of relative humidity was wrong in previous version. The updated profiles of equivalent potential temperature do not change the conclusions of this study.
8. The OC (organic carbon) from observations are converted to OM (organic matter), which is simulated in the model, by multiplying by 1.4 to account for hydrogen, oxygen, etc.

**Anonymous Referee #2**
In this study, the authors use the WRF-Chem model to simulate the seasonal variability of aerosol properties in the San Joaquin Valley. The authors investigate the roles of 1) horizontal resolution of model; 2) dust emission schemes; and 3) meteorology in modeling aerosol properties and compared the model results against ground-based (e.g. IMPROVE) and satellite (e.g. MISR and CALIPSO) observations. This paper has scientific merit to be published on ACP; however, some major revisions are needed.

**General comments:**
1. Uncertainties in dust schemes
First of all, the authors did not thoroughly describe the dust schemes in the paper, but only cited a paper by Zhao et al. (2010), in which the two dust schemes are used to simulate the dust emissions over Africa. The parameters "C", the empirical proportionality constants, in both schemes are tuned for the

African dust emissions.Whether the authors use updated or original values for "C" is never discussed in the paper. Since the dust emission schemes are associated with such large uncertainties (in terms of values of C), the discussions in section 4.2 (sensitivity to dust scheme) makes not much sense to the reviewer, because both schemes need to be tuned before any new case studies with different domains, simulation periods, and re-analysis inputs.

In our study, we use the original "C" in Ginoux et al. (2001) and Shaw et al. (2008). It is clarified in the revised manuscript. More analyses about the two dust emissions are also included in the revised manuscript. The low emission in GOCART is due to the source function for potential wind erosion. We agree that "C" in DUSTRAN needs to be tuned for better agreement with observations. As our simulations show high biases of dust at the surface, the "C" value in DUSTRAN are not likely the main reason for low aerosols in the boundary layer in the warm season.

In addition, in Zhao et al. (2010), the dust emission schemes are coupled with 8-bin version of MOSIAC, while in Zhao et al. (2013) with MADE/SORGAM. In this paper, the dust emission schemes are coupled with 4-bin version of MOSAIC. Please mention how the dust masses are partitioned in these four bins.

The dust masses are partitioned into four size bins (0.039-0.156 μm, 0.156-0.625 μm, 0.625-2.5 μm, and 2.5-10.0 μm dry diameter), respectively. Aerosols are considered to be spherical and internally mixed in each bin (Barnard et al., 2006; Zhao et al., 2013b). The bulk refractive index for each particle is calculated by volume averaging in each bin. Mie calculations as described by Ghan et al. (2001) are used to derive aerosol optical properties (such as extinction, single-scattering albedo, and the asymmetry parameter for scattering) as a function of wavelength. It is clarified in the revised manuscript. Discussion of the impacts of bin-size assumption is provided in the revised manuscript.

Please also discuss the relative importance of local dust vs. transported dust over SJV.

The simulated aerosol extinction in the free troposphere above the boundary layer is close to or larger than CALIOP, suggesting that aerosols transported from remote areas through chemical boundary conditions (e.g., the differences between the 20km_BC1 and 20km_D2 runs in Supplementary Fig. 3) may not be the major factor contributing to the underestimation of dust in

the boundary layer in the SJV. It is clarified in the revised manuscript.

[Figure]

Figure 1. Vertical distribution of seasonal mean 532 nm aerosol extinction coefficient (km$^{-1}$) from CALIOP, CALIOP_nodust, and the WRF-Chem (20km_D2, 20km_BC1 and 20km_NEI11) simulations over the red box region in Fig. 1a in WY2013.

2. Lack of in-depth analyses
In the paper, the authors demonstrate differences in modeled and observed aerosol properties without giving in-depth analyses. The quality of the paper can be significantly improved if the authors can provide more in-depth

analyses other than just quoting conclusions from other papers. Here are three examples:

Following three reviewers' comments, more analyses on differences in modeled and observed aerosol properties are given in section 4 of the revised manuscript.

Lines 239-242: To explain the underestimations of OC in 4km and 20km simulation, the authors quote the explanation from Fast et al. (2014): "low bias in WRF-Chem simulation is primarily due to incomplete understanding of SOA processes." To my knowledge, a simple version of VBS SOA scheme is used in Fast et al. (2014) but not in this Wu et al. paper. If this is the case, then the authors' explanation is definitely wrong. If the VBS SOA scheme is also adopted in this Wu et al. paper, then "incomplete understanding of SOA processes" does not explain the differences between the OC loadings in two cases with different horizontal resolutions because SOA processes are treated the same way in two cases.

Thanks for the insightful comment. We have checked our setting and confirmed that SOA processes are not included in our current setting. We tried to run the WRF-Chem model at 20 km resolution (20km_VBS2) following the settings in Fast et al. (2014). However, that simulation produces reasonably AOD in AMJ while AOD in other seasons are underestimated. We keep our current settings and discuss the impacts of SOA processes in the revised manuscript. The statement of "incomplete understanding of SOA processes" is removed in the revised manuscript.

[Figure]

Figure 2. Spatial distribution of seasonal mean 550 nm AOD from MISR, the 20km_D2 and 20km_VBS2 simulations in WY2013.

Lines 245-248: To explain the low bias in modeled sulfate, the author mention that low bias in sulfate is also shown at one site Bakersfield in Fast et al. (2014). However, in Fast et al. (2014), the sulfate concentrations over some other sites are reasonable compared to observations. The authors are trying to explain their model results (domain integrated; one-year simulation) by comparing against model results over one site and two-month period from Fast et al. (2014). The authors claim, "it [Fast et al. (2014)] suggests that improvement in understanding the photochemical processes involving sulfate is needed to reproduce seasonal variability of sulfate in the SJV. "; However, Fast et al. (2014) never studies the seasonal variability of aerosol properties. We have removed this statement and include more discussions (precursor and marine intrusions) in the revised manuscript.

Section 4.3 The Role of Meteorology: In this section, the authors focus on the role of instability only other than "meteorology". The other meteorological fields also strongly control the aerosol properties, but are never discussed or mentioned in the study. For example, between 4km and 20km, the surface wind fields, which are important for dust emissions, are definitely very different. The precipitation fields, which are important for wet removal processes, are definitely very different between two cases too. The reviewer

strongly suggests the authors add these results, because they can also partially explain the differences among three cases (4km, 4km_D2, 20km).

Evaluation of temperature, RH, wind speed and precipitation are included in section 4.3 of the revised manuscript and the supplementary. More discussions of meteorological impacts on aerosol simulations are also included in the revised manuscript. Biases in surface wind speed and precipitation may not be the main reasons for the identified aerosol biases in the boundary layers during the warm season.

**Specific comments:**
Figure 1: Add domain-integrated values of daily anthropogenic emissions (miug/day) in each sub figures. Similar to anthropogenic emissions, please add dust emissions for three cases too (not necessarily in figure 1).

We add the domain-averaged PM2.5 emission rate in each sub figure. Dust emissions are shown in Fig. 8 in the revised manuscript and the following figure.

[Figure]

Figure 3. Mean dust emission rate ($\mu g\ m^{-2}\ s^{-1}$) from the 4km and 4km_D2 runs.

Table 2 and Figure 6: it seems that table 2 and Figure 6 provide some same information. It may be better to merge table 2 and Figure 6.

Because some reader may be more interested in magnitude while other may be more interested in relative contribution, we prefer to keep both Table 2

(Table 3 in the revised manuscript) and Fig. 6 (Fig. 10 in the revised manuscript).

Line 337: Please explain the reason to use climatological fire emissions from GFED instead of using daily fire emission from GFED. The fire emissions from GFED are available for 2013 as mentioned on the website (http://www.globalfiredata.org/).

We use the standard emission preparation program (prep_chem_sources_v1.5) for the WRF-Chem model to generate our fire emissions. Currently, only GFEDV2.1 is available in this program. Since fire emissions are not the major issues in our current simulations, we keep current settings.

---

## Author Comment (AC3) · 7 Mar 2017

The reviewers' insightful comments are highly appreciated. Below we have listed the referees' comments in black and our response in blue.

We have made the following major revisions in the revised manuscript:
1. More descriptions of aerosol properties simulated in the model are added in the revised manuscript.
2. Two aerosol precursors ($NO_2$ and $SO_2$) observed by EPA are included to diagnose model biases in $NO_3$ and $SO_4$, respectively.
3. Analyses of meteorological variables, including temperature, relative humidity, wind speed and precipitation, are included.
4. Analysis of Ångström exponent is included to diagnose the model simulated aerosol particle size.
5. More quantitative information, including correlation and bias, is included in the discussion.
6. We have performed some sensitivity experiments to provide more in-depth analyses on model results, including changing the anthropogenic emission source (20km_NEI11), the chemical boundary conditions (20km_BC1) and the PBL scheme (20km_P7).
7. A bug in calculating equivalent potential temperature is fixed in the revised manuscript. The unit of relative humidity was wrong in previous version. The updated profiles of equivalent potential temperature do not change the conclusions of this study.
8. The OC (organic carbon) from observations are converted to OM (organic matter), which is simulated in the model, by multiplying by 1.4 to account for hydrogen, oxygen, etc.

**Anonymous Referee #3**
This paper shows the WRF-Chem simulation of aerosols in the SJV in California for one year and compares the results with observations of AOD from one AERONET site at Fresno and from MISR for a domain covering SJV, as well as measurements of aerosol mass concentrations of PM2.5, PM10, nitrate, sulfate, EC, OC, and dust from IMPROVE measurements. It tests the effects of using two different model resolution and two dust schemes, and attributes the model problems in matching observed AOD and PM10 to mainly the poor simulation of dust. It is stated in the "Introduction" that the paper a) "serves as the first step for future investigation of the aerosol impact on regional climate and water cycle in California" and b) provides a priori input for remote sensing retrievals for air quality for the MAIA mission.
While this paper has clearly shown the WRF-Chem performance over SJV that provides useful information, it lacks the vigor and thoroughness in the analysis and interpretation, and the information presented in the paper is

insufficient in helping understand the problems of the model. Given the goal of using such a model for MAIA retrieval and for climate study, much more in-depth analysis and vigorous diagnostics is necessary in order for the model improvements to be useful for those purposes. Although the content is suitable for ACP, major revisions are necessary before the paper can be considered again for publication.

**General comments:**

1. Dust simulations: The authors have concluded that the dust simulation is the major problem for model to capture the observed aerosol amount and variability in the warm months. Switching from GOCART to DUSTRAN just shows different problems but does not resolve the issue. However, there is no any explanation on the differences between the two schemes in terms of emission strength, source location, parameterization of dust mobilization, and deposition in order to understand why the dust amount and seasonal cycles are so different between the two schemes and yet none can capture the observations. Without understanding the cause of the problem, future improvement is not possible.

More descriptions and analyses of the two dust schemes are provided in the revised manuscript for better understanding the cause of the problem. For details, please see the last two paragraphs of section 3 in the revised manuscript.

2. Non-dust aerosols: Figure 4 clearly shows that the model does not have much skill to simulate sulfate and OC, but the problem has not been investigated. The ammonium is completely left out, which is an important part of total aerosol mass. Also, large fraction of aerosol is classified as "other", but it is not clear what the "other" aerosols are in both model and IMPROVE data.

Biases in simulated sulfate from precursor and marine intrusion are investigated in the revised manuscript.

The bias in OC is because SOA processes are not included in our simulation. It is still challenging to correctly represent SOA processes in regional climate models. We keep our current settings and discuss the impacts of SOA processes in the revised manuscript.

The ammonium is included in Fig. 4d of the revised manuscript. The performance of simulated ammonium is similar to nitrate.

"Other" refers to the difference of PM2.5 and the summation of specified PM2.5 (NO3, NH4, SO4, OM, EC, dust). It is clarified in the revised

manuscript. In the model, it includes sea salt and other inorganic matter simulated in MOSAIC. In IMPROVE, it includes all other aerosols observed.

3. Optical properties: It is also not clear how AOD and aerosol extinction are computed from the simulated aerosol mass. Is aerosol microphysics package used for calculating particle sizes and mixing state? How is mass-based aerosol converted to extinction and AOD? Is the relative humidity considered in these calculations?

Description of how AOD and aerosol extinction are computed is added in the revised manuscript and attached as follows. More details can be found in Barnard et al. (2006, ACP).

"Aerosols are considered to be spherical and internally mixed in each bin (Barnard et al., 2006; Zhao et al., 2013b). The bulk refractive index for each particle is calculated by volume averaging in each bin. Mie calculation as described by Ghan et al. (2001) is used to derive aerosol optical properties (such as extinction, single-scattering albedo, and the asymmetry parameter for scattering) as a function of wavelength."

4. Chemistry: Nitrate, sulfate, and a significant fraction of OC are secondary aerosols that are produced by chemical reactions of their gaseous precursors in the atmosphere. The authors attribute the high bias of model-simulated nitrate to "high bias in nitrate emission", which is erroneous. The diagnostics should involve investigations of nitrate precursors such as NOx and HNO3, and also the formation of nitrate via heterogeneous reactions on dust and sea salt surfaces and homogeneous reactions in the sulfate-nitrate-ammonium system. It is not clear how WRF-Chem deals with nitrate formations and which is the major reaction pathway for nitrate aerosol production.

Same as sulfate – it is formed via gas and aqueous phase reactions of SO2. Better diagnostics of the problem is needed.

Thanks for the comments. Analyses of $NO_2$ and $SO_2$ are included in Fig. 6 of the revised manuscript. We also notice that switching the PBL scheme can produce better simulation of nitrate. More diagnostics of model biases are included in section 4 of the revised manuscript.

[Figure]

Figure 1. (a) NO$_2$ and (b) SO$_2$ from EPA (OBS) and the 20km run at Fresno, CA.

5. Other physical processes: Dry and wet depositions are the major removal processes for aerosols. The seasonal cycles of these processes also need to be investigated. For example, can the differences in seasonal variations between model and obs be partly explained by the differences in simulated and measured precipitation amount that determines the wet removal of aerosols? Or if the winds are realistically simulated in WRF-Chem that not only affect the dust emission, but also advection, both have profound effect on aerosol temporal and spatial distributions?

6. Meteorological fields: The only meteorological field compared in the paper is the equivalent potential temperature, which provides information on the atmospheric stability. Other important met fields, such as precipitation and wind speed/direction, as mentioned above, plays key roles in aerosol removal, transport, and wind-driven emissions of dust and sea salt but have not even mentioned in the paper. In addition, these fields and the physical processes driven by them are resolution-dependent, so the role of these met fields should be examined at different spatial resolutions.

The seasonal variability of precipitation is well captured in the simulations, while the magnitude of precipitation is smaller than the observations during the warm season (Supplementary Table 2). Wet removal processes are thus not likely the primary reason for the aerosol biases in the warm season.

The model simulations underestimate wind speed in the cold season (Figure 9 in the revised manuscript). In the warm season, the 20km run underestimates wind speed except June while the 4km run overestimates wind speed, which indicates wind speed is not likely the main reason for AOD biases in the warm season.

Discussions of the impacts from precipitation, wind speed and other factors are included in section 4.3 of the revised manuscript.

[Figure]

Figure 2. Monthly mean of (a) 2-m temperature (°C); (b) 2-m relative humidity (%); (c) 10-m wind speed (m/s); (d) precipitation (mm/day) at Fresno, CA. The 20km run (not shown) is similar to the 20km_D2 run while the 4km run (not shown) is similar to the 4km_D2 run.

7. Lateral boundary conditions: The effects of lateral boundary condition should be examined, or at lease discussed, particularly because of SJV's geophysical locations that is susceptible to the transpacific transport. How much of the aerosol species and their precursor gases are regionally/locally produced vs. imported from the lateral boundary, and how they affect the seasonal cycle? In other words, are the features/problems mainly produced by WRF-Chem? How important is the lateral boundary conditions to different aerosol species?

The simulated aerosol extinction in the free troposphere above the boundary layer is close to or larger than CALIOP, suggesting that aerosols transported from remote areas through chemical boundary conditions (e.g., the differences between the 20km_BC1 and 20km_D2 runs in Supplementary Fig. 3) may not be the major factor contributing to the underestimation of dust in the boundary layer in the SJV. It is clarified in the revised manuscript. The impacts of the lateral boundary conditions to different PM2.5 species are small except SO4 (as shown in the following figure).

[Figure]

Figure 3. Vertical distribution of seasonal mean 532 nm aerosol extinction coefficient (km$^{-1}$) from CALIOP, CALIOP_nodust, and the WRF-Chem

(20km_D2, 20km_BC1 and 20km_NEI11) simulations over the red box region in Fig. 1a in WY2013.

[Figure]

Figure 4. Aerosol mass (μg m-3) for different species from OBS, the 20km_D2, 20km_BC1 and 20km_NEI11 simulations at Fresno, CA. NH4 observations are from EPA; other observations are from IMPROVE. PM2.5_NO3 represents NO3 with diameter ≤ 2.5 μm. Similar definition for NH4, EC, OM, SO4 and dust in the figures.

8. Emissions: It seems the anthropogenic and biomass burning emissions used in this work are not up to date. For example, why the authors choose to use NEI05 emissions instead of more recent ones (e.g., NEI 2011 or NEI 2014) to better match the simulated time period (2012-2013)? Why GFEDv2 is preferred instead of GFEDv3 that was released a few years ago or GFEDv4 that has been available since 2015?

The 2011 NEI was not available in the WRF-Chem emission datasets when we initiated this study. We have run two sensitivity experiments with the 2011 NEI (20km_NEI11) and 2005 NEI (20km_D2) at 20 km resolution with the DUSTRAN dust scheme. As shown in Fig. 4 and 5 here, the differences between NEI11 and NEI05 are small comparing to the identified model biases in this study.

[Figure]

Figure 5. Spatial distribution of seasonal mean 550 nm AOD from 20km_NEI11 (NEI11) and 20km_D2 (NEI05) in WY2013.

We use the standard emission preparation program (prep_chem_sources_v1.5) for the WRF-Chem model to generate our fire emissions. Currently, only GFEDV2.1 is available in this program. Since fire emissions are not the major issues in our current simulations, we keep current settings.

9. Model-data comparison: 1) For AOD, there is only one AERONET site in the study region, and MISR's spatial coverage is limited. Why not use MODIS, which has a much better spatial coverage to have a better representation of "monthly average", in addition or even instead of using MISR?
We have compared the MISR data with the MODIS dark target and deep blue combined AOD V6 (as shown in the following figure). The MODIS data at 1°x1° cannot resolve the sharp gradient of aerosols in the SJV.

[Figure]

Figure 6. Seasonal mean AOD from MODIS and MISR.

2) Which months are defined as "cold" or "warm" months?
Cold months are from October to March; warm months are from April to September. The descriptions are in Line 277 and 282 in the revised manuscript.

3) More statistical quantities are needed to mark the agreement between model and observations, including correlation coefficients and seasonal/annual bias.
Correlation coefficients are included in the revised manuscript. More quantitative information are provided in the revised manuscript.

4) The authors should avoid using the subjective adjectives, such as "good agreement", "reasonably well", etc., to describe the comparisons between model and observations. More objective and quantitative methods and presentations are needed.
Following your suggestions, more objective and quantitative presentations are included in the revised manuscript.

5) Given that air quality changes quite a bit day to day and air quality forecasts are given on daily bases, why all the comparisons are done on monthly time scale instead of daily or sub-daily?
One of our goals is to evaluate model performances in simulating regional climate on the subseasonal-to-seasonable time scale. Many previous studies

have evaluated the performance of WRF-Chem in daily or sub-daily scale. It is not the focus of this study.

10. The most important step forward is to understand the causes of deficiencies in the model and suggest/incorporate improvements for better results. However, the current paper does not offer those aspects. Following three reviewers' comments, more analyses about the causes of deficiencies in the model are included in section 4 of the revised manuscript. We summarize the model sensitivities in section 5 and indicate future directions for improvements.

**Specific comments**
Page 5, line 72-82: I wonder why Fast et al 2014 and Zhao et al 2013 were able to "reasonably" represent the observations with the same WRF-Chem model, either in the warm months (Fast) or on annual bases (Zhao), but this work has difficulties to do the same?
The WRF-Chem simulation is sensitive to various factors such as initial and boundary conditions, model parameterizations and emission sources. The performance of the WRF-Chem model are also different in different seasons and at different locations. Because we are focusing on different seasons and/or different locations, we can see different performances of the model simulations. Some sensitivity experiments are included in the revised manuscript to provide more in-depth analyses on model results.

Page 5, line 83: I don't think the word "extend" is appropriate – this study only focuses on SJV while Fast and Zhao showed large regions in CA.
Reworded as "we focus on simulating aerosol seasonal variability in the SJV, California using similar model configurations as that used in Fast et al. (2014) and Zhao et al. (2013b)."

Page 6, line 102-104: I don't get it – why simulation for SJV is critical to MAIA? Is MAIA only focuses on SJV?
SJV is a testbed for the MAIA retrieval algorithm development. It is clarified in the revised manuscript.

Page 7, line 116: Why are the original wavelength(s) from AEORNET that you used to interpolate to 550 nm?
AERONET AOD is interpolated to 0.55 µm from 0.50 µm and 0.675 µm. It is clarified in the revised manuscript.

Page 8, line 146: What does "speciated" mean here? There is no aerosol species information from the CALIOP data. Marine, polluted continental, etc. provided by CALIOP are aerosol types, not species.
Reworded as "Level 2 532 nm aerosol extinction data classify aerosols into 6 types" in the revised manuscript.

Page 9, line 179-180: How is convective transport (and removal) of aerosols simulated in 4-km resolution?
Convective transport (and removal) of aerosols are simulated at grid-scale in 4-km resolution. It is clarified in the revised manuscript.

Page 9-10, line 183-184: Was the overestimation by MOZART in the free troposphere a factor of 2 such that the concentrations had to be divided by 2? If the overestimation was only in the free troposphere, why the concentrations in the lower atmosphere and BL were also divided by 2?
The overestimation by MOZART is mainly in the free troposphere as shown in Fast et al. (2014) and our sensitivity experiment (20km_BC1). Lowering the boundary conditions of aerosols concentration by 50% greatly reduced the bias in simulated AOD for all regions of California. The impact of chemical boundary conditions at the surface is small in the SJV. For simplicity, all the boundary conditions by MOZART are divided by 2.

Page 10, line 198: Are the dust emissions in the GOCART and DUSTRAN also available in 20 and 4 km resolutions? What are the major differences between GOCART and DUSTRAN schemes?
Yes. More descriptions of GOCART and DUSTRAN schemes are included in last two paragraphs of section 3 in the revised manuscript.

Page 11, first paragraph in section 4.1: What PM2.5 species and precursor gases are emitted?
Nineteen gases (including SO2, NO, NH3 etc.) are emitted, while aerosol emissions include SO4, NO3, EC, organic aerosols, and total PM2.5 and PM10 masses. It is clarified in the revised manuscript.

Have you checked the domain budget between 4 and 20 km resolution to ensure the total emission for all species are identical with these different resolutions?
Yes, they are quite similar. Mean emission rates for the 4km and 20km runs are listed in Fig. 1 in the revised manuscript.

Page 11, line 215: How was AOD calculated without having information of PM2.5 composition? For example, dust and BC have very different mass to extinction conversion factor, known as mass extinction efficiency (MEE). There is no single MEE for a generic PM2.5 or PM10.

Aerosol composition is considered in AOD calculation. Different refractive index are assigned to different particles. Description of how AOD and aerosol extinction are included in the revised manuscript as the following.

"Aerosols are considered to be spherical and internally mixed in each bin (Barnard et al., 2006; Zhao et al., 2013b). The bulk refractive index for each particle is calculated by volume averaging in each bin. Mie calculation as described by Ghan et al. (2001) is used to derive aerosol optical properties (such as extinction, single-scattering albedo, and the asymmetry parameter for scattering) as a function of wavelength."

Page 12, line 237: As I said earlier, nitrate is not emitted but chemically produced. The precursor emission/concentration/transport/chemistry have to be examined to explain the nitrate.

$NO_3$ is included in PM2.5 emission dataset. $NO_2$, one precursor of $NO_3$, is evaluated in the revised manuscript.

Page 12, line 238: Why is simulation over Texas relevant here?

This discussion is removed.

Page 12, line 242: Be specific on what "SOA processes" is referred here.

This sentence is removed in the revised manuscript because SOA processes are not simulated in our settings.

Page 12, line 244 and 246: Be quantitative – what is the standard of "good agreement"?

Quantitative evaluations are provided in the revised manuscript.

Page 12, line 250: How large is the "large low bias"?

From 30% to 85%. It is clarified in the revised manuscript.

Page 13, line 253-254: "The 4km simulation has better agreement…", but only in the cold season.

It is clarified in the revised manuscript.

Page 13, line 254-255: "The 4km simulation captures seasonal variability of PM2.5 and its speciation": From Figure 4, the seasonal variability for the

PM2.5 species are very similar between the 4- and 20-km simulations, only the concentrations are higher from the 4km simulation. The seasonal variability of PM2.5 sulfate and OC are not capture by both 4 and 20 km simulations.

The seasonal variability of sulfate is not captured in the 4km simulation while 20km simulation has a correlation of 0.63. OM has a correlation of 0.93 for all the simulations. Reworded as "Both the 20km and 4km simulations approximately capture the seasonal variability of PM2.5 and most of its speciation" in the revised manuscript.

Page 13, line 267-268: The 4km_D2 overestimates PM2.5 by 52%, but it overestimates the PM2.5_dust by up to a factor of 4 in the warm season!
The quantitative information is provided in the revised manuscript.

Page 13, line 270-272: As I suggested earlier, please show correlation coefficients on all comparisons (in addition to the bias), which indicates how model and data agree on seasonal variations.
Correlations are provided in the revised manuscript.

Page 14, line 285-286: How much better does 4km_D2 agree with MISR than other simulations? Visually, JAS is still nowhere near MISR, and AMJ is higher than MISR. Please quantify the degree of agreement.
Quantitative information is provided in the revised manuscript.

Page 14, line 290-292: I don't understand the statement of "reasonably capture the vertical distribution", even though the model has "low biases in the boundary layer and high biases in the free troposphere". To me, this is rather "unreasonable".
Reworded as "roughly capture".

Page 15, line 298-299: "…suggesting relative good performance…": How good? Figure shows poor agreement between obs and model for sulfate and OC, so they are not "good" at all.
Reworded as "suggesting that dust is the primary factor contributing to the model biases in aerosol extinction" in the revised manuscript.

Page 15, line 303: How to explain that dust from 4km_D2 is way too high but the extinction in the boundary layer is still way too low?
The model doesn't simulate the unstable environment in the warm season. Although the dust emission at the surface is large in the 4km_D2 run, no

enough convective vertical mixing is produced in the simulations, resulting the low biases in the boundary layer. It is clarified in the revised manuscript.

Page 15, line 313 and 316: If the model has weak vertical mixing, the aerosols should be trapped within the BL and not transported to high altitudes. But the model actually overestimates the aerosol at high altitude – what is the source of high altitude aerosol?
High altitude aerosols are from horizontal transport primarily governed by chemical boundary conditions.

Page 16, line 321-322: This precisely indicates the need to quantify the role of chemical boundary conditions.
The role of chemical boundary conditions is discussed in the revised manuscript.

Page 16, line 323-324, "good performance…": But in JFM the model results are much higher (by a factor of infinity?) at above 1.5 km! How can that be evaluated as "good"?
Changed to "relatively good".

Page 16, line 330: "reasonable simulation", "good representation" – what are the measures of reasonable and good here?
Quantitative information are provided in Table 2 and 3 the revised manuscript.

Page 16, line 337: Please explain what "climatological fire emissions" mean.
Reworded as "monthly-varying fire emissions".

Page 16, line 339-340: Why can Wu et al do it right for South America fire but cannot do it for California? What are the major obstacles?
In our simulation for South America, it is a 7-day case. Daily satellite data are used to generate biomass burning emission. In this study, we are focusing on seasonal variations. Biomass burning emission is updated every month, which cannot capture the single fire event in this case.

Page 17, line 371-372: No need to spell out what GOCART and DUSTRAN stand for at the last part of the paper, since they have been introduced and used many times earlier in the text.
Most people don't read the whole paper, especially program managers. So we have all acronyms redefined to help them immediately understand what we are saying.

Page 17, line 383-385: Unfortunately, I cannot see how the evaluation in this study can be apply to other regions to ensure that aerosols are simulated correctly for the right reasons. This paper has shown the problems but has not shown how to solve the problems with what approach.

This sentence is removed in the revised manuscript.

---

## Author Response (AR2)

Comments to the Author:
Dear Authors,

I appreciate your efforts to conduct additional simulations and in-depth analyses to address the reviewers comments. I can see the manuscript is much improved with the revision.

The new reviewer report raises additional comments on the evaluations of model temperature profiles using the AIRS data, model design using the PBL schemes, and evaluation of profiles average of daytime and nighttime, etc. These comments are helpful for model diagnoses of aerosol and chemistry biases due to meteorology.

Please address these comments carefully.

Dear Editor,

Thank you for your consideration of the submitted manuscript. We have revised the manuscript in response to the reviewer's comments. The major changes are:
1. We have added the ERA-Interim reanalysis data to evaluate the vertical profiles of the model simulations.
2. Specific humidity profiles are added in the Supplementary Fig. 9.

The point-by-point response is listed below.

Sincerely,

Longtao Wu

**Reviewer #4**

While the other reviewers commented on the chemistry aspects, I will mostly focus on "the role of meteorology" and vertical mixing. Unfortunately, most of the discussion regarding vertical mixing is not robust (or highly questionable). Also I am surprised none of previous conclusions regarding PBL schemes (or vertical mixing treatments) in WRF are used to guide/help the investigation in this study.

Major comments

As one of the reviewers pointed out, AIRS profiles may not be appropriate to evaluate simulated profiles in the boundary layer. The fatal issue is that the AIRS profiles are not consistent with the surface observation. See LN438-439, the model overestimates surface temperature throughout the year comparing with surface observation. However, comparing with the AIRS profiles, the model only overestimates temperature near the surface in cold months while it underestimates temperature near the surface in warm months (Fig. 13). Such inconsistency may suggest AIRS profiles near the surface may not be reliable.

The vertical profiles from the ERA-Interim reanalysis dataset are added in the revised manuscript. Please see Fig. 13, Supplement Figs. 7, 8 and 9 in the revised manuscript for the comparison. Although there are some differences between AIRS and ERA-Interim, our conclusions are unchanged.

As shown in Figure 13, ACM2 predicts more stable boundary layer, how would more stable boundary layer leads to simulate lower surface NO3 and NH4 as discussed in LN473-477? Higher stability (particularly in cold season) should lead to more accumulation of pollutants near the surface.

The stability difference is quite small between ACM2 and YSU. We suspect the differences in surface NO3 and NH4 are not due to atmospheric stability changes. Figure 15 shows that more aerosols are transported above the surface in the 20km_P7 (ACM2) than in the 20km_D2 (YSU). The difference may be due to different parameterization methods of chemical transport in the PBL scheme. It is discussed in LN485-486.

The current evaluation of vertical profiles uses average of daytime and nighttime. The boundary layer structure is totally different during daytime and nighttime. I am not sure what the comparison of averaged profiles across daytime and nighttime really tell.

This study focuses on the seasonal variability of aerosols. Diurnal variability is beyond the scope of this study. The simulation of aerosol diurnal variability in California can be found in Fast et al. (2014).

Fast, J. D., Allan, J., Bahreini, R., Craven, J., Emmons, L., Ferrare, R., Hayes, P. L., Hodzic, A., Holloway, J., Hostetler, C., Jimenez, J. L., Jonsson, H., Liu, S., Liu, Y., Metcalf, A., Middlebrook, A., Nowak, J., Pekour, M., Perring, A., Russell, L., Sedlacek, A., Seinfeld, J., Setyan, A., Shilling, J., Shrivastava, M., Springston, S., Song, C., Subramanian, R., Taylor, J. W., Vinoj, V., Yang, Q., Zaveri, R. A., and Zhang, Q.: Modeling regional aerosol and aerosol precursor variability over California and its sensitivity to emissions and long-range transport during the 2010 CalNex and CARES campaigns, Atmos. Chem. Phys., 14, 10013-10060, doi:10.5194/acp-14-10013-2014, 2014.

Also the current study compares the different performance of YSU and ACM2. This might be a very poor choice for investigation of PBL schemes in WRF. Both YSU and ACM2 are nonlocal schemes and they have similar performance in most cases (comparing to the differences between local and nonlocal schemes). It is not clear to me how the ACM2 scheme performs better than the YSU scheme in this evaluation. Also both YSU and ACM2 schemes have different treatments for daytime and nighttime respectively, a more appropriate approach should be evaluating the daytime and nighttime performance separately.

Previous studies have shown that YSU and ACM2 have good performance in WRF and WRF-Chem simulations. For example, Hu et al. (2010) showed that "the YSU and ACM2 schemes give much less bias than with the MYJ scheme". Xie et al. (2012) concluded that "It is reasonable to infer that WRF, coupled with the ACM2 PBL physics option can be a viable producer of meteorological forcing to regional air quality modeling in the Pearl River Delta (PRD) Region". Cuchiara et al. (2014) showed that "the overall results did not indicate any

preferred PBL scheme for the Huston case. However, for ozone prediction the YSU scheme showed greatest agreements with observed values". Banks and Baldasano (2016) demonstrated that "the ACM2 scheme showed the lowest mean bias with respect to surface ozone at urban stations, while the YSU scheme preformed best with simulated nitrogen dioxide. The ACM2 and BouLac schemes performed better than the YSU scheme for air quality simulations." Banks et al. (2016) concluded that "non-local PBL schemes give the most agreeable solutions when compared with observations". Chen et al. (2017) showed that "as for the PM2.5 simulation, the combination of the YSU PBL, Goddard SW and GFDL LW schemes showed the greatest consistency with the observed values". It is clarified in the revised manuscript as "Previous studies showed that both the YSU and ACM2 schemes have good performance in simulating boundary layer properties (e.g., Hu et al., 2010; Xie et al., 2012; Cuchiara et al., 2014; Banks and Baldasano, 2016; Banks et al., 2016; Chen et al., 2017)."

The goal of this study is not evaluating which PBL scheme is better. The sensitivity experiment shown here is to demonstrate that the simulation of aerosols is sensitive to the PBL scheme. We have added some of the references above in the revised manuscript.

Reference:

Hu, X. M., J. W. Nielsen-Gammon, and F. Zhang (2010), Evaluation of three planetary boundary layer schemes in the WRF model, J. Appl. Meteorol. Climatol., 49(9), 1831–1844, doi:10.1175/2010JAMC2432.1.

Xie, B., J. C. H. Fung, A. Chan, and A. Lau (2012), Evaluation of nonlocal and local planetary boundary layer schemes in the WRF model, J. Geophys. Res., 117, D12103, doi:10.1029/2011JD017080.

Cuchiara, G.C., Li, X., Carvalho, J., & Rappenglück, B. (2014). Intercomparison of planetary boundary layer parameterization and its impacts on surface ozone concentration in the WRF/Chem model for a case study in Houston/Texas. Atmospheric Environment, 96,175–185. http://dx.doi.org/10.1016/j.atmosenv.2014.07.013

Banks, R.F., Baldasano, J.M., 2016. Impact of WRF model PBL schemes on air quality simulations over Catalonia, Spain. Science of the Total Environment, 572, 98-113, http://dx.doi.org/10.1016/j.scitotenv.2016.07.167

Banks, R. F., J. Tiana-Alsina, J. M. Baldasano, F. Rocadenbosch, A. Papayannis, S. Solomos, and C. G. Tzanis (2016), Sensitivity of boundary-layer variables to PBL schemes in the WRF model based on surface meteorological observations, lidar, and radiosondes during the HygrA-CD campaign, Atmos. Res., 176, 185–201.

Chen, D., X. Xie, Y. Zhou, J. Lang, T. Xu, N. Yang, Y. Zhao and X. Liu (2017) Performance Evaluation of the WRF-Chem Model with Different Physical Parameterization Schemes during an Extremely High PM2.5 Pollution Episode in Beijing. Aerosol and Air Quality Research, 17:262-277. doi: 10.4209/aaqr.2015.10.0610

LN454-456, The discussion may be wrong. Fig. 13c indeed shows a neutral (or slightly stable) boundary layer below 3 km AGL. This does not mean the model cannot capture the well-mixed boundary layer. Actually in the convective boundary layer, the observed profile of potential temperature is indeed slightly stable [Deardorff, 1972], that is why some PBL schemes (e.g.,

YSU) added the countergradient term to make the simulated profiles in the convective boundary layer slightly stable [Frech and Mahrt, 1995]. Again, I am surprised none of the previous efforts in terms of PBL scheme evaluation is surveyed before the numerical experiments and during the writing of the manuscript.

Figure 12c shows that the 4km_D2 experiment doesn't capture the well-mixed boundary layer of aerosols observed by CALIOP. While the AIRS observation and ERA-interim data show conditionally unstable lower troposphere which favors upward displacement of surface aerosols, the simulated stable boundary layer would limit the uplifting of aerosols from the surface, contributing to the low biases in the aerosols in the boundary layer. We think the misrepresentation of boundary layer stability is one source of errors for the discrepancies in the simulated aerosol profiles. We have added discussions of previous studies on PBL schemes in LN 221-224.

Vertical profile of RH is not a good choice to evaluate different vertical mixing treatments. Instead, specific humidity should be used for PBL evaluation.

Specific humidity is included in the Supplementary Fig. 9 in the revised manuscript. All the simulations show dry biases near the surface in the warm season comparing to ERA-Interim. However, it cannot explain the low bias of dust above the surface (0.3 – 3 km) relative to CALIOP measurements.

[Figure]

Supplementary Figure 9. Vertical profile of seasonal mean specific humidity (g/kg) in the WRF-Chem simulations, AIRS and ERA-Interim. The 20km run (not shown) is similar to the 20km_D2 run while the 4km run (not shown) is similar to the 4km_D2 run.

In summary, the current design of numerical experiments in terms of role of vertical mixing treatments and analysis of the results are not adequate to diagnose the model errors associated with predicted chemical species.

Other specific comments:
LN438-439, warm and dry biases near the surface usually mean too strong vertical mixing rather than "not enough convective vertical mixing".

A warm bias near the surface promotes convective vertical mixing, but a dry bias prohibits it. Combining the effects of temperature and humidity, we compute the equivalent potential temperature ($\theta_e$) (Fig. 13) that shows the convective instability of the atmospheric profiles. The discrepancy in $\theta_e$ between the observations and model simulations is quite clear. We think the neutral or slightly stable profiles in the model limit the uplifting of aerosols from the surface, contributing to the low biases of simulated aerosols in the boundary layer. We have clarified this in the revised manuscript. Text revised to clarify.

LN454, In Fig. 13c, I actually see unstable layer below 3km rather than "1.5km"

The ERA-Interim shows a neutral layer between 1.5 and 3 km.

LN457-458, The logic is confusing. "not enough convective vertical mixing" should result in high biases of simulated pollutants near the surface, rather than "low biases"

The simulated pollutants near the surface are biased high in the 4km_D2 experiment. The low biases are between 0.3 km and 3 km. It is clarified in the revised manuscript.

LN461-462, it is not clear, how vertical redistribution of pollutants by vertical mixing could change the column-integrated AOD.

Column-integrated AOD is the integral of aerosol extinction at each layer over the atmospheric column. As shown in the aerosol extinction profiles in Fig. 12, the simulations have low biases above the surface (0.3-3.0 km) compared to CALIOP. We think the simulated stable lower troposphere limits the vertical displacement of pollutant transport above the surface, which contributes to the low biases of aerosol extinction and AOD in the simulations. Text revised to clarify.

LN468-469, In terms of vertical gradient of equivalent potential temperature, I don't think the model did better in cold season than warm season. In cold season, particularly OND, the model significantly underestimates stability (Fig. 13a).

Comparing to ERA-Interim, the vertical gradient of equivalent potential temperature is well simulated in the cold season. In the warm season, both ERA-I and AIRS observed unstable environment while the model simulations produce neutral or stable lower troposphere.

LN478-480, I don't think you have proved vertical mixing of ACM2 is stronger than YSU. Fig. 13 actually shows the opposite.

The stability difference is quite small between ACM2 and YSU. We suspect the differences in surface NO3 and NH4 are not due to atmospheric stability changes. Figure 15 shows that more aerosols are transported above the surface in the 20km_P7 (ACM2) than in the 20km_D2 (YSU). The difference may be due to different parameterization methods of chemical transport in the PBL scheme. It is discussed in LN485-486.

LN484-486, how would "more conducive convective vertical transport in the PBL scheme" could lead to increase of aerosol in the boundary layer?

We mean "aerosol above the surface", as shown in Fig. 15 that more aerosols are uplifted above the surface in the 20km_P7 (ACM2) than in the 20km_D2 (YSU). It is clarified in the revised manuscript. Text revised to clarify.

LN492-493, why would "stable environment" lead to low biases of aerosol in the boundary layer and column-integrated AOD? "stable environment" should lead to accumulation of aerosol near the surface. Again, it is not clear how vertical re-distribution of aerosol would change column-integrated AOD.

You are right that stable environment leads to accumulation of aerosol immediately near the surface, resulting in low biases of aerosol above the surface between 0.3 and 3 km. The low bias in the boundary layer thus contributes to the low bias in the column-integrated AOD as the concentrations of aerosols above 3 km are very small. We have clarified this in the manuscript.

LN498-499, in Fig. 13, I don't think ACM2 performed better in terms of boundary layer structure than YSU.

Figure 15 shows more aerosols are transported above the surface in ACM2 than in YSU.

I saw three routine soundings in CA (http://weather.uwyo.edu/upperair/sounding.html), are you sure they are not in your domain?

The three sites are Edwards (34.90°N, 117.92°W), Vandenberg (34.75°N, 120.57°W), Oakland INT AP (37.75°N, 122.22°W). None of them are located in the region discussed in this paper (the SJV or the red dashed box in Fig. 1a).

References:
Deardorff, J. W. (1972), Theoretical expression for the counter gradient vertical heat flux, Journal of Geophysical Research, 77(30), 5900-5904.
Frech, M., and L. Mahrt (1995), A 2-Scale Mixing Formulation for the Atmospheric Boundary-Layer, Bound-Lay Meteorol, 73(1-2), 91-104.

---

## Author Response (AR3)

Comments to the Author:
The manuscript is much improved in the revision in responding to the reviewer's comment.

Please refer the following paper in your manuscript (in Page 5, Lines 67-72) related to the WRF-Chem simulation for the meteorology effects relative to chemistry effects determine the air quality in China, as it is one of the applications of WRF-Chem.

Gao, Y., X. Liu, C. Zhao, and M. Zhang (2011), Emission controls versus meteorological conditions in determining aerosol concentrations in Beijing during the 2008 Olympic Games, Atmospheric Chemistry and Physics, 11, 12437-12451.

please don't use "x" instead of the symbol of multiplication in the case of resolution, e.g., 1.1 km x 1.1 km.

Dear Editor,

Thank you for accepting this manuscript for publication at ACP. Gao et al. (2011) is cited in the updated manuscript. The symbol of multiplication is used to replace the "x" in the case of resolution.

Sincerely,

Longtao Wu